# Online Differential Privacy Bayesian Optimization with Sliced Wasserstein Compression

## Abstract

The increasing prevalence of streaming data and rising privacy concerns pose significant challenges for traditional Bayesian optimization (BO), which is often ill-suited for real-time, privacy-aware learning. In this paper, we propose a novel online locally differentially private BO framework that enables zero-order optimization with rigorous privacy guarantees in dynamic environments. Specifically, we develop a one-pass Gaussian process compression algorithm based on the sliced Wasserstein distance, which effectively addresses the challenges of kernel matrix scalability, memory efficiency, and numerical stability under streaming updates. We further establish a systematic non-asymptotic convergence analysis to characterize the privacy–utility trade-off of the proposed estimators. Extensive experiments on both simulated and real-world datasets demonstrate that our method consistently delivers accurate, stable, and privacy-preserving results without sacrificing efficiency.

## 1 Introduction

Bayesian optimization (BO) (Močkus, 1974; Jones et al., 1998) is a sample-efficient framework widely used for the global optimization of expensive, non-convex, or black-box functions, with applications in hyperparameter tuning, robotics, and scientific discovery (Snoek et al., 2012; Berkenkamp et al., 2023). In particular, BO iteratively selects query points using a probabilistic surrogate model and balances exploration and exploitation through the predictive mean and uncertainty, often achieving high-performance solutions with relatively few evaluations. To date, BO has been extensively studied, leading to numerous methodological advances, including local descent strategies (Müller et al., 2021; Nguyen et al., 2022), mixed-space optimization techniques (Neiswanger et al., 2022), scalable acquisition via Monte Carlo methods (Balandat et al., 2020), and extensions to iterative and bilevel problems (Fu et al., 2024), supported by theoretical analyses of high-dimensional Gaussian processes (Hvarfner et al., 2024). Furthermore, practical robustness has been enhanced through improved constraint handling (Nguyen et al., 2024), contextual uncertainty modeling (Tay et al., 2024), and meta-learning strategies for rapid adaptation (Ravi & Beatson, 2019).

Building on this line of work, several methods have sought to accelerate convergence by incorporating gradient information via finite differences or kernel-based estimation (Wu et al., 2017; Eriksson et al., 2019). For example, Müller et al. (2021) reformulated BO as an approximate gradient descent procedure, a formulation later extended by the gradient information BO framework (Wu et al., 2023), which reduces gradient uncertainty and guarantees convergence to low-gradient regions in reproducing kernel Hilbert spaces (RKHS). More recently, Sopa et al. (2025) adapted these methods to tackle high-dimensional problems. Nonetheless, the aforementioned BO methods remain predicated on static datasets and are not designed for streaming environments, thereby limiting their applicability in dynamic and continually evolving settings.

Real-time systems, such as IoT edge devices, dynamic pricing platforms (e.g., Uber surge pricing), and credit card fraud detection—produce large volumes of streaming data and require timely decisions while protecting sensitive information (e.g., locations, transactions, personal attributes). This motivation is reflected in our real-data analyses, including Uber price prediction and credit card fraud detection. In such settings, privacy protection is essential: Uber trip records contain highly sensitive location and behavioral data, and training models without privacy safeguards risks regulatory vio-

lations and loss of user trust. At the same time, data arrive continuously at scales too large for full storage, and models must be updated in near real time to remain accurate. Ignoring the streaming nature of the data and relying solely on offline batch training leads to rapidly outdated models as demand patterns or fraud strategies shift, resulting in degraded predictive performance. However, traditional Bayesian optimization methods are ill-suited for these scenarios: their computational cost grows as $\mathcal{O}(t^3)$ with the number of observations $t$, making them infeasible for high-frequency, large-scale data streams. They also assume access to a static dataset, rendering them incompatible with online settings where data arrive continuously. In contrast, our online Bayesian optimization framework under LDP is designed for streaming environments, provides per-iteration LDP guarantees, and maintains real-time computational efficiency.

The growing demand for real-time decision-making in streaming data environments has elevated online learning to a central paradigm, with stochastic gradient descent (SGD) serving as its primary optimization tool (Robbins & Monro, 1951; Bottou, 2010). Recent advances have extended SGD beyond classical settings to a variety of estimation settings, including online learning (Su & Zhu, 2023; Xie et al., 2025), contextual bandits (Ding et al., 2021), and high dimensional inference tasks (Han et al., 2024). Yet these methods remain rooted in the frequentist paradigm and rely heavily on heuristic exploration, and depend on gradient access, which constrains data efficiency and often results in slow convergence in complex, non-convex functions (Ruder, 2016). By contrast, BO does not require gradient information and provides a principled framework for balancing exploration and exploitation, thereby enabling more sample-efficient optimization in such settings (Jones et al., 1998). From a Bayesian standpoint, online learning has largely been investigated in sequential decision-making contexts, such as hyperparameter tuning (Snoek et al., 2012), black-box optimization (Frazier, 2018), and sequential hypothesis testing (She et al., 2021), but these methods typically emphasize decision efficiency over functional exploration and often lack expressive input–output modeling beyond classification. Consequently, they are ill-suited for streaming environments, where adaptive and sample-efficient exploration of the response surface is essential, highlighting the need for a scalable BO framework explicitly designed for online settings.

On the other hand, the increasing complexity and scale of data amplify the challenges of safeguarding individual privacy and sustaining public trust, particularly in applications that involve sensitive user information, such as financial transactions in banking or location data from mobile applications. Differential Privacy (DP) (Dwork, 2006; Dwork et al., 2014), one of the most widely adopted frameworks for privacy-preserving data analysis, provides a rigorous guarantees the output of a computation does not reveal sensitive information about any individual in the dataset. DP is typically implemented under two models: central DP (CDP), where a trusted server injects noise into aggregated data (Ponomareva et al., 2023), and local DP (LDP), where users privatize their data before sharing, thereby removing the need for a trusted server (Duchi et al., 2018; Lowy & Razaviyayn, 2023; Duchi & Ruan, 2024). Although substantial advances in both paradigms, most existing methods continue to be developed within the frequentist framework.

Recently, increasing attention has been devoted to privacy-preserving estimation in BO under the CDP framework. Early work by Heikkilä et al. (2017) proposed a distributed DP-Bayesian learning method that leverages secure multi-party aggregation and Gaussian mechanisms for efficient privacy-preserving inference. Subsequently, Dimitrakakis et al. (2017) introduced a Bayesian DP framework based on posterior sampling, establishing sensitivity bounds for arbitrary data metrics. Building on this foundation, Triastcyn & Faltings (2020) incorporated distributional information to provide more practical privacy guarantees, while Zhang & Zhang (2023) further advanced the line of research by designing an exact and efficient DP Metropolis–Hastings algorithm. In parallel, Li et al. (2023) investigated DP synthetic data generation using Bayesian networks and established statistical accuracy guarantees for marginal-based methods. Makhija et al. (2024) developed a federated Bayesian learning framework that trains personalized models across clients with rigorous DP guarantees, and Chew et al. (2025) introduced a risk-weighted pseudo-posterior distribution to address imbalanced data in DP deep learning. More recently, Sopa et al. (2025) proposed a DP gradient-informed BO method for high-dimensional problems with exponential convergence guarantees. Despite these advances, existing methods are primarily designed for batch learning and typically assume a trusted data curator. To the best of our knowledge, no scalable and statistically rigorous method has yet been developed for online BO under the LDP framework. This gap naturally motivates the following fundamental question:

Table 1: A comparison of recent results on differential privacy BO.

| Reference | Method | DP | Bayesian |
|---|---|---|---|
| Triastcyn & Faltings (2020) | Offline | CDP | True |
| Zhang & Zhang (2023) | Offline | CDP | True |
| Sopa et al. (2025) | Offline | CDP | True |
| Duchi & Ruan (2024) | Offline | LDP | False |
| Xie et al. (2025) | Online | LDP | False |
| Proposed | Online | LDP | True |

*Is it possible to develop an online, gradient-free, Bayesian optimization framework that provides rigorous LDP guarantees without sacrificing statistical efficiency?*

The main goal of this paper is to address the question outlined above. To this end, we propose a fully online LDP framework for real-time BO. Specifically, we introduce a novel one-pass, online, gradient-free LDP-BO algorithm that integrates a Sliced Wasserstein Compression (SWC) strategy, which enables efficient kernel compression to control memory growth while simultaneously ensuring privacy-preserving learning in streaming data environments. An overview of the proposed framework is provided in Figure 1. A comparative summary of our method against representative recent works in differential privacy BO is provided in Table 1. For brevity, we include one example from each category of related methods. The key contributions of this work are summarized as follows:

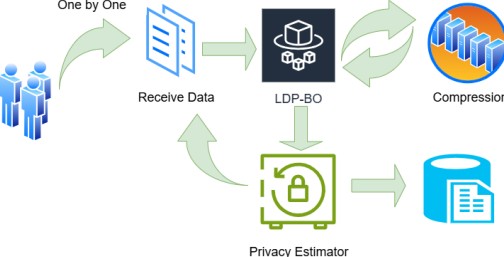

Figure 1: Flowchart of the proposed online privacy-preserving Bayesian framework. Data is processed sequentially, and privacy-preserving estimates are obtained using the LDP-BO algorithm. During this process, the kernel dictionary is compressed via the sliced Wasserstein distance to control memory growth.

- **Online LDP Bayesian estimation framework**: Our framework provides rigorous per-iteration LDP guarantees for BO in an online setting, thereby enabling privacy-preserving real-time estimation and addressing a key limitation of existing methods that typically require access to the entire dataset in dynamic environments. By constructing a surrogate model, we further develop a zeroth-order optimizer that eliminates the need for gradient information, making the framework well-suited for complex objective functions with non-differentiable points or discontinuities.

- **Efficient compression algorithm**: We propose an efficient compression algorithm based on the Sliced Wasserstein distance to manage the kernel dictionary in streaming data environments. The algorithm reduces memory overhead while preserving numerical stability, and we establish that the kernel dictionary size remains uniformly bounded, ensuring efficient BO without loss of model fidelity. Moreover, the proposed algorithm achieves $\mathcal{O}(1)$ time and space complexity per iteration. By eliminating the need to store or re-access historical data, our method avoids the $\mathcal{O}(t^3)$ computational cost and $\mathcal{O}(t)$ memory requirements inherent standard BO and inducing point-based batch methods.

- **Non-asymptotic analysis**: We establish non-asymptotic convergence rates for our estimator under decaying step sizes, addressing both strongly convex losses and the more general

smooth (but not necessarily convex) losses. The rates depend explicitly on the sample size, privacy budget, and BO compression error. Specifically, in the strongly convex setting, the estimation error achieves the same order as that of SGD, whereas under smoothness alone we provide guarantees of convergence to stationary points. Notably, our method achieves SGD-like convergence behavior without requiring access to exact gradients at any stage of the optimization process.

## 2 PROBLEM FORMULATION

In this paper, we consider an online learning framework in which independent and identically distributed (i.i.d.) observations $\{z_i\}_{i=1}^t$ with $t \geq 1$, arrive sequentially, where each $z_i = (x_i^\top, y_i)^\top$ consists of a covariate vector $x_i \in \mathbb{R}^p$ and a response $y_i \in \mathbb{R}$, jointly drawn from an underlying distribution $\mathcal{F}$. Specifically, we consider the following optimization problem:

$$\boldsymbol{\theta}^\star = \operatorname{argmin}_{\boldsymbol{\theta} \in \boldsymbol{\Theta}} \left( f(\boldsymbol{\theta}) := E_{\boldsymbol{z} \sim \mathcal{P}_{\boldsymbol{z}}}\left[\mathcal{L}(\boldsymbol{\theta}, \boldsymbol{z})\right] = \int \mathcal{L}(\boldsymbol{\theta}, \boldsymbol{z})\mathrm{d}\mathcal{P}_{\boldsymbol{z}}(\boldsymbol{z}) \right), \tag{1}$$

where $\mathcal{L}(\boldsymbol{\theta}, \boldsymbol{z})$ denotes a pre-specified loss function with respect to $\boldsymbol{\theta}$ and $\boldsymbol{z}$ is a random variable from the distribution $\mathcal{P}\boldsymbol{z}$.

We aim to estimate an unknown parameter $\boldsymbol{\theta}^\star$ from streaming data within the BO framework, where observations are received sequentially over time. The BO framework adopts a Gaussian process (GP) as a probabilistic surrogate model. By placing a GP prior with a twice-differentiable kernel $K$, the objective function $f$ can be efficiently approximated without explicit gradient computations. Given a collection of points $\mathcal{D} = \{\boldsymbol{\theta}_i\}_{i=1}^t$, the posterior distribution $f \mid \mathcal{D} \sim \mathrm{GP}(m_{\mathcal{D}}, K_{\mathcal{D}})$ yields closed-form estimates, while the gradient process $\nabla f \mid \mathcal{D}$ (Müller et al., 2021)

$$\nabla f(\boldsymbol{\theta}) \mid \mathcal{D} \sim N\left(\nabla m_{\mathcal{D}}(\boldsymbol{\theta}), \nabla^2 K_{\mathcal{D}}(\boldsymbol{\theta}, \boldsymbol{\theta})\right), \tag{2}$$

where

$$\nabla m_{\mathcal{D}}(\boldsymbol{\theta}) = \nabla m(\boldsymbol{\theta}) + \nabla K(\boldsymbol{\theta}, \mathcal{D})K(\mathcal{D}, \mathcal{D})^{-1}(f(\mathcal{D}) - m(\mathcal{D})),$$
$$\nabla^2 K_{\mathcal{D}}(\boldsymbol{\theta}, \boldsymbol{\theta}) = \nabla^2 K(\boldsymbol{\theta}, \boldsymbol{\theta}) - \nabla K(\boldsymbol{\theta}, \mathcal{D})K(\mathcal{D}, \mathcal{D})^{-1}\nabla K(\mathcal{D}, \boldsymbol{\theta}).$$

This procedure only depends on zeroth-order function evaluations, thereby eliminating the need for explicit gradient calculations. Since the true distribution $\mathcal{P}_{\boldsymbol{z}}$ is unknown, the expected risk $f(\boldsymbol{\theta})$ is intractable and is instead approximated by the empirical loss $\mathcal{L}(\boldsymbol{\theta}, \boldsymbol{z})$ based on observed data. For simplicity, we assume throughout this work that the prior mean function is zero, i.e., $m(\cdot) \equiv 0$.

Unfortunately, the standard BO framework suffers from two major limitations: (1) it does not scale to online learning, as the storage requirement for $\mathcal{D}$ grows unbounded as new data arrive sequentially, and (2) it is vulnerable to privacy breaches because repeated data queries during the optimization process may leak sensitive information, such as medical records (Liu et al., 2024) or consumer data (Hard et al., 2018). (Additional preliminaries on LDP are provided in Appendix A.1) To address these challenges, we propose GP-based BO framework to a privacy-preserving online setting that achieves computationally efficient estimation with reduced time and space complexity, while simultaneously providing rigorous individual-level privacy guarantees.

## 3 METHODOLOGY

In this section, we propose the online locally privacy-preserving estimation within the BO framework to the minimization problem (1).

### 3.1 ONLINE LOCALLY DIFFERENTIALLY PRIVATE BAYESIAN OPTIMIZATION

We first leverage BO to approximate the gradient of the underlying function defined in (1) through the gradient of a surrogate model. In particular, at each iteration, the BO procedure selects query points that minimize an acquisition function, thereby maximizing information gain in the optimization process (see Wu et al. (2023) for further details). In line with Müller et al. (2021), this paper adopts gradient information as the acquisition function, which is defined as

$$\mathrm{GI}(\boldsymbol{\xi}; \mathcal{D}, \boldsymbol{\theta}) = \mathrm{Tr}(\nabla^2 K_{\mathcal{D} \cup \boldsymbol{\xi}}(\boldsymbol{\theta}, \boldsymbol{\theta})), \tag{3}$$

where $\boldsymbol{\xi}$ denotes a candidate point in the parameter space $\boldsymbol{\Theta}$. This strategy minimizes the trace of the Hessian of the kernel, thereby reducing the uncertainty of gradient estimates. Furthermore, since the kernel $K$ is smooth and $\boldsymbol{\Theta}$ is compact, the acquisition function $\mathrm{GI}(\boldsymbol{\xi}; \mathcal{D}, \boldsymbol{\theta})$ is uniformly bounded above by a constant $L$ (Wu et al., 2023).

At each iteration, the candidate point $\boldsymbol{\xi}$ is obtained by optimizing $\mathrm{GI}(\boldsymbol{\xi}; \mathcal{D}, \boldsymbol{\theta})$ and subsequently incorporated into the kernel dictionary $\mathcal{D}$. In streaming settings with infinitely arriving data, however, the kernel dictionary would grow unbounded as iterations proceed, which fundamentally limits the applicability of BO in online learning. To overcome this issue, we propose a compression algorithm, i.e., SWC, based on the sliced Wasserstein distance to efficiently compress $\mathcal{D}$ (see Section 3.2 for details). This algorithm guarantees that the size of the kernel dictionary remains bounded independently of $t$, while ensuring that the compressed probability distribution converges to the domain of the true probability distribution.

Using the BO surrogate model, we then obtain the approximate gradient at iteration $t$ as

$$\widehat{\nabla \mathcal{L}}_t = \boldsymbol{\mu}_{\mathcal{D}_{t-1}} = \nabla K(\hat{\boldsymbol{\theta}}_{t-1}, \mathcal{D}_{t-1}) K(\mathcal{D}_{t-1}, \mathcal{D}_{t-1})^{-1} \mathcal{L}(\hat{\boldsymbol{\theta}}_{t-1}, \boldsymbol{z}_t). \tag{4}$$

This formulation enables iterative updates without requiring storage of historical raw data or direct access to the gradient of the objective function. Upon receiving the $t$-th sample $\boldsymbol{z}_t = (\boldsymbol{x}_t^\top, y_t)^\top$, the parameter estimate is updated via

$$\hat{\boldsymbol{\theta}}_t = \hat{\boldsymbol{\theta}}_{t-1} - \eta_t \widehat{\nabla \mathcal{L}}_t,$$

where $\eta_t$ denotes the step size at iteration $t$. Throughout the procedure, only the estimator $\hat{\boldsymbol{\theta}}_{t-1}$ and the kernel dictionary $\mathcal{D}_{t-1}$ are required, thereby ensuring greater flexibility and substantially reduced memory usage.

However, while the above procedure enables efficient online estimation, it does not inherently safeguard sensitive information. In streaming environments, where each newly arriving observation may expose individual data, privacy protection is indispensable. Unlike traditional centralized approaches to DP (Sopa et al., 2025), which inject noise into the entire algorithm in a post-hoc manner, our framework embeds privacy protection directly into each iteration. This design eliminates the reliance on a trusted data curator and achieves LDP by ensuring that data are privatized at the source before any aggregation occurs. To enforce rigorous LDP guarantees, we first clip the approximate gradient to a fixed bound $B > 0$, i.e.,

$$g_{t-1}(\hat{\boldsymbol{\theta}}_{t-1}) = \boldsymbol{\mu}_{\mathcal{D}_{t-1}} \cdot \min\left\{1, \frac{B}{\|\boldsymbol{\mu}_{\mathcal{D}_{t-1}}\|}\right\},$$

and then perturb it with noise drawn from a suitable distribution to ensure privacy. Common choices include Gaussian, Laplace, or more sophisticated mechanisms (Dwork et al., 2014; Dong et al., 2022). In this work, we adopt the Gaussian mechanism primarily for illustrative purposes, owing to its analytical simplicity. Nevertheless, our proposed framework is general and can be easily extended to other noise distributions. Let $\omega_t$ denote Gaussian noise with mean zero and covariance matrix $2(2B/\varepsilon_t)^2 \log(1.25/\delta_t) \mathbf{I}_p$, where $(\varepsilon_t, \delta_t)$ is the privacy budget allocated to the $t$-th iteration. The proposed private estimator is initialized at $\hat{\boldsymbol{\theta}}_0 = \tilde{\boldsymbol{\theta}}_0 = \mathbf{0}_p$ and updated as

$$\hat{\boldsymbol{\theta}}_t = \hat{\boldsymbol{\theta}}_{t-1} - \eta_t \{g_{t-1}(\hat{\boldsymbol{\theta}}_{t-1}) + w_t\}, \quad \tilde{\boldsymbol{\theta}}_t = \{(t-1)\tilde{\boldsymbol{\theta}}_{t-1} + \hat{\boldsymbol{\theta}}_t\}/t. \tag{5}$$

Notably, the optimization of the acquisition function, the SWC compression, and the posterior mean evaluation depend only on the kernel $K$, the compressed dictionary $\mathcal{D}_{t-2}$, and the previous parameter estimate $\hat{\boldsymbol{\theta}}_{t-1}$, making the proposed method well-suited to streaming environments. The proposed LDP-BO procedure is summarized in Algorithm 1.

By the post-processing property A.4 of LDP, we establish the following privacy guarantee for Algorithm 1.

**Theorem 3.1.** *Given an initial estimate $\hat{\boldsymbol{\theta}}_0 \in \mathbb{R}^p$, consider the iterates $\{\hat{\boldsymbol{\theta}}_t\}_{t \geq 1}$ defined in Algorithm 1. Then the final output $\tilde{\boldsymbol{\theta}}_t$ satisfies $(\max\{\varepsilon_1, \ldots, \varepsilon_t\}, \max\{\delta_1, \ldots, \delta_t\})$-LDP.*

Theorem 3.1 guarantees that each update of the proposed LDP-BO algorithm satisfies $(\max\{\varepsilon_1, \ldots, \varepsilon_t\}, \max\{\delta_1, \ldots, \delta_t\})$-LDP by introducing Gaussian noise calibrated to the sensitivity of the gradient. This mechanism safeguards the privacy of every individual sample at each

---

**Algorithm 1** Online Locally Differentially Private Bayesian Optimization Algorithm (LDP-BO).

---

1: **Input**: User-defined loss function $\mathcal{L}(\cdot, \boldsymbol{z})$, a clipping bound $B > 0$, learning rates $\{\eta_t\}_{t \geq 1}$, privacy parameters $\{(\varepsilon_t, \delta_t)\}_{t \geq 1}$, and a compression budget $\kappa > 0$.

2: **Initialize**: Non–data-dependent parameters $\hat{\boldsymbol{\theta}}_0 = \tilde{\boldsymbol{\theta}}_0 = \mathbf{0}_p$, and evaluation set $\mathcal{D}_{-1} = \emptyset$.

3: **for** $t = 1, 2, \ldots$ **do**

4:     Collect a new data point $\boldsymbol{z}_t = (\boldsymbol{x}_t^\top, y_t)^\top$.

5:     Select the candidate point $\boldsymbol{\xi} = \arg\min_{\boldsymbol{\xi}} \mathrm{GI}(\boldsymbol{\xi}; \mathcal{D}_{t-2}, \hat{\boldsymbol{\theta}}_{t-1})$.

6:     Update the compressed dictionary via SWC Algorithm 2 $\mathcal{D}_{t-1} = \mathrm{SWC}(\mathcal{D}_{t-2}, \boldsymbol{\xi})$.

7:     Evaluate the loss function at $\mathcal{L}(\hat{\boldsymbol{\theta}}_{t-1}, \boldsymbol{z}_t)$ at point $\boldsymbol{z}_t$.

8:     Compute the posterior mean $\boldsymbol{\mu}_{\mathcal{D}_{t-1}}$ by (4).

9:     Clip the gradient to obtain $g_{t-1}(\hat{\boldsymbol{\theta}}_{t-1}) = \boldsymbol{\mu}_{\mathcal{D}_{t-1}} \cdot \min\left\{1, \frac{B}{\|\boldsymbol{\mu}_{\mathcal{D}_{t-1}}\|}\right\}$.

10:     Perform the noisy gradient descent step and update $\hat{\boldsymbol{\theta}}_t$ and $\tilde{\boldsymbol{\theta}}_t$ by (5).

11: **end for**

12: **Output**: $\tilde{\boldsymbol{\theta}}_t$.

---

iteration while eliminating the need to store raw data. The analysis for time-varying privacy parameters $(\varepsilon_t, \delta_t)$ proceeds analogously to that of the constant-$(\varepsilon, \delta)$ case. Hence, for clarity of exposition, we focus on a fixed privacy level $(\varepsilon, \delta)$ in the subsequent discussion.

## 3.2 SLICED WASSERSTEIN COMPRESSION

As discussed above, a major challenge in streaming data settings is the unbounded growth of the kernel dictionary as new points are continuously arrived. To address this issue, we develop an SWC strategy that controls the growth of the dictionary while preserving the statistical fidelity of the surrogate model. Specifically, in Algorithm 1, whenever a candidate point $\boldsymbol{\xi}$ is selected by (3), the posterior distribution $\rho_{\tilde{\mathcal{D}}_t}$ is updated according to (2), where $\tilde{\mathcal{D}}_t = \mathcal{D}_{t-1} \cup \boldsymbol{\xi}$. To ensure computational efficiency, the enlarged dictionary $\tilde{\mathcal{D}}_t$ is subsequently compressed using the Sliced Wasserstein (SW) distance, which quantifies discrepancies between probability distributions through their one-dimensional projections (see Bonneel et al. (2015) for details).

Our primary goal is to guarantee that the compressed dictionary $\mathcal{D}_t$ satisfies

$$\mathrm{SW}_2(\rho_{\mathcal{D}_t}, \rho_{\tilde{\mathcal{D}}_t}) < \kappa,$$

for a prescribed budget parameter $\kappa$, where $\rho$ denotes the posterior density. We define the model order $M_t$ as the column dimension of the compressed kernel dictionary $\mathcal{D}_t$. This compression step ensures that $M_t \leq M_{t-1} + 1$, thereby keeping the dictionary size bounded over time. The detailed SWC procedure is provided in Algorithm 2.

---

**Algorithm 2** Sliced Wasserstein Compression (SWC).

---

1: **Input**: Previous dictionary $\mathcal{D}_{t-1}$, new acquisition point $\boldsymbol{\xi}$ and a compression budget $\kappa > 0$.

2: **Initialize**: $\tilde{\mathcal{D}}_t = \mathcal{D}_{t-1} \cup \boldsymbol{\xi}$ and index set $\mathcal{I} = \{1, \ldots, \tilde{M}_t\}$.

3: **while** $\mathcal{I} \neq \emptyset$ **do**

4:     **for** $j \in \mathcal{I}$ **do**

5:         Compute Sliced Wasserstein distance $\eta_j = \mathrm{SW}_2(\rho_{\mathcal{D}_{-j}}, \rho_{\tilde{\mathcal{D}}_t})$.

6:     **end for**

7:     Identify index with minimal distance $j^* = \arg\min_{j \in \mathcal{I}} \eta_j$.

8:     **if** $\eta_{j^*} > \kappa$ **then break**

9:     **else**

10:         $\mathcal{I} = \mathcal{I} \setminus \{j^*\}, \mathcal{D}_t = \tilde{\mathcal{D}}_{\mathcal{I}}$.

11:     **end if**

12: **end while**

13: **Output**: Compressed dictionary $\mathcal{D}_t$ such that $\mathrm{SW}_2(\rho_{\mathcal{D}_t}, \rho_{\tilde{\mathcal{D}}_t}) \leq \kappa$.

---

To ensure that the posterior distribution produced by Algorithm 2 converges to a stationary region, we impose the following assumption.

**Assumption 3.2.** *For any $c > 0$, let $\rho_t$ denote the true posterior density, and define the events:* $\psi_t = \{\mathrm{SW}_2(\rho_t, \rho_{t-1}) < c \mid \mathcal{D}_t\}, \tilde{\psi}_t = \{\mathrm{SW}_2(\rho_{\mathcal{D}_t}, \rho_{\mathcal{D}_{t-1}}) < c \mid \mathcal{D}_t\}$. *We assume that compression does not increase the probability of divergence relative to the original model, i.e.,* $P\{\psi_t\} \geq P\{\tilde{\psi}_t\}$.

Assumption 3.2 is mild, as the likelihood of the true posterior is at least as large as that of the sparse GP, a condition also adopted in Koppel et al. (2021). In our analysis, Assumption 3.2 serves as the Bayesian analogue of the nonexpansiveness property of projection operators. This property is essential for establishing an upper bound on the error introduced by kernel dictionary compression.

**Theorem 3.3.** *For the compression process in Algorithm 2, the model order $M_t$ of each posterior $\rho_{D_t}$ is uniformly bounded as*

$$M_t \leq \mathcal{O}\left(\frac{1}{\kappa}\right)^p \; \text{for all } t.$$

Theorem 3.3 establishes that, in the streaming setting, the kernel dictionary size in our BO framework remains uniformly bounded, with dependence only on the compression budget $\kappa$ and the input dimension $p$. By operating directly on one-dimensional sample projections, the proposed method circumvents explicit density estimation and thereby mitigates sensitivity to both ambient dimensionality and discretization errors (Kolouri et al., 2015).

## 4 THEORETICAL PROPERTIES

In this section, we investigate the finite-sample properties of the proposed estimator. Firstly, we establish theoretical guarantees for the estimator produced by Algorithm 1 under the strongly convex loss. In order to obtain the convergence property, we also need the following assumptions.

**Assumption 4.1.** *There exists a $B < \infty$ such that all $t \geq 1, \boldsymbol{\theta} \in \Theta$, we have $\|\nabla \mathcal{L}(\boldsymbol{\theta}, \boldsymbol{z}_t)\| \leq B$.*

**Assumption 4.2.** *For all $t \geq 1$, $\mathcal{L}(:, \boldsymbol{z}_t) \in \mathcal{H} = RKHS(K)$, where $K$ is the kernel used in Algorithm 1. Moreover, there exists a constant $C_{\mathcal{X}} < \infty$ such that for all $t$, $\|\mathcal{L}(:, \boldsymbol{z}_t)\|_{\mathcal{H}} \leq C_{\mathcal{X}}$*

**Assumption 4.3.** *Assume that the objective function $f(\boldsymbol{\theta})$ is differentiable, $\zeta$-smoothness, and $\lambda$-strongly convex, in the sense*

$$(i) \quad f(\boldsymbol{\theta}_1) - f(\boldsymbol{\theta}_2) \leq \langle \nabla f(\boldsymbol{\theta}_2), \boldsymbol{\theta}_1 - \boldsymbol{\theta}_2 \rangle + \frac{\zeta}{2}\|\boldsymbol{\theta}_1 - \boldsymbol{\theta}_2\|^2, \quad \forall \boldsymbol{\theta}_1, \boldsymbol{\theta}_2 \in \Theta \subseteq \mathbb{R}^p,$$

$$(ii) \quad f(\boldsymbol{\theta}_1) - f(\boldsymbol{\theta}_2) \geq \langle \nabla f(\boldsymbol{\theta}_2), \boldsymbol{\theta}_1 - \boldsymbol{\theta}_2 \rangle + \frac{\lambda}{2}\|\boldsymbol{\theta}_1 - \boldsymbol{\theta}_2\|^2, \quad \forall \boldsymbol{\theta}_1, \boldsymbol{\theta}_2 \in \Theta \subseteq \mathbb{R}^p.$$

Assumption 4.1 ensures that the sensitivity of the gradient is uniformly bounded, a condition frequently imposed in LDP optimization to control the amount of noise required for privacy see, e.g., Song et al. (2013); Avella-Medina et al. (2023). In practice, this condition can be achieved using Mallow weights (Avella-Medina et al., 2023). Assumption 4.2 requires the target function to lie within the kernel-induced space, a condition that is commonly assumed in the literature on theoretical analyses of Bayesian optimization, enabling convergence and estimation bounds under standard regularity conditions (Wu et al., 2023; Sopa et al., 2025). Assumption 4.3 imposes strong convexity and smoothness on the loss function, which are standard conditions for the convergence analysis of (stochastic) gradient optimization methods. Similar conditions can be found in Vaswani et al. (2022); Zhu et al. (2023).

Recall that $\hat{\boldsymbol{\theta}}_t$ is the estimate obtained at the $t$-th iteration of the proposed LDP-BP Algorithm 1 under $(\varepsilon, \delta)$-LDP, while $\boldsymbol{\theta}^\star$ denotes the true parameter value. The theorem below provides a non-asymptotic bound on the mean squared error of the estimate at iteration $t$.

**Theorem 4.4 ($(\varepsilon, \delta)$-LDP).** *Under Assumptions 4.1-4.3, there exist some positive constants $a_p$ and $c_p$ that depends on the dimension $p$ and define $t_0 = \min\{t : \lambda \geq 2a_p^2\eta_t, \lambda\eta_t t \geq 8\alpha \log t\}$, such that for $t \geq t_0$, $\hat{\boldsymbol{\Delta}}_t = \hat{\boldsymbol{\theta}}_t - \boldsymbol{\theta}^\star$ satisfies*

$$E(\|\hat{\boldsymbol{\Delta}}_t\|_2^2) \lesssim t^{-\alpha}\{(\eta c_p B^2 \log(1.25/\delta)/(\lambda\varepsilon^2) + \eta(L + p\kappa + 2B^2)/\lambda + \|\hat{\boldsymbol{\Delta}}_0\|_2^2\},$$

*when the step-size is chosen to be $\eta_t = \eta t^{-\alpha}$ with $\eta > 0$ and $1/2 < \alpha < 1$.*

Theorem 4.4 establishes that the mean squared error $E(\|\hat{\boldsymbol{\Delta}}_t\|_2^2)$ converges at rate $\mathcal{O}(t^{-\alpha})$ under the step size $\eta_t = \eta t^{-\alpha}$. The bound consists of three components: the privacy-induced noise term $B^2 \log(1.25/\delta)/(\lambda \varepsilon^2)$, the compression error $L + p\kappa$, and the error from the initial estimate. Notably, $L$ can be made arbitrarily small by minimizing the acquisition function over $p+1$ points (Wu et al., 2023). Furthermore, as the compression budget $\kappa \to 0$, the rate coincides with that of Xie et al. (2025). Unlike their result, which requires a restrictive assumption on the conditional covariance of gradient noise, our analysis avoids this condition, thereby providing broader applicability.

Although standard in stochastic approximation Chen et al. (2020); Sherman et al. (2021); Kovalev & Gasnikov (2022), global strong convexity is unrealistic for BO, which often involves multimodal objectives. Importantly, our theory is not confined to this setting. We have introduced significantly weaker conditions (Assumptions B.3-B.5), requiring only smoothness, local strong convexity near each global minimum, and a mild gap–distance condition. Under these assumptions, Corollary B.6 shows that the estimator $\hat{\boldsymbol{\theta}}_t$ converges to the set of global minimizers $\boldsymbol{\Theta}^{\mathrm{opt}}$ at the same rate $O(t^{-\alpha})$, as in the strongly convex case.

Although non-convexity rules out guarantees of global optimality, our analysis relies only on the weaker assumption of $\zeta$-smoothness, under which we establish convergence to an approximate stationary point. In non-convex settings with multiple local minima, convergence is typically analyzed through gradient norms rather than parameter estimates (Garrigos & Gower, 2023).

**Theorem 4.5.** *Under Assumption 4.1, 4.2 and 4.3 (i), there exist some positive constants $c'$, when the step-size is chosen to be $\eta_t = \eta t^{-\alpha}$ with $\eta > 0$ and $1/2 < \alpha < 1$, it follows that for every $t \geq 1$*

$$\min_{1 \leq i \leq t} E\|\nabla f(\hat{\boldsymbol{\theta}}_i)\|^2 \leq c' \frac{(f(\hat{\boldsymbol{\theta}}_0) - f(\boldsymbol{\theta}^\star)) + \zeta(L + p\kappa + B^2) + pB^2/\varepsilon^2 \log(1.25/\delta)}{t^{1-\alpha}}.$$

Theorem 4.5 establishes an $O(t^{-(1-\alpha)})$ convergence rate of the gradient norm under a step size $\eta_t = \eta t^{-\alpha}$ in $\zeta$-smooth optimization without assuming strong convexity. With a fixed step size and no privacy, the rate reduces to the classical $O(t^{-1/2})$ result (Fang et al., 2023; Bu et al., 2023; Wu et al., 2023). The weaker $\zeta$-smoothness assumption still enables meaningful gradient-based analysis, and by controlling the BO approximation error, our method achieves rates comparable to classical non-convex optimization (Garrigos & Gower, 2023). Notably, our guarantees avoid restrictive conditions such as fixing the Lipschitz constant to a specific value (e.g., 1), as required in prior work (Béthune et al., 2023).

In contrast to Theorem 4.4, which relies on strong convexity to establish a convergence rate for parameter estimation, the lack of convexity precludes direct control over the parameter error, thereby presenting a fundamental challenge. To address this, Theorem 4.5 leverages recursive moment bounds on the gradients and averaging techniques, yielding a convergence rate in gradient norm and guaranteeing convergence to an approximate stationary point. These findings align with existing literature (Stich, 2019; Garrigos & Gower, 2023): strong convexity enables rapid parameter recovery, whereas the general analysis guarantees convergence to stationarity in non-convex settings.

## 5 EXPERIMENTS

We assess the finite-sample performance of our method on two synthetic datasets and one real-world dataset, comparing it with LDP-SGD (Xie et al., 2025) in the parametric case and with a non-private deep neural network (Schmidhuber, 2015) in the nonparametric case. We compare the estimates of the coefficients based on 100 simulation replications. Details about the data generating process can be found in Appendix D. It is important to highlight that traditional Bayesian optimization (BO) methods are not suitable for streaming data and, as such, can only be effectively compared on small-scale datasets. We discuss this issue in detail in Appendix E.1.

**Example 5.1** (Parametric Models)**.** We evaluate the proposed LDP-BO algorithm on synthetic data under three regression settings: linear, logistic, and ReLU. We generate $T = 20{,}000$ i.i.d. samples with features $\boldsymbol{x}_t \sim N(0, \mathbf{I}_p)$ and true parameters $\boldsymbol{\theta} = \mathbf{1}_p$, considering dimensions $p \in 2, 5$. The compressed budget is set to $\kappa \in 0.1, 0.2$, and the privacy budget is either fixed at $\varepsilon \in 1, 2$ or randomly drawn from $U(1,2)$ per iteration, with $\delta = 0.2$. For comparison, we include LDP-SGD (Xie et al., 2025), as well as non-private BO and SGD as benchmarks.

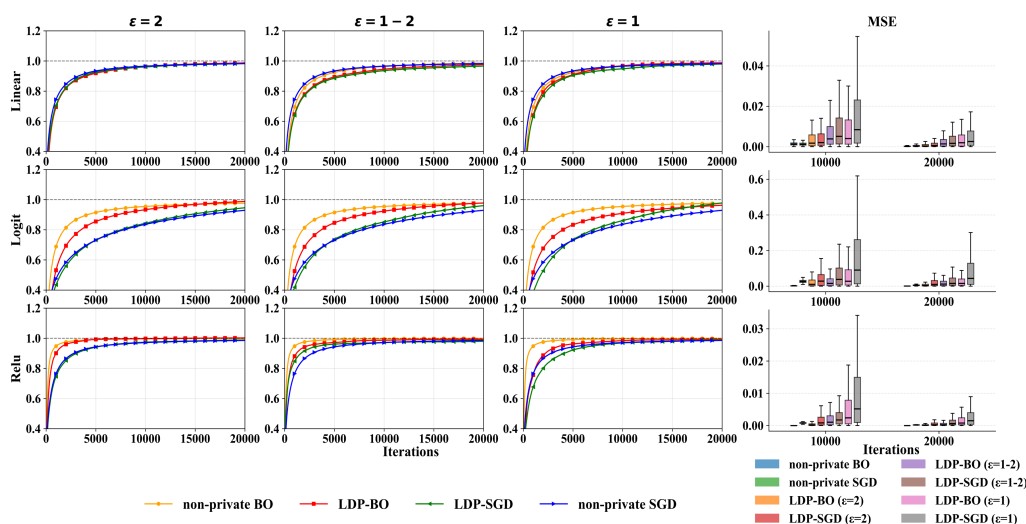

Figure 2: Evolution of the first-dimension coefficient estimate (true value $= 1$) and MSEs over iterations for linear, logistic, and ReLU models (rows) in Example 5.1. Columns correspond to privacy budgets $\varepsilon = 2$, $\varepsilon \sim U(1, 2)$, and $\varepsilon = 1$, and Boxplots of coefficient MSEs.

The first three columns of Figure 2 shows the evolution of the average of the first-dimension coefficient estimate (true value $= 1$) over iterations. For simple models (linear), LDP-BO and LDP-SGD closely track their non-private counterparts, while in more complex models (logistic, ReLU), BO-based methods outperform SGD across all privacy levels. The last column of Figure 2 reports Mean-Squared Errors (MSEs) of the estimates, calculated as $\mathrm{MSE} = \sum_{j=1}^{p} \mathrm{MSE}_j/p = \sum_{j=1}^{p} \sum_{i=1}^{t} (\hat{\theta}_{i,j} - \theta_j)^2/(tp)$, where LDP-BO consistently achieves lower error and variability than LDP-SGD, especially in complex settings. Under strong privacy ($\varepsilon = 1$), LDP-BO converges faster and attains accuracy comparable to non-private BO and SGD. These results highlight LDP-BO's modeling advantage in nonlinear problems, mitigating the utility loss common in gradient-based methods. Results for $p = 5, 20$ and varying compression budgets, reported in Appendix D, are similar.

**Example 5.2** (Nonparametric Models)**.** In this example, we evaluate our approach under nonparametric settings using the Sine and Friedman functions. A Gaussian process regression model is employed to estimate the unknown function, with kernel parameters optimized via our proposed LDP-BO framework (see Appendix D for details). We compare its utility against a non-private deep neural network (denoted as DNN) (Schmidhuber, 2015) trained incrementally with one data point per iteration.

We generate $T = 10{,}000$ i.i.d. samples with features $\boldsymbol{x}_t \sim U(-1, 1)$. For the Sine function, $y_t = \sin(2\pi x_t) + \varepsilon_t$; for the Friedman function, $y_t = \sin(\pi x_{1t} x_{2t}) + (x_{3t} - 0.5)^2 + x_{4t} + x_{5t} + \varepsilon_t$, where $\varepsilon_t \sim \mathcal{N}(0, 0.1^2)$. We set the compression budget to $\kappa = 0.1$, the privacy budget to $(\varepsilon, \delta) = (1, 0.2)$ and $B = 2$. We report the MSE of averaged estimators at sample sizes $n = 2000, 5000$, and $10000$, and provide function fitting plots at $n = 10000$ using 100 randomly generated test points.

Figure 3 presents the prediction errors (calculated as $\mathrm{error}_t = \frac{t-1}{t}\mathrm{error}_{t-1} + \frac{1}{t}(y_t - \hat{y}_t)^2$ in the online setting) and function fitting results for the proposed LDP-BO method and the DNN baseline. The LDP-BO method consistently outperforms the non-private DNN, even under privacy constraints. The boxplots show that LDP-BO achieves lower variance and fewer outliers, indicating greater stability and robustness across trials. The fitted curves further demonstrate that LDP-BO closely tracks the true function, capturing both global trends and fine-scale structure—particularly in high-value regions critical for optimization. In contrast, the DNN exhibits larger deviations and unstable oscillations, reflecting weaker generalization and poorly calibrated uncertainty.

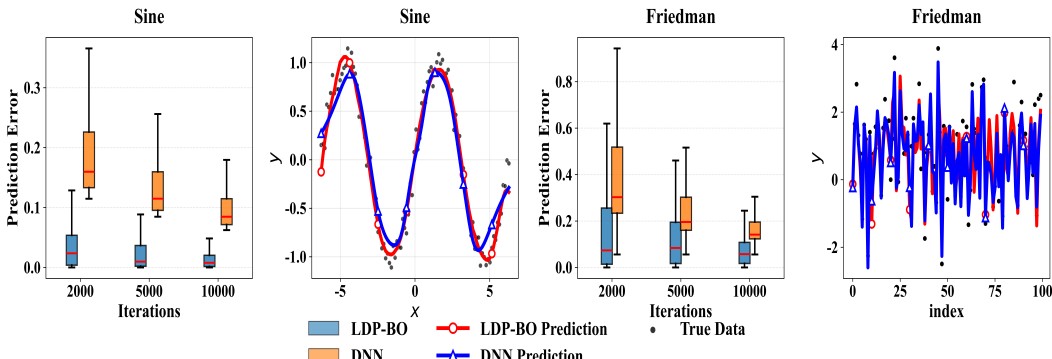

Figure 3: Predcition errors and function fitting plots of the proposed LDP-BO and DNN methods in Example 5.2.

**Example 5.3** (Real Data Analysis). In this example, we apply LDP-SGD to real Uber Fares Dataset [1] and Credit Card Fraud Detection Dataset[2]. Uber Fares Dataset comprises approximately 21,000 historical trip records collected between 2014 and 2015 in New York City. The selected features include distance, hour of day, day of week and passenger count; see Appendix D for full preprocessing details. These predictors, which collectively capture spatial, temporal, and demand-related determinants of Uber fare variations, have been similarly employed in prior studies (Khandelwal et al., 2021; Silveira-Santos et al., 2023; Huynh et al., 2025). The response is chosen to be the fare.

Credit Card Fraud Detection Dataset comprises approximately 20,000 transaction records made in September 2013. The dataset consists of transaction records where each transaction is represented by PCA-transformed features. The top 5 principal components are selected to capture the most significant variations in the data, which is a common practice in fraud detection studies (Bestami Yuksel et al., 2020; Ogundile et al., 2024). The target variable is binary, indicating whether the transaction is fraudulent or legitimate. Since the data is already in its principal component form, no further preprocessing is required.

The Table 2 compares the performance of LDP-BO, DP-BO, and LDP-SGD under $(\varepsilon, \delta) = (1, 0.2)$ on the Uber and Credit datasets at different sample sizes of 2000, 5000, 10000, and 20000. The results show that LDP-BO consistently outperforms LDP-SGD across all metrics, achieving lower prediction error for Uber and higher accuracy for Credit. While the offline method (Sopa et al., 2025), is only applied to the first 2000 samples due to its computational limitations, LDP-BO demonstrates similar performance in smaller sample sizes. As the sample size increases, LDP-BO continues to exhibit improved accuracy and stability, whereas offline methods face significant challenges and cannot scale to larger datasets. This trend highlights the reduced estimation variance and enhanced stability of LDP-BO, even under strict privacy constraints.

Table 2: Performance on Uber (average prediction error) and Credit (accuracy) for different methods at various sample sizes.

| Sample Size | Uber | | | Credit | | |
|---|---|---|---|---|---|---|
| | LDP-BO | DP-BO | LDP-SGD | LDP-BO | DP-BO | LDP-SGD |
| 2000 | 5.471 | 5.129 | 17.412 | 0.941 | 0.944 | 0.913 |
| 5000 | 2.224 | * | 10.271 | 0.944 | * | 0.929 |
| 10000 | 1.409 | * | 3.252 | 0.951 | * | 0.940 |
| 20000 | 0.782 | * | 1.794 | 0.969 | * | 0.952 |

---

[1] https://www.kaggle.com/datasets/yasserh/uber-fares-dataset
[2] https://www.kaggle.com/mlg-ulb/creditcardfraud

AUTHOR CONTRIBUTIONS

If you'd like to, you may include a section for author contributions as is done in many journals. This is optional and at the discretion of the authors.

ACKNOWLEDGMENTS

Use unnumbered third level headings for the acknowledgments. All acknowledgments, including those to funding agencies, go at the end of the paper.

ETHICS STATEMENT

Our research strictly adheres to the ICLR Code of Ethics requirements in all aspects.

REPRODUCIBILITY STATEMENT

Algorithms 1-2, Section 5 and Appendix D have provided detailed information to ensure the reproduction of core results. We provide open access to the code with sufficient instructions, as described in supplemental material. We set $\eta_t = \eta_0 t^{-\alpha}$ with $\eta_0 = 0.2, \alpha = 0.505$, and the random seed to 1. The kernel choice is specified in Section D.

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

# A BACKGROUND ON LDP AND SLICED WASSERSTEIN DISTANCE

## A.1 DIFFERENTIAL PRIVACY

In this section, we begin with the basic concepts and properties of Local Differential Privacy (LDP), Rényi Differential Privacy (RDP) and Gaussian Differential Privacy (GDP). The intuition underlying LDP is that a randomized algorithm produces outputs that are statistically similar, even when a single individual's information in the dataset is modified or removed, thereby ensuring the protection of individual privacy. The formal definition of LDP is presented below.

**Definition A.1.** $((\varepsilon, \delta)$-LDP (Xiong et al., 2020)) Let $\mathcal{X}$ be the sample space for an individual data, a randomized algorithm $\mathcal{A} : \mathcal{X} \to \mathbb{R}$ is $(\varepsilon, \delta)$-LDP if and only if for any pair of input single values $\boldsymbol{z}, \boldsymbol{z}' \in \mathcal{X}$ and for any $S \subseteq \mathbb{R}$, the inequality below holds

$$P\left(\mathcal{A}(\boldsymbol{z}) \in S\right) \leq e^{\varepsilon} \cdot P\left(\mathcal{A}(\boldsymbol{z}') \in S\right) + \delta.$$

In contrast to CDP, LDP imposes a stricter requirement in which each individual perturbs their data locally before submission. This design eliminates the need for a trusted data curator and is particularly well suited to streaming environments, where data are continuously generated and transmitted. To formalize the guarantee, we introduce the notion of sensitivity, which quantifies the maximum change in an algorithm's output resulting from the modification of a single data entry.

**Definition A.2.** For any deterministic function $g : \mathcal{X} \to \mathbb{R}$ and any pair of input single values $\boldsymbol{z}, \boldsymbol{z}' \in \mathcal{X}$, the $\ell_p$-sensitivity of $g$ is defined as

$$\Delta_p(g) = \sup_{\boldsymbol{z}, \boldsymbol{z}' \in \mathcal{X}} \|g(\boldsymbol{z}) - g(\boldsymbol{z}')\|_p.$$

Among various LDP mechanisms, we introduce the following Gaussian mechanism for illustrative purposes, as it facilitates clear exposition.

**Definition A.3.** (The Gaussian Mechanism (Dwork, 2006)) Let $g : \mathcal{X} \to \mathbb{R}$ be a deterministic function with $\Delta_2(g) < \infty$. For $\boldsymbol{w} \in \mathbb{R}$ with coordinates $w_1, w_2, \cdots, w_p$ be i.i.d samples drawn from $N(0, 2(\Delta_2(g)/\varepsilon)^2 \log(1.25/\delta))$, $g(\boldsymbol{z}) + \boldsymbol{w}$ is $(\varepsilon, \delta)$-LDP.

The post-processing and parallel composition properties are fundamental to LDP, enabling complex algorithms to be systematically constructed from simpler components.

**Proposition A.4.** (Post-processing Property for LDP (Xiong et al., 2020)) Let $\mathcal{A}$ be an $(\varepsilon, \delta)$-LDP algorithm and $g$ be an arbitrary function which takes $\mathcal{A}(\boldsymbol{z})$ as input, then $g(\mathcal{A}(\boldsymbol{z}))$ is also $(\varepsilon, \delta)$-LDP.

**Proposition A.5.** (Parallel Composition for LDP (Xiong et al., 2020)) Suppose $n$ mechanisms $\{\mathcal{A}_1, \ldots, \mathcal{A}_n\}$ satisfy $(\varepsilon_i, \delta_i)$-LDP, respectively, and are computed on disjoint subsets of data, then a mechanism formed by $(\mathcal{A}_1(\boldsymbol{z}_1), \ldots, \mathcal{A}_n(\boldsymbol{z}_n))$ satisfies $(\max(\varepsilon_i), \max(\delta_i))$-LDP.

As an alternative to standard LDP, RDP was introduced by Mironov (2017) as a generalization of LDP based on Rényi divergence, providing a more structured and flexible framework for privacy accounting. RDP quantifies privacy loss through the Rényi divergence of order $q > 1$ between the output distributions of an algorithm on adjacent datasets. For two probability distributions $P$ and $Q$, the Rényi divergence of order $q$ is defined as

$$D_q(P\|Q) = \frac{1}{q-1} \log E_Q \left\{ \left(\frac{P}{Q}\right)^{q-1} \right\},$$

whenever the expectation exists. This divergence provides a smooth and fine-grained measure of dissimilarity that depends on the order $q$, thereby enabling more precise tracking of cumulative privacy loss under composition compared to the standard $(\varepsilon, \delta)$-LDP framework. Formally, RDP is defined as follows:

**Definition A.6.** (RDP, Mironov (2017)). Let $\mathcal{A}$ be a randomized algorithm, and let $\boldsymbol{z}$ and $\boldsymbol{z}'$ be two adjacent datasets. For any real number $\alpha > 1$, the algorithm $\mathcal{A}$ satisfies $(q, \varepsilon)$-RDP if

$$D_q\big(\mathcal{A}(\boldsymbol{z}) \,\|\, \mathcal{A}(\boldsymbol{z}')\big) \leq \varepsilon,$$

where $\mathcal{A}(\boldsymbol{z})$ denotes the distribution of the output of $\mathcal{A}$ on data $\boldsymbol{z}$.

Building on this hypothesis testing framework, Dong et al. (2022) introduced GDP, a privacy notion with a natural statistical interpretation: determining whether an individual's data is included in a dataset is at least as difficult as distinguishing between $N(0,1)$ and $N(\mu,1)$ based on a single observation, for some $\mu > 0$. Formally, GDP is defined as follows:

**Definition A.7.** *(GDP, Dong et al. (2022) Let $\mathcal{A}$ be a randomized algorithm.*

*1. $\mathcal{A}$ satisfies $f$-DP if, for any $\alpha$-level test of $H_0$, the power function $\beta(\alpha)$ satisfies $\beta(\alpha) \leq 1 - f(\alpha)$, where $f$ is convex, continuous, non-increasing, and $f(\alpha) \leq 1 - \alpha$ for all $\alpha \in [0,1]$.*

*2. $\mathcal{A}$ satisfies $\mu$-GDP if it is $f$-DP with $f(\alpha) \geq \Phi(\Phi^{-1}(1-\alpha) - \mu)$ for all $\alpha \in [0,1]$, where $\Phi(\cdot)$ denotes the standard normal CDF.*

## A.2 SLICED WASSERSTEIN DISTANCE

**Definition A.8.** *(Wasserstein Distance (Villani, 2021)) The Wasserstein distance $W_p(u,\nu)$ quantifies the optimal transport cost between two probability distributions $u$ and $\nu$, defined as the minimal expected cost required to redistribute mass from $u$ to $\nu$. For univariate distributions, it admits the closed-form*

$$W_p(u,\nu) = \left( \int_{\mathcal{X}} \left| x - F_\nu^{-1}(F_u(x)) \right|^p du(x) \right)^{1/p} = \left( \int_0^1 \left| F_u^{-1}(t) - F_\nu^{-1}(t) \right|^p dt \right)^{1/p},$$

*where $F(\cdot)$ denotes the cumulative distribution function (CDF). In particular, if $u = N(m_1, \sigma_1^2)$ and $\nu = N(m_2, \sigma_2^2)$, are univariate Gaussian distributions, their 2-Wasserstein distance admits the analytic form $W_2(u,\nu) = \sqrt{(m_1 - m_2)^2 + (\sigma_1 - \sigma_2)^2}$.*

**Definition A.9.** *(Sliced Wasserstein (SW) Distance (Bonneel et al., 2015)) The Sliced Wasserstein distance generalizes the Wasserstein distance to higher dimensions via Radon transforms. Specifically, it projects multivariate distributions onto one-dimensional subspaces determined by directions $\boldsymbol{\theta} \in \mathbb{S}^{p-1}$, computes the Wasserstein distance between these projections, and then averages across directions:*

$$SW_p(u,\nu) = \left( \int_{\boldsymbol{\theta} \in \mathbb{S}^{p-1}} W_p^p(\mathcal{R}u_{\boldsymbol{\theta}}, \mathcal{R}\nu_{\boldsymbol{\theta}}) \, d\boldsymbol{\theta} \right)^{1/p}.$$

In practice, the SW distance is typically approximated using Monte Carlo sampling over $m$ random directions: $SW_p(u,\nu) \approx \{ \sum_{l=1}^m W_p^p(\mathcal{R}u_{\boldsymbol{\theta}}, \mathcal{R}\nu_{\boldsymbol{\theta}})/m \}^{1/p}$. For our experiments, we used a value of $m = 100$.

# B ADDITIONAL COROLLARIES

In this section, we additionally present two corollaries that provide non-asymptotic error bounds for the LDP-BO algorithm under specific privacy definitions.

**Corollary B.1** ( $(q,\varepsilon)$**-RDP**). *Suppose the conditions of Theorem 4.4 hold. Under $(q,\varepsilon)$-Rényi Differential Privacy (RDP), where noise $\omega_t = B\sqrt{q/(2\varepsilon)} \cdot N(0, \mathbf{I}_p)$ is added at each iteration in Algorithm 1, the expected estimation error satisfies*

$$E(\|\hat{\boldsymbol{\Delta}}_t\|_2^2) \lesssim t^{-\alpha}\{(\eta c_p B^2 q/(2\lambda\varepsilon) + \eta(L + p\kappa + 2B^2)/\lambda + \|\hat{\boldsymbol{\Delta}}_0\|_2^2\}.$$

**Corollary B.2** ($\mu$**-GDP**). *Suppose the conditions of Theorem 4.4 hold. Under $\mu$-Gaussian Differential Privacy (GDP), where noise $\omega_t = \frac{2B}{\mu} \cdot N(0, \mathbf{I}_p)$ is added at each iteration in Algorithm 1, the expected estimation error satisfies*

$$E(\|\hat{\boldsymbol{\Delta}}_t\|_2^2) \lesssim t^{-\alpha}\{(\eta c_p B^2/(\lambda\mu^2) + \eta(L + p\kappa + 2B^2)/\lambda + \|\hat{\boldsymbol{\Delta}}_0\|_2^2\}.$$

Corollaries B.1–B.2 present the expected estimation error under two specific privacy definitions, $(q,\varepsilon)$-RDP and $\mu$-GDP. The bounds follow the same structure as Theorem 4.5, with identical second and third components, while the first component varies by privacy definition. Specifically, Corollary B.1 shows that $(\alpha,\varepsilon)$-Rényi DP improves the bound from $\mathcal{O}\left(t^{-\alpha} \cdot B^2 \log(1/\delta)/(\lambda\varepsilon^2)\right)$ to $\mathcal{O}\left(t^{-\alpha} \cdot B^2\alpha/(\lambda\varepsilon)\right)$, whereas Corollary B.2 demonstrates that $\mu$-Gaussian DP yields a bound of order $\mathcal{O}\left(t^{-\alpha} \cdot B^2/(\lambda\mu^2)\right)$.

Furthermore, Theorem 4.5 is stated under the global strong convexity Assumption 4.3. The same convergence rate, however, can be established under a local strong convexity condition in a neighborhood of the optimum, using standard localization arguments. Hence, in what follows we replace global strong convexity with the following weaker local curvature assumption. In this setting, the optimal point need not be unique; we denote the set of optimal points by $\Theta^{opt}$. We begin by stating the conditions required for our analysis.

**Assumption B.3.** *There exists positive constants $C_s$ and $C_{hl}$ such that for any $\boldsymbol{\theta}_1, \boldsymbol{\theta}_2 \in \Theta$,*

$$\|\nabla f(\boldsymbol{\theta}_1) - \nabla f(\boldsymbol{\theta}_2)\| \leq C_s \|\boldsymbol{\theta}_1 - \boldsymbol{\theta}_2\|,$$
$$\|\nabla^2 f(\boldsymbol{\theta}_1) - \nabla^2 f(\boldsymbol{\theta}_2)\| \leq C_{hl} \|\boldsymbol{\theta}_1 - \boldsymbol{\theta}_2\|.$$

*There exists $\tilde{\lambda}_{min} > 0$ such that for any $\boldsymbol{\theta}^{opt} \in \Theta^{opt}$, $\lambda_{min}(\nabla^2 f(\Theta^{opt})) \geq \tilde{\lambda}_{min}$.*

Smoothness assumptions on the gradient and Hessian are standard in the optimization literature; see, e.g., Jin et al. (2021); Vlaski & Sayed (2021). The local strong convexity condition ensures that every local minimum is a strong attractor. In particular, by the second part of B.3 there exists a constant $r_{good}^L > 0$ such that for any $\boldsymbol{\theta}^{opt} \in \Theta^{opt}$,

$$\lambda_{\min}\big(\nabla^2 f(\boldsymbol{\theta})\big) \; \geq \; \tfrac{\tilde{\lambda}_{\min}}{2}, \quad \forall \|\boldsymbol{\theta} - \boldsymbol{\theta}^{opt}\| \leq r_{good}^L.$$

Moreover, we assume that optimal points are separated at this scale, i.e., $\|\boldsymbol{\theta} - \boldsymbol{\theta}'\| > r_{good}^L$ for any $\boldsymbol{\theta}, \boldsymbol{\theta}' \in \Theta^{opt}$.

**Assumption B.4.** *$\Theta^{opt}$ is a countable set. There exists a positive constant $C_{tf}$ and a positive integer $\beta_{tf}$ such that for any $\boldsymbol{\theta} \in \Theta, \boldsymbol{\theta}^{opt} \in \Theta^{opt}$,*

$$\|\boldsymbol{\theta} - \boldsymbol{\theta}^{opt}\|^2 \leq C_{tf}\big(1 + \big(f(\boldsymbol{\theta}) - f_{\min}\big)^{\beta_{tf}}\big).$$

**Assumption B.5.** *We define $r_{good} \triangleq \frac{r_{good}^L}{9}$,*

$$R_{good}(\boldsymbol{\theta}^{opt}) \triangleq \{\boldsymbol{\theta} : \|\boldsymbol{\theta} - \boldsymbol{\theta}^{opt}\| \leq r_{good}\}, R_{good}^L(\boldsymbol{\theta}^{opt}) \triangleq \{\boldsymbol{\theta} : \|\boldsymbol{\theta} - \boldsymbol{\theta}^{opt}\| \leq r_{good}^L\}.$$

*We let $R_{good} \triangleq \bigcup_{\boldsymbol{\theta}^{opt} \in \Theta^{opt}} R_{good}(\boldsymbol{\theta}^{opt})$. There exist positive constant $b_0$ and $\tilde{\lambda}$ such that for any $\boldsymbol{\theta} \in \Theta$, if $\|\nabla f(\boldsymbol{\theta})\| \leq b_0$ and $\lambda_{\min}(\nabla^2 f(\boldsymbol{\theta})) > -\tilde{\lambda}$, then $\boldsymbol{\theta} \in R_{good}$.*

Assumption B.4 allows us to use the objective function value to bound the error. Intuitively, it ensures that the objective function landscape resembles a basin, preventing significant deviations in the path (Zhong et al., 2023). Under Assumption B.5, we ensure that all saddle points are escapable, which holds if all saddle points are strict and finite, as is often the case in practice Ge et al. (2015); Sun et al. (2015).

**Corollary B.6.** *Suppose that Assumptions 4.1-4.2 and Assumptions B.3-B.5 hold. The step size parameter $\alpha$ satisfies that $\frac{1}{2} < \alpha < 1$. Then for any $\boldsymbol{\theta}^{opt} \in \Theta^{opt}$, we have*

$$\big(\|\hat{\boldsymbol{\theta}}_t - \boldsymbol{\theta}^{opt}\|^2 \mathbf{1}\big\{\lim_{k \to \infty} \hat{\boldsymbol{\theta}}_k = \boldsymbol{\theta}^{opt}\big\}\big) = O(t^{-\alpha}).$$

Corollary B.6 establishes that, under non-convexity, the convergence rate of the estimated parameters $\hat{\boldsymbol{\theta}}_t$ to the optimal solution $\boldsymbol{\theta}^{opt}$ follows the same rate as in Theorem 4.4 for global strong convexity, i.e., $O(t^{-\alpha})$. This result holds for any $\boldsymbol{\theta}^{opt} \in \Theta^{opt}$, the set of optimal solutions, and demonstrates that local strong convexity is sufficient to guarantee the same convergence rate typically associated with global strong convexity. However, due to the shift from global to local strong convexity, there is no longer a unique global optimum; instead, the set $\Theta^{opt}$ may contain multiple optimal solutions (Zhong et al., 2023). Despite this, the algorithm still converges to a solution within this set at the same rate, showing that the convergence behavior is maintained.Corollary B.6 establishes that, under non-convexity, the convergence rate of the estimated parameters $\hat{\boldsymbol{\theta}}_t$ to the optimal solution $\boldsymbol{\theta}^{opt}$ follows the same rate as in Theorem 4.4 for global strong convexity, i.e., $O(t^{-\alpha})$. This result holds for any $\boldsymbol{\theta}^{opt} \in \Theta^{opt}$, the set of optimal solutions, and demonstrates that local strong convexity is sufficient to guarantee the same convergence rate typically associated with global strong convexity. However, due to the shift from global strong convexity to nonconvexity, there is no longer a unique global optimum; instead, the set $\Theta^{opt}$ may contain multiple optimal solutions (Zhong et al., 2023). Despite this, the algorithm still converges to a solution within this set at the same rate, showing that the convergence behavior is maintained.

# C    SUPPORTING LEMMAS

**Lemma C.1.** *Let $\rho_0$ be the corresponding true population posterior distribution. Suppose the following conditions hold:*

*i. For any measurable subset $\mathcal{A} \subseteq [0,1]^p$ with Lebesgue measure $\lambda(\mathcal{A}) \geq (K_p t)^{-1}$, where $K_p \in (0,1]$ is a constant, $\mathcal{A}$ contains at least one sample point $\boldsymbol{\theta}_t$.*

*ii. For all $t \geq 1$, the kernel matrix is positive definite $K_t \succ 0$.*

*iii. The covariance kernel is translation-invariant, taking the form $K(\boldsymbol{\theta}, \boldsymbol{\theta}') = K(\beta\|\boldsymbol{\theta} - \boldsymbol{\theta}'\|)$ for some scale parameter $\beta > 0$.*

*iv. There exist constants $\delta \in (0, 1/2)$ and $b_1, b_2 > 0$ such that for all $t \geq 1$, $P_{\Pi}\left(\beta > t^{\delta}\right) < b_1 e^{-b_2 t}$, where $P_{\Pi}$ denotes the probability under the Gaussian prior $\Pi$ for $\beta$.*

*Then, the posterior distribution without compression $\rho_t$ is asymptotically consistent, i.e. for every $c > 0$,*

$$P\left(\mathrm{SW}_2(\rho_t, \rho_0) < c \mid \mathcal{D}_t\right) \to 1 \quad (a.s.).$$

*Proof of Lemma C.1.* The results of this lemma are well established, with detailed proofs provided in Theorem 6 of Choi & Schervish (2007). □

**Lemma C.2.** *Assuming the regularity conditions specified in Lemma C.1, which guarantee the well-behaved geometry of the target distribution, Algorithm 2 achieves $\kappa$-approximate convergence under the SW metric. Specifically, for any $c > 0$*

$$\lim_{t \to \infty} P\left\{\mathrm{SW}_2(\rho_{\mathcal{D}_t}, \rho_{\mathcal{D}_{t-1}}) < c + \kappa \mid \mathcal{D}_t\right\} = 1.$$

*Proof of Lemma C.2.* Using triangle inequality, we obtain

$$\mathrm{SW}_2(\rho_{\mathcal{D}_t}, \rho_{\mathcal{D}_{t-1}}) \leq \mathrm{SW}_2(\rho_{\mathcal{D}_t}, \rho_{\tilde{\mathcal{D}}_t}) + \mathrm{SW}_2(\rho_{\tilde{\mathcal{D}}_t}, \rho_{\mathcal{D}_{t-1}}),$$

The first term corresponds exactly to the stopping criterion in Algorithm 2, and is therefore bounded above by $\kappa$. Consequently, following the argument of Koppel et al. (2021), we have the following containment relationship for any $c' > 0$:

$$\{\mathrm{SW}_2(\rho_{\mathcal{D}_t}, \rho_{\mathcal{D}_{t-1}}) < c'\} \subset \{\mathrm{SW}_2(\rho_{\mathcal{D}_t}, \rho_{\tilde{\mathcal{D}}_t}) + \mathrm{SW}_2(\rho_{\tilde{\mathcal{D}}_t}, \rho_{\mathcal{D}_{t-1}}) < c'\}$$
$$\subset \{\mathrm{SW}_2(\rho_{\tilde{\mathcal{D}}_t}, \rho_{\mathcal{D}_{t-1}}) + \kappa < c'\}.$$

Taking prior probability with respect to $\Pi$, it follows that

$$P_{\Pi}\{\mathrm{SW}_2(\rho_{\mathcal{D}_t}, \rho_{\mathcal{D}_{t-1}}) < c'\} \leq P_{\Pi}\{\mathrm{SW}_2(\rho_{\mathcal{D}_t}, \rho_{\tilde{\mathcal{D}}_t}) + \mathrm{SW}_2(\rho_{\tilde{\mathcal{D}}_t}, \rho_{\mathcal{D}_{t-1}}) < c'\}$$
$$\leq P_{\Pi}\{\mathrm{SW}_2(\rho_{\tilde{\mathcal{D}}_t}, \rho_{\mathcal{D}_{t-1}}) + \kappa < c'\}$$
$$\leq P_{\Pi}\{\mathrm{SW}_2(\rho_{\tilde{\mathcal{D}}_t}, \rho_{\mathcal{D}_{t-1}}) < c' - \kappa\}$$

By Assumption 3.2, which states that $P_{\Pi}\{\psi_t\} \geq P_{\Pi}\{\tilde{\psi}_t\}$, we have

$$P_{\Pi}\{\mathrm{SW}_2(\rho_{\tilde{\mathcal{D}}_t}, \rho_{\mathcal{D}_{t-1}}) < c' - \kappa\} \leq P_{\Pi}\{\mathrm{SW}_2(\rho_t, \rho_{t-1}) < c' - \kappa\}$$

By Lemma C.1 the supremum of the probability of the right-hand side of tends 1 as $t \to \infty$ for $c = c' - \kappa > 0$. Therefore

$$\limsup_{t \to \infty} P_{\Pi}\{\mathrm{SW}_2(\rho_{\mathcal{D}_t}, \rho_{\mathcal{D}_{t-1}}) < c'\} = 1.$$

Exploiting the continuity of both the GP posterior and the SW metric, we conclude that the above limit exists. Substituting $c' = c + \kappa$, Lemma C.2 follows. □

**Lemma C.3.** *For a vector $v \in \mathbb{R}^p$, define the projection operator $\Pi_B(v) = v \cdot \min\{1, \frac{B}{\|v\|}\}$, which projects $v$ onto the Euclidean ball $B_B(0)$ of radius $B$ centered at the origin. Under Assumption 4.1, we have, $\forall t \geq 1$,*

$$\|\Pi_B\left(\boldsymbol{\mu}_{\mathcal{D}_t}\right) - \nabla\mathcal{L}(\boldsymbol{\theta}_t, z_t)\| \leq \|\boldsymbol{\mu}_{\mathcal{D}_t} - \nabla\mathcal{L}(\boldsymbol{\theta}_t, z_t)\|.$$

*Proof of Lemma C.3.* Notice that $\Pi_B(x) = \arg\min_{x' \in B_B(0)} \|x - x'\|$, that is, $\Pi_B(x)$ is the Euclidean projection of $x$ onto the ball $B_B(0)$. Now, let $y \in B_B(0)$. Since $B_B(0)$ is convex, for any $0 < \eta < 1$, the convex combination $z := \eta y + (1 - \eta)\Pi_B(x) = \Pi_B(x) + \eta(y - \Pi_B(x))$, also belongs to $B_B(0)$, i.e., $z \in B_B(0)$.

We then obtain

$$
\begin{aligned}
\|x - \Pi_B(x)\|^2 &\leq \|x - z\|^2 = \|x - \Pi_B(x) - \eta(y - \Pi_B(x))\|^2 \\
&= \|x - \Pi_B(x)\|^2 + \eta^2 \|y - \Pi_B(x)\|^2 - 2\eta\langle x - \Pi_B(x), y - \Pi_B(x)\rangle,
\end{aligned}
\tag{6}
$$

where the inequality follows from the definition of $\Pi_B(x)$ as the closest point in $B_B(0)$ to $x$. Thus, we have

$$
\langle x - \Pi_B(x), \Pi_B(x) - y\rangle + \frac{\eta}{2}\|y - \Pi_B(x)\|^2 \geq 0.
$$

As $0 < \eta < 1$ is arbitrary, we obtain

$$
\langle x - \Pi_B(x), \Pi_B(x) - y\rangle = \lim_{\eta \to 0^+} \langle x - \Pi_B(x), \Pi_B(x) - y\rangle + \frac{\eta}{2}\|y - \Pi_B(x)\|^2 \geq 0
$$

for all $y \in B_B(0)$. Using inequality (6), we can further derive the following bound:

$$
\begin{aligned}
\|\boldsymbol{\mu}_{\mathcal{D}_t} - \nabla\mathcal{L}(\boldsymbol{\theta}_t, z_t)\|^2 &= \|\boldsymbol{\mu}_{\mathcal{D}_t} - \Pi_B(\boldsymbol{\mu}_{\mathcal{D}_t}) + \Pi_B(\boldsymbol{\mu}_{\mathcal{D}_t}) - \nabla\mathcal{L}(\boldsymbol{\theta}_t, z_t)\|^2 \\
&= \|\boldsymbol{\mu}_{\mathcal{D}_t} - \Pi_B(\boldsymbol{\mu}_{\mathcal{D}_t})\|^2 + \|\Pi_B(\boldsymbol{\mu}_{\mathcal{D}_t}) - \nabla\mathcal{L}(\boldsymbol{\theta}_t, z_t)\|^2 \\
&\quad + 2\langle \boldsymbol{\mu}_{\mathcal{D}_t} - \Pi_B(\boldsymbol{\mu}_{\mathcal{D}_t}), \Pi_B(\boldsymbol{\mu}_{\mathcal{D}_t}) - \nabla\mathcal{L}(\boldsymbol{\theta}_t, z_t)\rangle \\
&\geq \|\Pi_B(\boldsymbol{\mu}_{\mathcal{D}_t}) - \nabla\mathcal{L}(\boldsymbol{\theta}_t, z_t)\|^2,
\end{aligned}
$$

where the final inequality follows from the fact that both the first and last terms on the right-hand side of (6) are nonnegative, since by Assumption 4.1 we have $\nabla\mathcal{L}(\boldsymbol{\theta}_t, z_t) \in B_B(0)$. □

**Lemma C.4.** *Assume Assumption 4.1 and Assumption 4.2 hold. let $\boldsymbol{\theta} \in \Theta$ and let $\mathcal{D}$ denote a set containing points $\boldsymbol{\theta}$. Denote $g(\boldsymbol{\theta}_t) = \Pi_B(\nabla K(\boldsymbol{\theta}_t, \mathcal{D}_t)K(\mathcal{D}_t, \mathcal{D}_t)^{-1}f(\boldsymbol{\theta}_t))$. Then, there exists some constant $c_1 > 0$ such that*

$$
\|\nabla f(\boldsymbol{\theta}_t) - g(\boldsymbol{\theta}_t)\|^2 \leq c_1(L + p\kappa).
$$

*Proof of Lemma C.4.* Combining Assumption 4.2 with Lemma C.3 of Wu et al. (2023), we obtain

$$
\|\nabla f(\boldsymbol{\theta}_t) - g(\boldsymbol{\theta}_t)\|^2 \leq \|\nabla f(\boldsymbol{\theta}_t) - \nabla K(\boldsymbol{\theta}_t, \mathcal{D}_t)K(\mathcal{D}_t, \mathcal{D}_t)^{-1}f(\boldsymbol{\theta}_t, z_t)\|^2 \leq C_{\mathcal{X}}Tr(\nabla^2 K_{D_t}(\boldsymbol{\theta}_t, \boldsymbol{\theta}_t)),
$$

Since $D_t$ is obtained by compressing $\tilde{D}_t = D_{t-1} \cup \boldsymbol{\xi}$, we then have

$$
\text{SW}_2(\rho_{\mathcal{D}_t}, \rho_{\tilde{\mathcal{D}}_t}) \leq \kappa.
$$

Using the expression of the Sliced Wasserstein distance for multivariate normal distributions, it follows that

$$
\begin{aligned}
&\text{SW}_2{}^2(\rho_{\mathcal{D}_t}, \rho_{\tilde{\mathcal{D}}_t}) \\
&= E_{\boldsymbol{\theta}\sim\mathcal{U}(\mathbb{S}^{p-1})}\left[(\boldsymbol{\theta}^\top(\boldsymbol{\mu}_{t+1}|_{\mathcal{D}_t} - \boldsymbol{\mu}_{t+1}|_{\tilde{\mathcal{D}}_t}))^2 + \left(\sqrt{\boldsymbol{\theta}^\top \Sigma_{t+1}|_{\mathcal{D}_t}\boldsymbol{\theta}} - \sqrt{\boldsymbol{\theta}^\top \Sigma_{t+1}|_{\tilde{\mathcal{D}}_t}\boldsymbol{\theta}}\right)^2\right] \\
&\leq \kappa^2.
\end{aligned}
$$

This implies $E_{\boldsymbol{\theta}\sim\mathcal{U}(\mathbb{S}^{p-1})}\{(\sqrt{\boldsymbol{\theta}^\top \Sigma_{t+1}|_{\mathcal{D}_t}\boldsymbol{\theta}} - \sqrt{\boldsymbol{\theta}^\top \Sigma_{t+1}|_{\tilde{\mathcal{D}}_t}\boldsymbol{\theta}})^2\} \leq \kappa^2$. Notice that $\boldsymbol{\theta}$ is the projection on the unit sphere. We then have $E_{\boldsymbol{\theta}\sim\mathcal{U}(\mathbb{S}^{p-1})}\left[\boldsymbol{\theta}^\top \Sigma\boldsymbol{\theta}\right] = \frac{1}{p}tr(\Sigma)$. Therefore, we obtain

$$
tr(\Sigma_{t+1}|_{\mathcal{D}_t}) - tr(\Sigma_{t+1}|_{\tilde{\mathcal{D}}_t}) = p \cdot E_{\boldsymbol{\theta}\sim\mathcal{U}(\mathbb{S}^{p-1})}\left[\boldsymbol{\theta}^\top \Sigma_{t+1}|_{\mathcal{D}_t}\boldsymbol{\theta} - \boldsymbol{\theta}^\top \Sigma_{t+1}|_{\tilde{\mathcal{D}}_t}\boldsymbol{\theta}\right].
$$

Hence,

$$
\boldsymbol{\theta}^\top(\Sigma_{t+1}|_{\mathcal{D}_t} - \Sigma_{t+1}|_{\tilde{\mathcal{D}}_t})\boldsymbol{\theta} = \left(\sqrt{\boldsymbol{\theta}^\top \Sigma_{t+1}|_{\mathcal{D}_t}\boldsymbol{\theta}} + \sqrt{\boldsymbol{\theta}^\top \Sigma_{t+1}|_{\tilde{\mathcal{D}}_t}\boldsymbol{\theta}}\right)\left(\sqrt{\boldsymbol{\theta}^\top \Sigma_{t+1}|_{\mathcal{D}_t}\boldsymbol{\theta}} - \sqrt{\boldsymbol{\theta}^\top \Sigma_{t+1}|_{\tilde{\mathcal{D}}_t}\boldsymbol{\theta}}\right)
$$

Without loss of generality, assume the operator (spectral) norms of $\sqrt{\boldsymbol{\theta}^\top \Sigma_{t+1}|_{\mathcal{D}_t} \boldsymbol{\theta}}$ and $\sqrt{\boldsymbol{\theta}^\top \Sigma_{t+1}|_{\tilde{\mathcal{D}}_t} \boldsymbol{\theta}}$ are uniformly bounded by $C$. We then have

$$\boldsymbol{\theta}^\top (\Sigma_{t+1}|_{\mathcal{D}_t} - \Sigma_{t+1}|_{\tilde{\mathcal{D}}_t}) \boldsymbol{\theta} \leq 2C \left( \sqrt{\boldsymbol{\theta}^\top \Sigma_{t+1}|_{\mathcal{D}_t} \boldsymbol{\theta}} - \sqrt{\boldsymbol{\theta}^\top \Sigma_{t+1}|_{\tilde{\mathcal{D}}_t} \boldsymbol{\theta}} \right)$$

Therefore, we obtain

$$tr(\Sigma_{t+1}|_{\mathcal{D}_t}) - tr(\Sigma_{t+1}|_{\tilde{\mathcal{D}}_t}) \leq 2Cp\kappa.$$

As established in the discussion of BO (Wu et al., 2023), there exists some constant $L > 0$ such that

$$tr(\Sigma_{t+1}|_{\tilde{\mathcal{D}}_t}) = tr(\nabla^2 K_{D \cup \mathbf{z}}(\boldsymbol{\theta}, \boldsymbol{\theta})) \leq L.$$

Consequently, we obtain that, for some constant $c_1 > 0$,

$$\|\nabla \mathcal{L}(\boldsymbol{\theta}_t, z_t) - \boldsymbol{\mu}_{\mathcal{D}_t}\|^2 \leq c_1 (L + p\kappa).$$

$\square$

**Lemma C.5.** *Let $g_t(\boldsymbol{\theta}_t)$ be defined as in Algorithm 1. Under Assumptions 4.1 and 4.2, there exists some constant $c_1 > 0$ such that*

$$\|g_t(\boldsymbol{\theta}_t) - g(\boldsymbol{\theta}_t)\|^2 \leq 2B^2.$$

*Proof of Lemma C.5.* Using Lemma C.3, the effect of the projection operator $\Pi_B$ can be removed from the analysis. Consequently, we obtain

$$\begin{aligned}
\|g_t(\boldsymbol{\theta}_t) - g(\boldsymbol{\theta}_t)\|^2 &= \left\| \Pi_B \left( \boldsymbol{\mu}_{\mathcal{D}_t}(z_t) \right) - \Pi_B (\nabla K(\boldsymbol{\theta}_t, \mathcal{D}_t) K(\mathcal{D}_t, \mathcal{D}_t)^{-1} f(\boldsymbol{\theta}_t)) \right\|^2 \\
&\leq \left\| \Pi_B \left( \boldsymbol{\mu}_{\mathcal{D}_t}(z_t) \right) \right\|^2 + \left\| \Pi_B (\nabla K(\boldsymbol{\theta}_t, \mathcal{D}_t) K(\mathcal{D}_t, \mathcal{D}_t)^{-1} f(\boldsymbol{\theta}_t)) \right\|^2 \\
&\leq B^2 + B^2 \\
&\leq 2B^2.
\end{aligned}$$

$\square$

**Lemma C.6.** *(1) Suppose that $f \colon \mathbb{R}^p \to \mathbb{R}$ is a $\lambda$-strongly convex function, we have*

$$\langle \nabla f(\boldsymbol{\theta}_1) - \nabla f(\boldsymbol{\theta}_2), \boldsymbol{\theta}_1 - \boldsymbol{\theta}_2 \rangle \geq \lambda \|\boldsymbol{\theta}_1 - \boldsymbol{\theta}_2\|_2^2, \quad \forall \boldsymbol{\theta}_1, \boldsymbol{\theta}_2 \in \mathbb{R}^p,$$

*and if $f$ is twice-differentiable, then $\nabla^2 f(\boldsymbol{\theta}) \succeq \lambda I, \quad \forall \boldsymbol{\theta} \in \mathbb{R}^p$.*

*(2) Suppose that $f \colon \mathbb{R}^p \to \mathbb{R}$ is a convex and $\zeta$-smooth function, we have for any $\boldsymbol{\theta}_1, \boldsymbol{\theta}_2 \in \mathbb{R}^p$,*

$$\|\nabla f(\boldsymbol{\theta}_1) - \nabla f(\boldsymbol{\theta}_2)\|_2^2 \leq \zeta \langle \nabla f(\boldsymbol{\theta}_1) - \nabla f(\boldsymbol{\theta}_2), \boldsymbol{\theta}_1 - \boldsymbol{\theta}_2 \rangle,$$

*and*

$$\|\nabla f(\boldsymbol{\theta}_1) - \nabla f(\boldsymbol{\theta}_2)\|_2 \leq \zeta \|\boldsymbol{\theta}_1 - \boldsymbol{\theta}_2\|_2.$$

*If $f$ is twice-differentiable, then $\nabla^2 f(\boldsymbol{\theta}) \preceq \zeta I, \quad \forall \boldsymbol{\theta} \in \mathbb{R}^p$.*

*Proof of Lemma C.6.* The results of this lemma are standard and can be found in the convex optimization literature; see, for example, Boyd & Vandenberghe (2004) for detailed proofs. $\square$

# D  ADDITIONAL EXPERIMENTAL RESULTS

## D.1  ADDITIONAL RESULTS

In this subsection, we provide details of data generating processes and additional results in Section 5.

**Example 5.1 (Continued).** We evaluate the proposed algorithm and the competing methods under linear, logistic and ReLU regression models, respectively.

**Linear regression**. We sample $T = 20000$ i.i.d. data points $\{(\boldsymbol{x}_t, y_t)\}_{t=1}^T$, where the covariates are drawn as $\boldsymbol{x}_t \sim N(0, \mathbf{I}_p)$, and the responses are generated according to

$$y_t = \boldsymbol{x}_t^\top \boldsymbol{\theta} + \varepsilon_t,$$

with true parameter vector $\boldsymbol{\theta} = \mathbf{1}_p$ and noise terms $\varepsilon_t \overset{\text{i.i.d.}}{\sim} N(0,1)$. We employ the Huber loss function $\rho_c$ with threshold $c = 1$, and incorporate gradient sensitivity control to ensure stability. The overall objective function is given by

$$\mathcal{L}(\boldsymbol{\theta}) = \frac{1}{T} \sum_{t=1}^T \rho_c\left(y_t - \boldsymbol{x}_t^\top \boldsymbol{\theta}\right) \cdot \min\left(1, \frac{2}{\|\boldsymbol{x}_t\|^2}\right).$$

This reweighting scheme effectively bounds the influence of high-magnitude gradients, serving as a form of implicit gradient clipping that enhances robustness during optimization.

**Logistic regression**. The feature vectors $\boldsymbol{x}_t \in \mathbb{R}^d$ are sampled independently from a standard normal distribution, $\boldsymbol{x}_t \sim N(0, \mathbf{I}_p)$. Binary labels $y_t \in \{-1, +1\}$ are generated according to the logistic model:

$$P(y_t = 1 \mid \boldsymbol{x}_t) = \frac{1}{1 + \exp(-\boldsymbol{x}_t^\top \boldsymbol{\theta})},$$

where the true parameter vector $\boldsymbol{\theta} = \mathbf{1}_p$ defines the underlying decision boundary. The learning objective is defined via the binary cross-entropy loss, which measures the discrepancy between the predicted probabilities and the true labels. Specifically, we minimize the following empirical risk:

$$\mathcal{L}(\boldsymbol{\theta}) = -\frac{1}{T} \sum_{t=1}^T \left[y_t \log(p_t) + (1 - y_t)\log(1 - p_t)\right] \cdot \min\left(1, \frac{2}{\|\boldsymbol{x}_t\|^2}\right),$$

where, $p_t = P(y_t = 1 \mid x_t)$ represents the predicted probability of the positive class for sample t, given by the sigmoid function applied to the linear combination of features and parameters.

**ReLU regression**. We generate synthetic data $\{(\boldsymbol{x}_t, y_t)\}_{t=1}^T$ according to the model:

$$y_t = \text{ReLU}(\boldsymbol{x}_t^\top \boldsymbol{\theta}),$$

with true parameter vector $\boldsymbol{\theta} = \mathbf{1}_p$. The objective is to minimize the squared loss, which quantifies the discrepancy between the predicted values and the true responses. The empirical risk is thus defined as:

$$\mathcal{L}(\boldsymbol{\theta}) = \frac{1}{T} \sum_{t=1}^T \rho_c\left(y_t - \text{ReLU}(\boldsymbol{x}_t^\top \boldsymbol{\theta})\right) \cdot \min\left(1, \frac{2}{\|\boldsymbol{x}_t\|^2}\right).$$

This setup allows us to evaluate how effectively each method can handle nonlinear transformations and non-continuous derivative functions, as introduced by the ReLU activation. By applying this nonlinearity, we test the robustness of various algorithms in approximating complex, discontinuous mappings while maintaining low prediction error.

Figure 4 presents additional results for $p = 5$. The first three columns of Figure 4 illustrate the trajectory of the first-dimensional coefficient estimate (true value = 1) across iterations in the $p = 5$ setting. For the linear model, both LDP-BO and LDP-SGD closely track their non-private counterparts. In nonlinear models (logistic and ReLU), however, BO-based methods consistently outperform SGD-based approaches under all privacy regimes. The last column of Figure 4 reports MSE of the parameter estimates, revealing that LDP-BO achieves consistently lower error and reduced variability compared to LDP-SGD in complex settings. Even under strong privacy constraints ($\varepsilon = 1$),

LDP-BO exhibits faster convergence and attains accuracy on par with non-private BO and SGD. These results underscore the modeling advantage of LDP-BO in handling nonlinear problems in moderate-dimensional ($p = 5$) scenarios, where it effectively mitigates the utility degradation often associated with gradient-based private optimization.

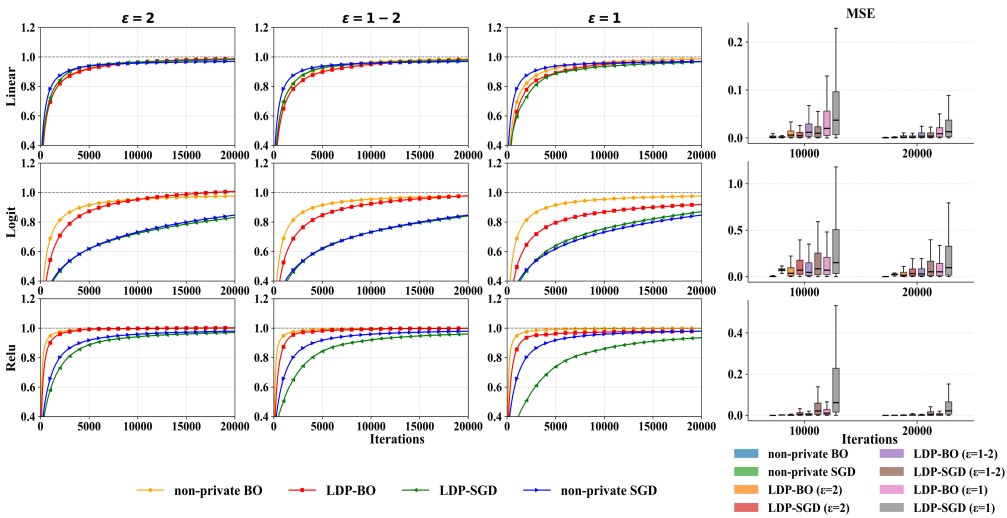

Figure 4: Left figure represents evolution of the first-dimension coefficient estimate (true value $= 1$) over iterations for linear, logistic, and ReLU models (rows) in Example 5.1. Columns correspond to privacy budgets $\varepsilon = 2$, $\varepsilon \sim U(1, 2)$, and $\varepsilon = 1$. Right figure represents boxplots of coefficient MSEs across three models under different privacy budgets in Example 5.1.

In addition, to assess performance in a moderate-dimensional scenario, we extended Example 5.1 to include experiments with covariate dimension $p = 20$. As shown in Table 3, LDP-BO continues to exhibit strong estimation and prediction accuracy. The conclusions mirror those in the low-dimensional setting: for a fixed privacy budget, LDP-BO consistently matches or outperforms LDP-SGD across linear, logit, and ReLU regression models.

Table 4 compares the runtime (in minutes) between LDP-BO and LDP-SGD across different models and dimensions, based on 50 replications. As expected, LDP-BO consistently takes more time than LDP-SGD due to the inherent exploration process of Bayesian Optimization, which is unavoidable. However, the results clearly show that LDP-BO significantly outperforms LDP-SGD, particularly in more complex models (Logit and ReLU). This demonstrates the trade-off between time and performance, where LDP-BO sacrifices some computational efficiency for much better results in challenging settings.

The compression budget strikes a balance between prediction time and prediction accuracy. A smaller compression budget retains more essential information, leading to improved results at the cost of increased computational time. Figure 5 further illustrates the impact of different compression budgets (0.1 and 0.2) on the performance of linear, logistic, and ReLU regression models under varying privacy budgets ($\varepsilon = 2$, $\varepsilon = U(1, 2)$, and $\varepsilon = 1$). Across all settings, a smaller compression budget (0.1, represented by red lines) consistently leads to better performance compared to a larger budget (0.2, represented by blue lines), as evidenced by faster convergence and higher final accuracy. This improvement is particularly pronounced in complex models such as logistic and ReLU regression, where the underlying data structure is more nonlinear and intricate. In these cases, a smaller compression budget helps preserve a greater amount of critical kernel information during the Bayesian optimization process, which is essential for accurately modeling complex decision boundaries. Therefore, tighter compression—achieved through a smaller budget—is especially beneficial in complex models, as it enables the algorithm to retain more informative data points, leading to more reliable and accurate parameter estimates. The results suggest that carefully controlling the compression budget is crucial for balancing efficiency and utility, with more complex problems generally requiring stricter (i.e., smaller) compression budgets to achieve optimal performance.

Table 3: MSE $(\times 10^{-3})$ of LDP-BO and LDP-SGD for linear, logit and ReLU regression with $p = 20$ under different privacy levels. Means (standard deviations) are computed over 50 repetitions.

| Model | Privacy level | $t$ | LDP-BO | LDP-SGD |
|---|---|---|---|---|
| Linear | No DP | 5,000 | 8.79 (3.08) | 12.56 (5.65) |
| | | 10,000 | 2.78 (0.97) | 3.97 (1.79) |
| | | 15,000 | 1.29 (0.45) | 1.84 (0.83) |
| | | 20,000 | 0.73 (0.26) | 1.05 (0.47) |
| | $\varepsilon = 2$ | 5,000 | 19.75 (6.91) | 28.21 (12.69) |
| | | 10,000 | 9.37 (3.28) | 13.39 (6.03) |
| | | 15,000 | 5.06 (1.77) | 7.23 (3.25) |
| | | 20,000 | 3.04 (1.06) | 4.35 (1.96) |
| Logit | No DP | 5,000 | 4.35 (1.52) | 6.22 (2.80) |
| | | 10,000 | 1.17 (0.41) | 1.67 (0.75) |
| | | 15,000 | 0.52 (0.18) | 0.745 (0.34) |
| | | 20,000 | 0.29 (0.10) | 0.418 (0.19) |
| | $\varepsilon = 2$ | 5,000 | 31.56 (11.05) | 57.39 (25.83) |
| | | 10,000 | 24.99 (8.75) | 45.44 (20.45) |
| | | 15,000 | 21.07 (7.37) | 38.31 (17.24) |
| | | 20,000 | 18.33 (6.42) | 33.33 (15.00) |
| ReLU | No DP | 5,000 | 4.40 (1.54) | 6.28 (2.83) |
| | | 10,000 | 1.20 (0.42) | 1.71 (0.77) |
| | | 15,000 | 0.54 (0.19) | 0.77 (0.35) |
| | | 20,000 | 0.30 (0.10) | 0.43 (0.19) |
| | $\varepsilon = 2$ | 5,000 | 28.86 (10.10) | 52.48 (23.62) |
| | | 10,000 | 21.26 (7.44) | 38.66 (17.40) |
| | | 15,000 | 16.92 (5.92) | 30.77 (13.85) |
| | | 20,000 | 13.18 (4.61) | 23.97 (10.79) |

Table 4: Runtime comparison (in minutes) between LDP-BO and LDP-SGD for different models and dimensions over 50 replications.

| Model | Linear | | Logit | | ReLU | |
|---|---|---|---|---|---|---|
| | LDP-BO | LDP-SGD | LDP-BO | LDP-SGD | LDP-BO | LDP-SGD |
| $p = 2$ | 29.58 | 0.78 | 31.78 | 0.84 | 32.33 | 0.80 |
| $p = 5$ | 75.55 | 1.45 | 138.92 | 1.73 | 144.08 | 1.51 |
| $p = 20$ | 92.78 | 3.12 | 145.42 | 3.85 | 148.52 | 3.20 |

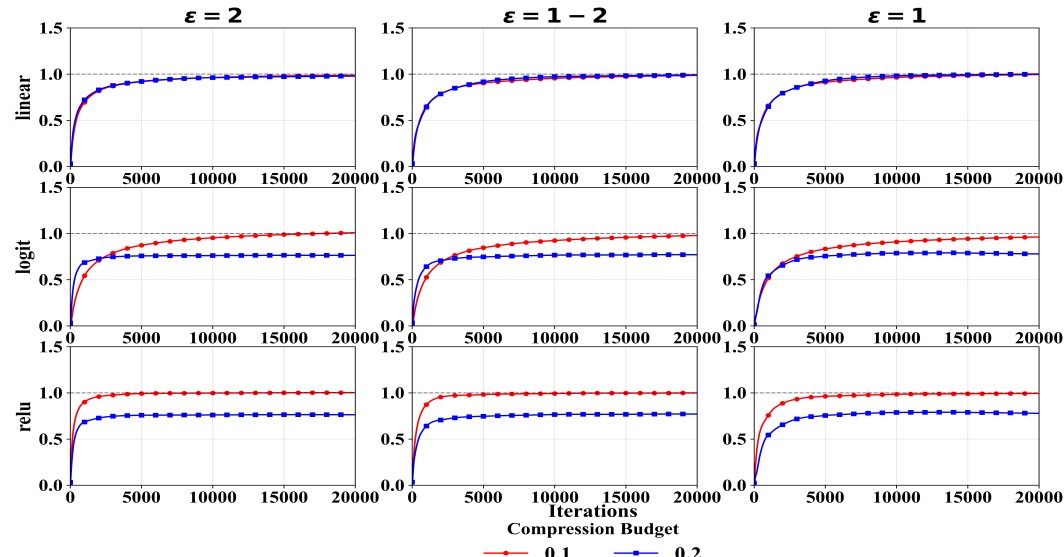

Figure 5: Results of experiments with different compression budget, where dimension $p = 5$, and privacy budget $\delta = 0.2$. Each row corresponds to a different model: linear regression, logistic regression, and ReLU regression. Each column represents a different privacy budget $\varepsilon = 2, Unif(1, 2), 1$, ordered from highest to lowest noise intensity.

**Example 5.2 (Continued).** In this example, we perform LDP-BO with $(\varepsilon, \delta) = (1, 0.2), \kappa = 0.1$ and $B = 2$. The following is a detailed description of the models, including the Sine function and the Friedman function.

**Sine function.** We apply an exact Gaussian process regression model designed under privacy constraints. The model employs a constant mean function $m(\boldsymbol{x}) = 0$ and a scaled radial basis function (RBF) covariance kernel:

$$K(\boldsymbol{x}, \boldsymbol{x}') = \sigma_{\text{output}}^2 \exp\left(-\frac{\|\boldsymbol{x} - \boldsymbol{x}'\|^2}{2\ell^2}\right),$$

The kernel contains two trainable parameters: the length scale $\ell$, which controls the smoothness of the function, and the output scale $\sigma_{\text{output}}$, which modulates the amplitude of the output. The model is trained by minimizing the negative log marginal likelihood (NLL), which serves as our objective function:

$$\mathcal{L}(\boldsymbol{\theta}) = -\log p(y \mid \boldsymbol{x}, \boldsymbol{\theta}) = \frac{1}{2}y^\top K_y^{-1}y + \frac{1}{2}\log|K_y| + \frac{1}{2}\log(2\pi),$$

where $K_y = K + \sigma_{\text{noise}}^2\mathbf{I}$ denotes the noise-perturbed covariance matrix. This loss function naturally balances data fit (first term) and model complexity (second term), providing a probabilistically principled measure of model adequacy. We set $\sigma_{\text{noise}}^2 = 10^{-4}$.

We optimize the parameters in log space to ensure positivity and improve numerical stability. The trainable parameter vector is thus $\boldsymbol{\theta} = (\log \ell, \log \sigma_{\text{output}})$, making this a two-dimensional optimization problem. The actual kernel parameters are recovered via exponentiation: $\ell = \exp(\log \ell)$, $\sigma_{\text{output}} = \exp(\log \sigma_{\text{output}})$. This formulation enables efficient Bayesian optimization of the kernel parameters while providing a tractable and interpretable objective for privacy-preserving parameter optimization. The entire framework offers a rigorous foundation for adaptive, nonparametric regression under DP constraints.

**Friedman function.** We propose an adaptive Gaussian process GP regression framework employing automatic relevance determination (ARD) to handle multidimensional input spaces in sequential

learning scenarios. The model utilizes a constant mean function and a scaled radial basis function (RBF) covariance kernel with ARD:

$$K(\boldsymbol{x}, \boldsymbol{x}') = \sigma_{\text{output}}^2 \exp\left(-\frac{1}{2}\sum_{j=1}^{p}\frac{(x_j - x_j')^2}{\ell_j^2}\right),$$

where each input dimension $p$ has its own trainable length scale $\ell_j$, allowing the model to automatically learn the relevance of each feature. The output scale $\sigma_{\text{output}}$ can be either optimized or fixed to modulate function amplitude. In our simulations, we fixed it to 1.

The training objective minimizes the negative log marginal likelihood:

$$\mathcal{L}(\boldsymbol{\theta}) = -\log p(y \mid \boldsymbol{x}, \boldsymbol{\theta}) = \frac{1}{2}y^\top K_y^{-1}y + \frac{1}{2}\log|K_y| + \frac{1}{2}\log(2\pi),$$

where $\boldsymbol{\theta} = (\log \ell_1, \log \ell_2, \ldots, \log \ell_p)$ represents the $p$-dimensional hyperparameter vector optimized in log space to ensure positivity and numerical stability. The ARD formulation enables automatic feature selection by assigning larger length scales to less relevant dimensions, effectively suppressing their contribution to the covariance function.

This approach provides a principled probabilistic framework for high-dimensional regression, with the optimization complexity scaling linearly with the input dimension $p$. The model maintains computational tractability through exact inference while offering interpretable insights into feature relevance through the learned length scales, making it particularly suitable for Bayesian optimization in parameterized spaces.

We included cumulative regret evaluations for the Sine and Friedman test functions from Example 5.2. Unlike the earlier parameter-estimation examples, this analysis focuses on predictive performance. As shown in Table 5, LDP-BO attains substantially lower cumulative regret than the DNN-based baseline on both benchmarks. This demonstrates that, under the same privacy constraints, our method is much more sample-efficient and can identify high-reward regions of the search space significantly faster than the competing approach, highlighting its effectiveness in prediction tasks.

Table 5: Cumulative regret on the Sine and Friedman functions.

| Method | Sine | Friedman |
|---|---|---|
| LDP-BO | 207.873 | 1270.889 |
| DNN-based baseline | 622.921 | 2275.447 |

**Example 5.3 (Continued).** The Uber Fares Dataset preprocessing pipeline starts with comprehensive cleaning to enhance data robustness. We remove records with invalid fare amounts, such as negative values or extreme outliers beyond predefined percentile thresholds, and handle missing values in key fields. Following this, feature engineering is conducted to extract meaningful signals from the raw data.

Original features such as `passenger_count` are retained to account for the impact of group travel on fare pricing. Spatial information is derived from the provided geographic coordinates: `pickup_longitude` and `pickup_latitude` (indicating where the trip began), along with `dropoff_longitude` and `dropoff_latitude` (marking the destination). From these, we compute the Manhattan distance between pickup and drop-off points—a more accurate proxy for actual travel distance in New York City's grid-like street layout than Euclidean distance.

Temporal patterns are captured by extracting features from the `pickup_patetime` field, including the hour of the day and day of the week, which help model variations in demand, traffic congestion, and surge pricing dynamics.

The final feature set combines cleaned original variables with these engineered spatial and temporal features, forming the input for downstream regression models designed to accurately predict fare amounts. We adopt a Gaussian regression framework with a 4-dimensional parameter space for

possible complex relationships. Among privacy-preserving methods, LDP-SGD applied to a linear model is the only one supporting both LDP and online parameter estimation; thus, we use it as a baseline for comparing prediction error across methods.

Figure 6 compares the performance of LDP-BO and LDP-SGD under $(\varepsilon, \delta) = (1, 0.2)$ across sample sizes of 5000, 10000, and 20000 in terms of the prediction error. It show that LDP-BO consistently outperforms LDP-SGD across all metrics, achieving lower prediction error and exhibiting narrower interquartile ranges as sample size increases. This trend indicates reduced estimation variance and improved stability for LDP-BO.

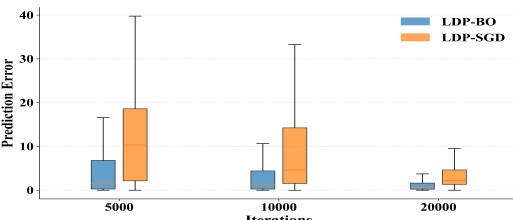

Figure 6: Fare prediction errors of LDP-BO and LDP-SGD in Example 5.3.

Credit Card Fraud Detection Dataset comprises approximately 20,000 transaction records made in September 2013. We construct a Logistic Regression model using PCA-transformed features from the dataset. Since the data is already in its principal component form, no additional preprocessing is required. We use the top 5 principal components to capture the most significant variations in the data, following common practice in fraud detection studies (Bestami Yuksel et al., 2020; Ogundile et al., 2024). The target variable is whether the transaction is fraudulent (1) or legitimate (0).

The table 6 presents the results of an ablation study on the choice of $\kappa$ for the Uber regression task and the Credit classification task. For the Uber dataset, the average prediction error is reported, while for the Credit dataset, we report the classification accuracy. As $\kappa$ increases, we observe that the performance for Uber degrades, with a notable increase in average prediction error, particularly for $\kappa = 0.5$. On the other hand, for the Credit dataset, accuracy decreases as $\kappa$ increases, with a sharp drop for $\kappa = 0.5$.

Given the trade-off between performance and computational time, we choose $\kappa = 0.1$ as a reasonable compromise. This value provides a good balance between accuracy and efficiency, as demonstrated by its results, which are relatively close to the best-performing configurations for both tasks. We therefore use $\kappa = 0.1$ for comparisons with other methods in the main body of the text.

Table 6: Ablation on $\kappa$ for the Uber regression task and the Credit classification task. For Uber we report average prediction error, for Credit we report accuracy.

| $\kappa$ | Uber | Credit |
|---|---|---|
| 0.05 | 0.711 | 0.971 |
| 0.10 | 0.782 | 0.969 |
| 0.20 | 1.243 | 0.958 |
| 0.50 | 3.745 | 0.921 |

## D.2 ABLATIONS

we have added comprehensive ablation and sensitivity studies. Specifically, we conduct these experiments on the linear regression model from Example 5.1, where the response is generated as $y_t = \boldsymbol{x}_t^\top \boldsymbol{\theta}^\star + \varepsilon_t$. We systematically vary three key parameters of our proposed LDP-BO procedure and evaluate their effect on the MSE: the privacy budget $\varepsilon \in [0.5, 10]$, the initial step size $\gamma_0 \in [0.1, 2]$ in the schedule $\eta_t = \gamma_0 t^{-\alpha}$, and the compression threshold $\kappa \in [0.01, 0.5]$. Table 7 reports the results for different choices of these tuning parameters. In each experiment, a single parameter is varied while the remaining parameters are fixed at their default values.

The findings indicate the following:

- Privacy budget $\varepsilon$: increasing $\varepsilon$ weakens privacy protection and consequently improves estimation accuracy;

- Initial step size $\gamma_0$: the proposed method is robust to the choice of initial step size over a broad range;

- Compression threshold $\kappa$: $\kappa$ induces a clear trade-off between estimation quality and runtime, with smaller values leading to faster execution but slightly reduced accuracy.

Table 7: Ablation study on $\varepsilon$, $\gamma_0$ and compression parameter $\kappa$. Reported values are MSE ($\times 10^{-3}$) averaged over 50 repetitions; computation time for different values of $\kappa$ is given in minutes.

| Privacy Budget $\varepsilon$ | | Initial Step Size $\gamma_0$ | | Compression Parameter $\kappa$ | | |
|---|---|---|---|---|---|---|
| $\varepsilon$ | MSE ($\times 10^{-3}$) | $\gamma_0$ | MSE ($\times 10^{-3}$) | $\kappa$ | Time (minutes) | MSE ($\times 10^{-3}$) |
| 0.5 | 18.10 | 0.1 | 8.61 | 0.01 | 318.7 | 1.41 |
| 1 | 3.46 | 0.2 | 1.83 | 0.05 | 165.3 | 1.59 |
| U(1, 2) | 2.73 | 0.3 | 2.01 | 0.10 | 33.0 | 1.88 |
| 2 | 1.81 | 0.5 | 2.41 | 0.20 | 29.2 | 4.50 |
| 5 | 1.65 | 1 | 9.63 | 0.50 | 18.8 | 16.30 |
| | | 2 | 20.10 | | | |

In practice, $\kappa$ reflects the trade-off between computational efficiency and predictive accuracy. A simple approach is to perform cross-validation on a small held-out prefix of the data stream over a short grid of $\kappa$ values, and select the largest $\kappa$ that maintains acceptable prediction error. This procedure is fast and avoids extensive hyperparameter searches.

### D.3 Non-stationary streaming data

We have added experimental studies for non-stationary settings, focusing on parameter drift in the linear model of Example 5.1. These experiments use privacy parameters ($(\varepsilon, \delta) = (2, 0.2)$) and a compression budget of $\kappa = 0.1$ in $T = 20000$ samples. Following (Barber et al., 2023), we consider two types of non-stationarity:

- **Case 1: Abrupt regime shifts.** The regression coefficient $\boldsymbol{\theta}$ switches among three fixed vectors over successive time segments:

$$\boldsymbol{\theta}^{(1)} = (1, 2, 1, 0, 0), \quad \boldsymbol{\theta}^{(2)} = (0, -1, -2, -1, 0), \quad \boldsymbol{\theta}^{(3)} = (0, 0, 1, 2, 1),$$

  with $\boldsymbol{\theta}_t = \boldsymbol{\theta}^{(1)} \mathbb{I}(1 \le t \le T/3) + \boldsymbol{\theta}^{(2)} \mathbb{I}(T/3 < t \le 2T/3) + \boldsymbol{\theta}^{(3)} \mathbb{I}(2T/3 < t \le T)$.

- **Case 2: Smooth concept drift.** The regression coefficient evolves linearly from

$$\boldsymbol{\theta}_{\text{start}} = (1, 2, 1, 0, 0), \qquad \boldsymbol{\theta}_{\text{end}} = (0, 0, 1, 2, 1),$$

  according to $\boldsymbol{\theta}_t = (1 - \alpha_t)\,\boldsymbol{\theta}_{\text{start}} + \alpha_t\,\boldsymbol{\theta}_{\text{end}}, \; \alpha_t = (t-1)/(T-1)$.

Table **??** shows that LDP-BO consistently outperforms LDP-SGD in both cases, achieving lower prediction error and more stable performance under the same $(\varepsilon, \delta)$-LDP budget, and approaching the performance of the non-private baseline. The suboptimal result at 15,000 samples in Case 1 corresponds to the regime shift around 13,000 samples; with larger sample sizes, LDP-BO converges more rapidly than LDP-SGD.

Similar to (Barber et al., 2023), we generate data via $\boldsymbol{x}_t \sim \mathcal{N}(0, \mathbf{I}_5)$ and $y_t = \boldsymbol{x}_t^\top \boldsymbol{\theta}_t + \varepsilon_t$ for $t = 1, \ldots, T = 20{,}000$, where $\boldsymbol{\theta}_t \in \mathbb{R}^5$ and $\varepsilon_t \sim \mathcal{N}(0, 1)$ is Gaussian noise. We consider the following two scenarios:

1. **Abrupt regime shifts**: We consider $T = 20{,}000$ observations and define three fixed coefficient vectors

$$\boldsymbol{\theta}^{(1)} = (1, 2, 1, 0, 0), \quad \boldsymbol{\theta}^{(2)} = (0, -1, -2, -1, 0), \quad \boldsymbol{\theta}^{(3)} = (0, 0, 1, 2, 1).$$

The time horizon $\{1, \ldots, T\}$ is equally divided into three segments, and we set

$$\boldsymbol{\theta}_t = \begin{cases} \boldsymbol{\theta}^{(1)}, & 1 \le t \le T/3, \\ \boldsymbol{\theta}^{(2)}, & T/3 < t \le 2T/3, \\ \boldsymbol{\theta}^{(3)}, & 2T/3 < t \le T. \end{cases}$$

In other words, with $T = 20{,}000$, abrupt regime shifts occur at the two equally spaced change points $t = T/3$ and $t = 2T/3$.

2. **Smooth concept drift**: We let $\boldsymbol{\theta}_t$ evolve linearly over time:

$$\boldsymbol{\theta}_t = (1 - \alpha_t)\boldsymbol{\theta}_{\text{start}} + \alpha_t \boldsymbol{\theta}_{\text{end}}, \quad \alpha_t = \frac{t-1}{T-1},$$

where $\boldsymbol{\theta}_{\text{start}} = (1, 2, 1, 0, 0)$ and $\boldsymbol{\theta}_{\text{end}} = (0, 0, 1, 2, 1)$.

Table 8 presents the MSE ($\times 10^{-2}$) of LDP-BO and LDP-SGD across two cases, where $(\epsilon, \delta) = (2, 0.2)$ and $\kappa = 0.1$. In Case 1, LDP-BO consistently outperforms LDP-SGD, particularly as the sample size increases. The performance gap becomes more significant in Case 2, where the data exhibits more complexity. LDP-BO remains more robust and accurate in handling non-stationary data, demonstrating superior performance over LDP-SGD even as the sample size grows. This highlights the advantage of LDP-BO in adapting to evolving data streams, where changes or fluctuations in the data are more pronounced.

Table 8: MSE ($\times 10^{-2}$) of LDP-BO and LDP-SGD in Case 1 and Case 2 under different privacy levels. Means (standard deviations) are computed over 50 repetitions.

| Case | Privacy level | $t$ | LDP-BO | LDP-SGD |
|------|---------------|------|--------|---------|
| Case 1 | No DP | 5,000 | 1.11 (0.58) | 1.59 (0.72) |
| | | 10,000 | 0.60 (0.31) | 0.86 (0.39) |
| | | 15,000 | 0.93 (0.48) | 1.33 (0.60) |
| | | 20,000 | 1.08 (0.56) | 1.55 (0.70) |
| | $\varepsilon = 2$ | 5,000 | 1.48 (0.76) | 2.11 (0.95) |
| | | 10,000 | 1.20 (0.62) | 1.71 (0.77) |
| | | 15,000 | 55.44 (28.51) | 79.20 (35.64) |
| | | 20,000 | 3.42 (1.76) | 4.88 (2.20) |
| Case 2 | No DP | 5,000 | 3.29 (1.70) | 4.70 (2.12) |
| | | 10,000 | 1.72 (0.89) | 2.46 (1.11) |
| | | 15,000 | 1.54 (0.79) | 2.20 (0.99) |
| | | 20,000 | 1.59 (0.82) | 2.27 (1.02) |
| | $\varepsilon = 2$ | 5,000 | 6.10 (3.14) | 8.72 (3.92) |
| | | 10,000 | 3.33 (1.71) | 4.76 (2.14) |
| | | 15,000 | 3.94 (2.02) | 5.63 (2.53) |
| | | 20,000 | 5.83 (3.00) | 8.33 (3.75) |

### D.4 COMPARISON OF KERNEL MATRIX APPROXIMATION METHODS

We have compared SWC with two widely used kernel matrix approximation methods: random feature truncation (Liu et al., 2021) and Nyström approximation (Abedsoltan et al., 2024). Random Feature Truncation selects a fixed-dimensional subset of features by a low-dimensional random feature space. Nyström approximation selects a set of reference points approximate to the kernel matrix. We apply all three kernel approximation methods to the three models in Example 5.1 (linear, logistic, and ReLU regression), using exactly the same parameter settings as in that example, see Pages 30-31. To isolate the effect of approximation, no privacy noise is added. All methods are evaluated on prediction error and kernel computation time. For fairness, the baselines use a fixed feature budget of $M_t = 128$ while SWC adaptively selects its effective order $M_t$ via data-driven pruning based on the threshold $\kappa$. As reported in Table 9, SWC achieves lower prediction error

with fewer components ($M_t < 128$) and competitive kernel computation time. Unlike fixed-budget methods, SWC maintains per-iteration efficiency independent of $t$, remaining tractable in large-scale online settings while preserving strong estimation performance.

Table 9: Comparison of SWC with random feature truncation and Nyström approximation over 50 repetitions.

| Model | Metric | SWC | Random | Nyström |
|---|---|---|---|---|
| linear | MSE ($\times 10^{-3}$) | 2.21 | 81.9 | 2.83 |
| | $M_t$ | 31 | 128 | 128 |
| | Time/s | $5.8 \times 10^{-3}$ | $2.5 \times 10^{-5}$ | $1.6 \times 10^{-2}$ |
| ReLU | MSE ($\times 10^{-3}$) | 1.58 | 13.9 | 3.05 |
| | $M_t$ | 44 | 128 | 128 |
| | Time/s | $7.3 \times 10^{-3}$ | $2.4 \times 10^{-5}$ | $1.9 \times 10^{-2}$ |
| Logit | MSE ($\times 10^{-3}$) | 4.27 | 41.8 | 6.73 |
| | $M_t$ | 61 | 128 | 128 |
| | Time/s | $9.6 \times 10^{-3}$ | $2.5 \times 10^{-5}$ | $2.1 \times 10^{-2}$ |

We further evaluate the variation of the kernel matrix order ($M_t$) over 50 simulations for different models (Linear, Logit, and ReLU) using the Sliced Wasserstein Compression (SWC) method. As shown in the figure, the kernel matrix order does not grow to the upper bound. Instead, it primarily depends on the model complexity: the more complex the model, the higher the matrix order. However, even in more complex models such as Logit and ReLU, $M_t$ remains significantly lower than the upper bound, demonstrating that SWC adapts to the data distribution and efficiently compresses the kernel matrix without excessive increase in order. This behavior highlights SWC's ability to manage computational complexity effectively while preserving model accuracy, making it well-suited for dynamic and non-stationary data scenarios where model complexity can vary.

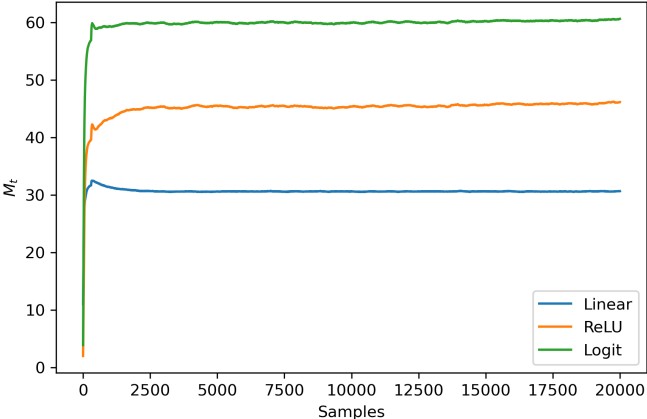

Figure 7: Variation of kernel matrix order ($M_t$) over 50 simulations for different models.

### D.5 MORE PRIVACY MECHANISMS

Our framework is compatible with standard DP mechanisms—Gaussian, Laplace, GDP Dong et al. (2022), RDP (Mironov, 2017), etc., as long as the noise scale is calibrated using the derived sensitivity. We further provides a clearer and unified description of calibration across mechanisms. We have compared four mechanisms: direct ($\varepsilon, \delta$)-DP calibration, GDP, RDP, and Laplace, under the same privacy budget ($\varepsilon, \delta$) = (2, 0.2), converting each to an equivalent ($\varepsilon, \delta$)-guarantee for linear model of Example 5.1. Table 10 reports empirical performance. Results show that our conclusions are robust across mechanisms, with GDP calibration yielding the strongest predictive accuracy under matched privacy guarantees.

Table 10: MSE ($\times 10^{-3}$) with standard deviations for various privacy mechanisms evaluated at different sample sizes.

|                        | 5000        | 10000       | 15000        | 20000        |
| ---------------------- | ----------- | ----------- | ------------ | ------------ |
| $(\varepsilon, \delta)$-DP | 6.05 (1.10) | 1.62 (0.35) | 0.976 (0.21) | 0.513 (0.12) |
| $\varepsilon$-DP       | 6.80 (1.25) | 1.85 (0.40) | 1.10 (0.25)  | 0.600 (0.14) |
| $\mu$-GDP              | 3.81 (0.75) | 1.34 (0.28) | 0.66 (0.14)  | 0.34 (0.08)  |
| RDP                    | 4.46 (0.85) | 1.46 (0.30) | 0.77 (0.16)  | 0.46 (0.11)  |

# E    DISCUSSIONS

## E.1    COMPUTATIONAL COMPLEXITY OF SWC

At online step $t$, let the current (uncompressed) dictionary be $\widetilde{D}_t$ with size $\tilde{M}_t = |\widetilde{D}_t| = |D_{t-1}| + 1$. Algorithm 2 iteratively removes points from $\widetilde{D}_t$ until the sliced Wasserstein distance between the compressed dictionary $D_t$ and $\widetilde{D}_t$ exceeds the budget $\kappa$. In each iteration, SWC computes $\eta_j = \mathrm{SW}_2(\rho_{D_{-j}}, \rho_{\widetilde{D}_t})$ for all $j$ in the current index set $\mathcal{I}$ and removes the index with minimal distance. Hence, in the worst case the algorithm evaluates at most $1 + 2 + \cdots + \tilde{M}_t = \mathcal{O}(\tilde{M}_t^2)$ sliced Wasserstein distances. A single sliced Wasserstein distance computed with $L$ random projections in $\mathbb{R}^p$ has cost

$$C_{\mathrm{SW}}(\tilde{M}_t) = \mathcal{O}\big(L(\tilde{M}_t \log \tilde{M}_t + p\tilde{M}_t)\big),$$

following standard implementations of sliced Wasserstein metrics (e.g., Rabin et al. (2011); Bonneel et al. (2015)). Therefore the total cost of SWC at step $t$ is

$$\mathcal{O}\big(\tilde{M}_t^2 C_{\mathrm{SW}}(\tilde{M}_t)\big) = \mathcal{O}\big(L\tilde{M}_t^3 \log \tilde{M}_t + Lp\tilde{M}_t^3\big).$$

Crucially, Theorem 3.3 shows that, for fixed compression budget $\kappa$ and dimension $p$, the dictionary size $\tilde{M}_t$ is uniformly bounded for all $t$. As a consequence, the per-iteration complexity of SWC is $\mathcal{O}(1)$ with respect to the time index $t$. In practice, the values of $\tilde{M}_t$ observed in our experiments lie in a moderate range, so the $\tilde{M}_t^2$ factor remains small and the resulting runtime is far below that of traditional GP-based BO, whose memory and time costs grow at least as $\mathcal{O}(t^2)$ with the number of observations. If the complexity remains too high, one possible approach to further reduce it is to use low-rank updates, which we consider as a potential strategy for future work to optimize the complexity.

**Computational time.** We compare the computational efficiency of different methods on a desktop computer equipped with a 3.00 GHz Intel Core i7-9700 CPU and 8GB RAM. Computational times are recorded for sample sizes ranging from $n = 200$ to $n = 2000$.

Figure 8 shows the computational time versus sample size for three methods: our proposed LDP-BO, the offline method Sopa et al. (2025), and the online method without SWC. As the sample size increases, LDP-BO demonstrates nearly constant computational time, reflecting its linear complexity $\mathcal{O}(t)$. In contrast, both Offline and Without SWC methods show cubic growth, indicating $\mathcal{O}(t^3)$ complexity. The MSE comparison in Table 11 demonstrates that our LDP-BO method, even with compression (SWC), incurs only a minimal loss in accuracy, further confirming the effectiveness of our approach in balancing both runtime and performance.

Table 11: Comparison of MSE for different sample sizes $n$ over 50 repetitions.

| Method | Sample Size $n$ | | | | |
| ------ | ------------- | ------------- | ------------- | ------------- | ------------- |
|        | 200           | 500           | 1000          | 1500          | 2000          |
| LDP-BO | 0.120 (0.020) | 0.090 (0.015) | 0.075 (0.010) | 0.060 (0.008) | 0.055 (0.007) |
| No SWC | 0.110 (0.020) | 0.080 (0.014) | 0.068 (0.009) | 0.055 (0.007) | 0.052 (0.006) |
| Offline | 0.090 (0.015) | 0.055 (0.010) | 0.045 (0.008) | 0.043 (0.007) | 0.041 (0.006) |

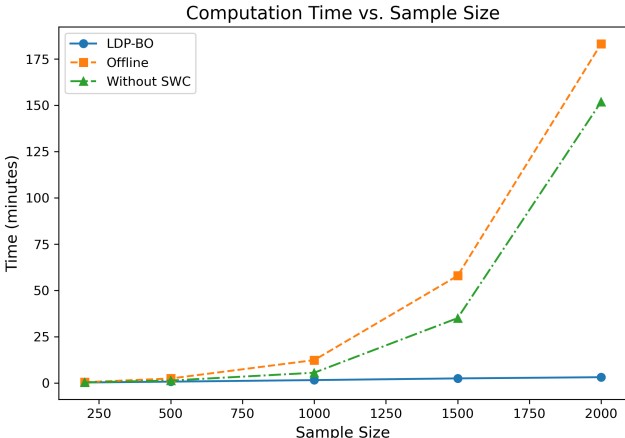

Figure 8: Change in computation times of our proposed LDP-BO and baselines (Offline and LDP-BO without SWC) as sample size increases from 200 to 2000 in Example 5.1 (Linear Model) over 50 repetitions.

### E.2 CLIPPING V.S. MALLOW'S WEIGHTING

In practice, we employ Mallow's weights rather than gradient clipping to ensure boundedness. Mallow-type weighting directly adjusts the loss rather than truncating the estimating equation. For example, in the linear regression setting, the empirical loss is

$$\mathcal{L}(\boldsymbol{\theta}, \boldsymbol{x}_t, y_t) = \rho_c\big(y_t - \boldsymbol{x}_t^\top \boldsymbol{\theta}\big) \, \min\left(1, \frac{2}{\|\boldsymbol{x}_t\|^2}\right),$$

where $\rho_c(\cdot)$ is a Huber-type loss and $\min(1, 2/\|\boldsymbol{x}_t\|^2)$ is a Mallows-type weight that caps the influence of large covariate values. It preserves consistency and asymptotic unbiasedness even under noise or privacy constraints. In contrast, gradient clipping alters the estimating equation itself and typically introduces a non-vanishing bias that depends on the clipping threshold.

Prior work by Avella-Medina et al. (2023) and Xie et al. (2025) has shown that Mallow-type weighting yields consistent estimators under privacy, whereas clipping may lead to biased or unstable estimates. To illustrate this in our setting, we replicate Example 5.1 with a logistic regression model under Mallow weighting $\omega(\boldsymbol{x}) = \min(1, 2/\|\boldsymbol{x}\|^2)$ and under cliping bound $\sqrt{2}$. This setting ensures that both methods have the same sensitivity. Table 12 shows that Mallow weighting retains tight concentration around the true value (1.0) across all privacy levels, while clipping consistently produces upward-biased estimates.

Table 12: Mean (standard deviation) of the estimated value under the logistic model across 50 replications.

| Method | No DP | $\varepsilon = 2$ | $\varepsilon \in [1, 2]$ | $\varepsilon = 1$ |
|---|---|---|---|---|
| Mallow weights | $1.00\,(0.02)$ | $0.99\,(0.05)$ | $1.02\,(0.06)$ | $0.98\,(0.08)$ |
| Clipping | $1.15\,(0.02)$ | $1.18\,(0.05)$ | $1.20\,(0.07)$ | $1.19\,(0.08)$ |

### E.3 EMPIRICAL VERIFICATION ASSUMPTION 3.2

Assumption 3.2 is a mild assumption, relying on a consistency property formalized in Lemma C.1. This consistency and non-expansive projection assumption is standard in the online Gaussian process regression and nonparametric Bayesian regression literature (e.g., Schmidhuber (2015); Koppel et al. (2021)). To empirically validate Assumption 3.2, we performed an ablation study comparing LDP-BO with and without SWC using the linear regression model from Example 5.1. To visually verify Assumption 3.2, we did not apply any privacy protection in this experiment and set $\kappa = 0.1$.

Figure 9 show that the SW distance increases after applying compression (SWC), indicating more variability. However, this does not lead to a higher probability of divergence compared to the original model, confirming that compression does not negatively impact the model's ability to learn and update, as stated in Assumption 3.2.

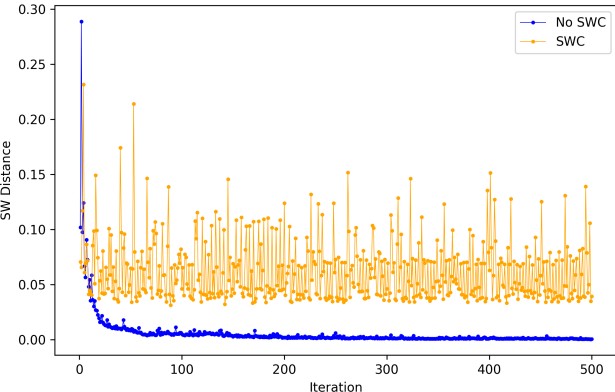

Figure 9: Comparison of Mallow's Weights and Gradient Clipping under logistic model.

### E.4 LIMITATIONS

While our method is designed to enable privacy-preserving streaming Bayesian Optimization (BO), there are some inherent limitations that must be considered. First, at very small privacy budgets $\varepsilon$, we observe a risk of utility collapse: model accuracy can deteriorate as privacy protection becomes increasingly stringent. This phenomenon is well documented in privacy-preserving machine learning and highlights the need for careful calibration of the privacy budget. Second, while SWC effectively controls kernel growth, it may introduce bias through the choice of projection directions used in the compression step. Such bias can obscure fine-grained structure in the data distribution, particularly in highly structured or multimodal settings. We plan to explore further refinements in future work.

### E.5 FUTURE WORK

**Federated learning.** For completeness, we also outline how the LDP-BO update naturally extends to federated learning (FL). Consider $N$ clients, where client $j$ holds i.i.d. samples from $\mathcal{P}_j$. The central server aims to solve

$$\boldsymbol{\theta}^\star = \operatorname{argmin}_{\boldsymbol{\theta} \in \boldsymbol{\Theta}} \left( f(\boldsymbol{\theta}) := \sum_{j=1}^N p_j E_{\boldsymbol{z_j} \sim \mathcal{P}_j} \left[ \mathcal{L}_j(\boldsymbol{\theta}, \boldsymbol{z}_j) \right] \right),$$

where $p_j$ is the weight of the $j$th client and $\mathcal{L}_j(\cdot, \boldsymbol{z}_j)$ is the loss function. At time point $t \geq 1$, each client performs a locally private update using a noisy BO-based gradient: $\boldsymbol{\theta}_t^j = \boldsymbol{\theta}_{t-1}^j - \eta_t g_{t-1}(\boldsymbol{\theta}_{t-1}^j) + \eta_t \boldsymbol{\omega}_t^j$, where $\boldsymbol{\omega}_t^j$ is properly calibrated LDP noise, and the BO gradient approximation is

$$g_{t-1}(\boldsymbol{\theta}_{t-1}) = \boldsymbol{\mu}_{\mathcal{D}_{t-1}} = \nabla K(\boldsymbol{\theta}_{t-1}, \mathcal{D}_{t-1}) K(\mathcal{D}_{t-1}, \mathcal{D}_{t-1})^{-1} \mathcal{L}(\boldsymbol{\theta}_{t-1}, \boldsymbol{z}_t),$$

with $\boldsymbol{\mu}_{\mathcal{D}_{t-1}}$ representing the posterior expectation given $\mathcal{D}_{t-1}$. The central server aggregates the local updates $\boldsymbol{\theta}_{t+1} = \sum_{j=1}^N p_j \boldsymbol{\theta}_t^j$, broadcasts $\boldsymbol{\theta}_{t+1}$ to all clients, and repeats for $\bar{\boldsymbol{\theta}}_t^j$ rounds, yielding the final estimator $\bar{\boldsymbol{\theta}}_T$. The detailed theoretical analysis will be left for our future research.

**Reinforcement learning.** Our framework can naturally extend to reinforcement learning (RL) by applying LDP-BO to optimize the expected return $J(\boldsymbol{\theta})$ of a policy $\pi_{\boldsymbol{\theta}}$. The BO loop operates over policy parameters, while local differential privacy is enforced on the observed returns.

- Local privatization of returns. At iteration $t$, the algorithm selects $\boldsymbol{\theta}_t$, runs episodes under $\pi_{\boldsymbol{\theta}_t}$, and locally privatizes the resulting return ($r_t$):

$$r_t = J(\boldsymbol{\theta}_t) + \omega_t,$$

where $\omega_t^j$ is properly calibrated LDP noise. Since only a scalar reward is privatized, sensitivity follows directly from standard bounded-reward assumptions in RL, and $\sigma^2$ is calibrated accordingly.

- BO surrogate update with SWC. The privatized return $r_t$ is incorporated into the BO surrogate. The privatized return is added to the kernel surrogate, and SWC maintains a compact dictionary,

$$\mathcal{D}_t = \text{SWC}(\mathcal{D}_{t-1}, \boldsymbol{\theta}_t).$$

ensuring the model size does not grow with time and enabling continual RL operation.

- Acquisition step. The next policy parameter is chosen by minimizing the Gaussian information (GI) acquisition rule:

$$\boldsymbol{\theta}_{t+1} = \arg\min_{\boldsymbol{\theta}} \text{GI}(\boldsymbol{\theta}; \mathcal{D}_t, \boldsymbol{\theta}_t) = \arg\min_{\boldsymbol{\theta}} \text{Tr}(\nabla^2 K_{\mathcal{D}_t \cup \boldsymbol{\theta}}(\boldsymbol{\theta}_t, \boldsymbol{\theta}_t))$$

yielding a fully online, privacy-preserving BO loop for policy search, where $K_{\mathcal{D}_t \cup \boldsymbol{\theta}}$ represents posterior covariance given $\mathcal{D}_t \cup \boldsymbol{\theta}$.

A promising direction for future work is to analyze how LDP noise affects the exploration–exploitation trade-off, building on BO-based RL approaches such as (Wilson et al., 2014; Balakrishnan et al., 2020; Müller et al., 2021) Wilson et al. (2014), Balakrishnan et al. (2020), and Müller et al. (2021).

## F  ALL TECHNIQUE PROOFS

*Proof of Theorem 3.1.* Consider two neighboring data points $\boldsymbol{z}_t$ and $\boldsymbol{z}_t'$ for $t \geq 1$, differing in exactly one entry, i.e., $d_H(\boldsymbol{z}_t, \boldsymbol{z}_t') = 1$. Recall that

$$\boldsymbol{\mu}_{t-1} = \nabla K(\boldsymbol{\theta}_{t-1}, \mathcal{D}) K(\mathcal{D}, \mathcal{D})^{-1} \mathcal{L}(\mathcal{D}, \boldsymbol{z}_t),$$
$$\tilde{\boldsymbol{\mu}}_{t-1} = \nabla K(\boldsymbol{\theta}_{t-1}, \mathcal{D}) K(\mathcal{D}, \mathcal{D})^{-1} \mathcal{L}(\mathcal{D}, \boldsymbol{z}_t').$$

and

$$g_t = \boldsymbol{\mu}_{t-1} \cdot \min\left\{1, \frac{B}{\|\boldsymbol{\mu}_{t-1}\|}\right\}, \tilde{g}_t = \tilde{\boldsymbol{\mu}}_{t-1} \cdot \min\left\{1, \frac{B}{\|\tilde{\boldsymbol{\mu}}_{t-1}\|}\right\}.$$

It follows that the global sensitivity of the estimated gradient at time $t$ is

$$\|g_t - \tilde{g}_t\| = \left\| \boldsymbol{\mu}_{t-1} \cdot \min\left\{1, \frac{B}{\|\boldsymbol{\mu}_{t-1}\|}\right\} - \tilde{\boldsymbol{\mu}}_{t-1} \cdot \min\left\{1, \frac{B}{\|\tilde{\boldsymbol{\mu}}_{t-1}\|}\right\} \right\|$$
$$\leq \left( \left\| \boldsymbol{\mu}_{t-1} \cdot \min\left\{1, \frac{B}{\|\boldsymbol{\mu}_{t-1}\|}\right\} \right\| + \left\| \tilde{\boldsymbol{\mu}}_{t-1} \cdot \min\left\{1, \frac{B}{\|\tilde{\boldsymbol{\mu}}_{t-1}\|}\right\} \right\| \right)$$
$$\leq B + B = 2B.$$

Hence, by adding noise sampled from $\mathcal{N}\left(0, 2(2B/\varepsilon_t)^2 \log(1.25/\delta_t)\mathbf{I}_p\right)$ at each iteration, the gradient update is guaranteed to satisfy $(\varepsilon_t, \delta_t)$-LDP. Moreover, by the parallel composition property of DP, the cumulative output $\tilde{\boldsymbol{\theta}}_t$ produced by Algorithm 1 satisfies $(\max\{\varepsilon_1, \ldots, \varepsilon_t\}, \max\{\delta_1, \ldots, \delta_t\})$-LDP.

Without loss of generality, we assume that the first iteration of Algorithm 1 satisfies $(\varepsilon_1, \delta_1)$-LDP. Since the initial estimate $\hat{\boldsymbol{\theta}}_0$ is deterministic, it follows directly that $\hat{\boldsymbol{\theta}}_1$ also satisfies $(\varepsilon_1, \delta_1)$-LDP. At the second iteration, $\hat{\boldsymbol{\theta}}_2$, depends on both the privatized output $\hat{\boldsymbol{\theta}}_1$ and the disjoint sample $\boldsymbol{z}_2$. It follows from Proposition A.4 that the two-fold composed algorithm $(\hat{\boldsymbol{\theta}}_1, \hat{\boldsymbol{\theta}}_2)$ satisfies $(\max\{\varepsilon_1, \varepsilon_2\}, \max\{\delta_1, \delta_2\})$-LDP when the samples $\boldsymbol{z}_1$ and $\boldsymbol{z}_2$ are disjoint. Iteratively applying this argument, we conclude that after $t$ iterations the entire sequence of updates satisfies $(\max\{\varepsilon_1, \ldots, \varepsilon_t\}, \max\{\delta_1, \ldots, \delta_t\})$-LDP. By the post-processing property, both $\hat{\boldsymbol{\theta}}_t$ and its averaged version $\tilde{\boldsymbol{\theta}}_t$ inherit the same privacy guarantees. □

*Proof of Theorem 3.3.* Our proof builds upon the framework of Koppel et al. (2021), which depends on the Hellinger distance, but here we adapt the analysis to the Sliced Wasserstein distance. Let $\rho_{\mathcal{D}_t}$ denote the posterior distribution at iteration $t$, where $\mathcal{D}_t$ is a dictionary of size $M_t$. When a new sample $\boldsymbol{\theta}_t$ is incorporated at iteration $t+1$, the dictionary is augmented to $\tilde{\mathcal{D}}_{t+1} = [\mathcal{D}_t; \boldsymbol{\theta}_t]$, increasing its size to $M_t + 1$. The stopping criterion for Algorithm 2 is violated whenever

$$\min_{j=1,\ldots,M_t+1} \eta_j \leq \kappa. \tag{7}$$

Notice that (7) provides a lower bound on the approximation error $\eta_{M_t+1}$ incurred by removing the newly added point $\boldsymbol{\theta}_t$. In particular, if $\eta_{M_t+1} \leq \kappa$, then the criterion in (7) is satisfied, and the model order remains unchanged. Consequently, $\eta_{M_t+1}$ can serve as a proxy for $\eta_j$ for all $j = 1, \ldots, M_t+1$.

For the case of the Sliced Wasserstein distance between multivariate Gaussian distributions, the approximation error $\eta_{M_t+1}$ depends only on the changes in the mean vector and covariance matrix induced by incorporating the new sample. $\boldsymbol{\theta}_t$. Specifically,

$$\eta_{M_t+1} \propto \left( \boldsymbol{\mu}_{t+1}|_{\mathcal{D}_t} - \boldsymbol{\mu}_{\mathcal{D}_t}, \ \boldsymbol{\Sigma}_{t+1}|_{\mathcal{D}_t} - \boldsymbol{\Sigma}_{\mathcal{D}_t} \right),$$

where $\boldsymbol{\mu}_{t+1}|_{\mathcal{D}_t}$ and $\boldsymbol{\Sigma}_{t+1}|_{\mathcal{D}_t}$ denote the mean and covariance conditioned on the dictionary $\mathcal{D}_t$, respectively, and $\boldsymbol{\mu}_{\mathcal{D}_t}, \boldsymbol{\Sigma}_{\mathcal{D}_t}$ are the corresponding quantities without $\boldsymbol{\theta}_t$.

Although there is no closed-form expression directly linking these mean and covariance differences to the Sliced Wasserstein distance, one can interpret the problem geometrically in terms of the Hilbert subspace defined by the current dictionary, $\mathcal{H}_{\mathcal{D}_t} := \mathrm{span}\{K(\mathcal{D}_j, \cdot)\}_{j=1}^{M_t}$. In particular, the approximation quality is governed by the distance between the kernel evaluation at the new point $K(\boldsymbol{\theta}_t, \cdot)$ and the subspace $\mathcal{H}_{\mathcal{D}_t}$. Intuitively, if this distance is small, the new point contributes little additional information and can be safely excluded without degrading the fidelity of the surrogate model, thereby satisfying the compression criterion. The approximation quality is then determined by the distance from the kernel evaluation at the new point to the current dictionary's Hilbert subspace:

$$\mathrm{dist}\left(K(\boldsymbol{\theta}_t, \cdot), \mathcal{H}_{\mathcal{D}_t}\right) := \min_{\mathbf{v} \in \mathbb{R}^{M_t}} \left\| K(\boldsymbol{\theta}_t, \cdot) - \mathbf{v}^\top \boldsymbol{\nu}_{\mathcal{D}_t}(\cdot) \right\|_{\mathcal{H}},$$

where $\mathcal{H}_{\mathcal{D}_t} := \mathrm{span}\{K(\mathcal{D}_j, \cdot)\}_{j=1}^{M_t}$ denotes the subspace spanned by the kernel functions in the current dictionary.

Therefore, if there exists some constant $c > 0$ such that $\mathrm{dist}(K(\boldsymbol{\theta}_t, \cdot), \mathcal{H}_{\mathcal{D}_t}) \leq c$, then there exists some $\kappa > 0$ for which $\eta_{M_t+1} \leq \kappa$. This ensures that the approximation error remains sufficiently small, and hence the model order does not increase. Since $\boldsymbol{\theta}$ lies in a compact set and $K$ is continuous, the range of the kernel embedding $\phi(\boldsymbol{\theta}) := K(\boldsymbol{\theta}, \cdot)$ is compact (Engel et al., 2004). Consequently, the number of balls of radius $c$ required to cover $\phi(\boldsymbol{\theta})$ is finite and determined by the covering number of $\phi(\boldsymbol{\theta})$ at scale $c$ (Anthony & Bartlett, 2009).

In particular, there exists a finite constant $M^\infty$ such that, if $M_t = M^\infty$, then $\mathrm{dist}(K(\boldsymbol{\theta}_t, \cdot), \mathcal{H}_{\mathcal{D}_t}) \leq c$, and consequently $\eta_{M_t+1} \leq \kappa$. Therefore, $M_t \leq M^\infty$ for all $t$. As shown by Engel et al. (2004), for a Lipschitz continuous Mercer kernel defined on a compact domain $\boldsymbol{\theta} \subset \mathbb{R}^p$, the covering number satisfies

$$M \leq \mathcal{O}\left(\frac{1}{\kappa}\right)^p.$$

We have completed the proof of this theorem. $\qquad\square$

*Proof of Theorem 4.4.* Recall that

$$\hat{\boldsymbol{\theta}}_t = \hat{\boldsymbol{\theta}}_{t-1} - \eta_t \left( g_{t-1}(\hat{\boldsymbol{\theta}}_{t-1}) + \omega_t \right).$$

Define the shifted functions

$$\tilde{g}_{t-1}(\boldsymbol{\Delta}) = g_{t-1}(\boldsymbol{\Delta} + \boldsymbol{\theta}^\star), \quad \tilde{g}(\boldsymbol{\Delta}) = g(\boldsymbol{\Delta} + \boldsymbol{\theta}^\star), \quad \tilde{f}(\boldsymbol{\Delta}) = f(\boldsymbol{\Delta} + \boldsymbol{\theta}^\star),$$

which correspond to a change of variables centered at the true parameter $\boldsymbol{\theta}^\star$. We then have

$$\begin{aligned}
\hat{\boldsymbol{\Delta}}_t &= \hat{\boldsymbol{\Delta}}_{t-1} - \eta_t g_{t-1}(\hat{\boldsymbol{\theta}}_{t-1}) + \eta_t \omega_t \\
&= \hat{\boldsymbol{\Delta}}_{t-1} - \eta_t \nabla \tilde{f}(\hat{\boldsymbol{\Delta}}_{t-1}) + \eta_t \{ \nabla \tilde{f}(\hat{\boldsymbol{\Delta}}_{t-1}) - \tilde{g}(\hat{\boldsymbol{\Delta}}_{t-1}) \} \\
&\quad + \eta_t \{ \tilde{g}(\hat{\boldsymbol{\Delta}}_{t-1}) - \tilde{g}_{t-1}(\hat{\boldsymbol{\Delta}}_{t-1}) \} + \eta_t \omega_t \\
&= \hat{\boldsymbol{\Delta}}_{t-1} - \eta_t \nabla \tilde{f}(\hat{\boldsymbol{\Delta}}_{t-1}) + \eta_t \xi_{1t} + \eta_t \xi_{2t} + \eta_t \omega_t,
\end{aligned}$$

where $\xi_{1t} = \nabla \tilde{f}(\hat{\boldsymbol{\Delta}}_{t-1}) - \tilde{g}(\hat{\boldsymbol{\Delta}}_{t-1}), \quad \xi_{2t} = \tilde{g}(\hat{\boldsymbol{\Delta}}_{t-1}) - \tilde{g}_{t-1}(\hat{\boldsymbol{\Delta}}_{t-1})$.

Therefore,

$$
\begin{aligned}
\|\hat{\boldsymbol{\Delta}}_t\|_2^2 =& \|\hat{\boldsymbol{\Delta}}_{t-1}\|_2^2 - 2\eta_t \left\langle \hat{\boldsymbol{\Delta}}_{t-1}, \nabla \tilde{f}(\hat{\boldsymbol{\Delta}}_{t-1}) - \xi_{1t} - \xi_{2t} - \omega_t \right\rangle \\
&+ \eta_t^2 \left\| \nabla \tilde{f}(\hat{\boldsymbol{\Delta}}_{t-1}) - \xi_{1t} - \xi_{2t} - \omega_t \right\|_2^2.
\end{aligned} \tag{8}
$$

Notice that $E[\omega_t] = 0$, the expectation of gradient estimate $\nabla f(\hat{\boldsymbol{\theta}}_{t-1})$ is $g(\hat{\boldsymbol{\theta}}_{t-1})$, and $g_{t-1}(\hat{\boldsymbol{\theta}}_{t-1}) - g(\hat{\boldsymbol{\theta}}_{t-1})$ is a transformation of the martingale difference sequence $\nabla \mathcal{L}(\hat{\boldsymbol{\theta}}_{t-1}, \boldsymbol{z}_t) - \nabla f(\hat{\boldsymbol{\theta}}_{t-1})$. This implies that

$$
E\left[ \left\langle \hat{\boldsymbol{\Delta}}_{t-1}, \xi_{1t} + \xi_{2t} + \omega_t \right\rangle \right] = 0.
$$

Meanwhile, applying Lemma C.6(i) to the pair $(\boldsymbol{\theta}^\star, \hat{\boldsymbol{\theta}}_{t-1})$, we obtain

$$
\langle \nabla \tilde{f}(\hat{\boldsymbol{\Delta}}_{t-1}), \hat{\boldsymbol{\Delta}}_{t-1} \rangle \geq \tilde{f}(\hat{\boldsymbol{\Delta}}_{t-1}) + \frac{\lambda}{2}\|\hat{\boldsymbol{\Delta}}_{t-1}\|_2^2 \geq \frac{\lambda}{2}\|\hat{\boldsymbol{\Delta}}_{t-1}\|_2^2.
$$

Using the upper equations above, we obtain

$$
E\{2\eta_t \langle \hat{\boldsymbol{\Delta}}_{t-1}, \nabla \tilde{f}(\hat{\boldsymbol{\Delta}}_{t-1}) - \xi_{1t} - \xi_{2t} - \omega_t \rangle\} \geq \frac{\lambda}{2}\|\hat{\boldsymbol{\Delta}}_{t-1}\|_2^2. \tag{9}
$$

Applying Lemma C.6(ii) to the pair $(\boldsymbol{\theta}^\star, \hat{\boldsymbol{\theta}}_{t-1})$, we obtain the gradient norm bound $\|\nabla \tilde{f}(\hat{\boldsymbol{\Delta}}_{t-1})\|_2 \leq \zeta\|\hat{\boldsymbol{\Delta}}_{t-1}\|_2$. In addition, Lemma C.4 and Lemma C.5 jointly provide explicit upper bounds on the second moments of the stochastic error terms: $E(\|\xi_{1t}\|_2^2) \leq c_1(L + p\kappa)$ and $E(\|\xi_{2t}\|_2^2) \leq 2B^2$.

Using Young's inequality, we then have

$$
\begin{aligned}
& E\{\|\nabla f(\hat{\boldsymbol{\Delta}}_{t-1}) - \xi_{1t} - \xi_{2t} - \omega_t\|_2^2\} \\
\leq & 4\|\nabla f(\hat{\boldsymbol{\Delta}}_{t-1})\|_2^2 + 4E(\|\xi_{1t}\|_2^2) + 4E(\|\xi_{2t}\|_2^2) + 4E\|\omega_t\|_2^2 \\
\leq & 4\zeta^2\|\hat{\boldsymbol{\Delta}}_{t-1}\|_2^2 + 8B^2 + 4c_1(L + p\kappa) + 32pB^2/\varepsilon^2 \log(1.25/\delta).
\end{aligned} \tag{10}
$$

Replacing the appropriate terms in (8) with (9) and (10), we have

$$
E(\|\hat{\boldsymbol{\Delta}}_t\|_2^2) \leq (1 - \lambda\eta_t + c'\eta_t^2)\|\hat{\boldsymbol{\Delta}}_{t-1}\|_2^2 + cp\eta_t^2 B^2/\varepsilon^2 \log(1.25/\delta) + 4\eta_t^2(c_1(L + p\kappa) + 2B^2).
$$

Therefore, there exists some positive constant $a_p$ depending on the dimension $p$ such that

$$
E(\|\hat{\boldsymbol{\Delta}}_t\|_2^2) \leq (1 - \lambda\eta_t + a_p^2\eta_t^2)\|\hat{\boldsymbol{\Delta}}_{t-1}\|_2^2 + a_p\eta_t^2 B^2/\varepsilon^2 \log(1.25/\delta) + 4\eta_t^2(c_1(L + p\kappa) + 2B^2),
$$

Define $t_0 = \min\{t : \lambda \geq 2a_p^2\eta_t, \lambda\eta_t t \geq 8\alpha \log t\}$. Then, for any $t \geq t_0$ and some constant $b_p = O(a_p)$, the equation simplifies to

$$
E(\|\hat{\boldsymbol{\Delta}}_t\|_2^2) \leq (1 - \lambda\eta_t/2)\|\hat{\boldsymbol{\Delta}}_{t-1}\|_2^2 + b_p\eta_t^2 B^2/\varepsilon^2 \log(1.25/\delta) + 4\eta_t^2(c_1(L + p\kappa) + 2B^2),
$$

Note that $\exp(-t\lambda\eta_t/4) \leq \exp(-\lambda\eta t^{1-\alpha}/4) \leq t^{-2\alpha} \leq t^{-\alpha}$ for $t \geq 2t_0$. Therefore, using the same arguments as in Chen et al. (2020), for $t \geq 2t_0$, we have

$$
\begin{aligned}
E(\|\hat{\boldsymbol{\Delta}}_t\|_2^2) \leq & \exp(-t\lambda\eta_t/4)E\|\hat{\boldsymbol{\Delta}}_{t/2}\|_2^2 + 2b_p\eta_{t/2}B^2 \log(1.25/\delta)/(\lambda\varepsilon^2) + 8\eta_{t/2}^2(c_1(L + p\kappa) + 2B^2) \\
\leq & \exp(-t\lambda\eta_t/4)(E\|\hat{\boldsymbol{\Delta}}_{n_0}\|_2^2 + 2b_p\eta_{n_0}B^2 \log(1.25/\delta)/(\lambda\varepsilon^2) \\
& + 8\eta_{n_0}(c_1(L + p\kappa) + 2B^2)/\lambda) + 2b_p\eta(t/2)^{-\alpha}B^2 \log(1.25/\delta)/(\lambda\varepsilon^2) \\
& + 8\eta(t/2)^{-\alpha}(c_1(L + p\kappa) + 2B^2)/\lambda \\
\leq & \exp(-t\lambda\eta_t/4)\{c(1 + \|\hat{\boldsymbol{\Delta}}_0\|_2^2) + 2b_p\eta_{n_0}B^2 \log(1.25/\delta)/(\lambda\varepsilon^2) \\
& + 8\eta_{n_0}(c_1(L + p\kappa) + 2B^2)/\lambda\} + 2b_p\eta(t/2)^{-\alpha}B^2 \log(1.25/\delta)/(\lambda\varepsilon^2) \\
& + 8\eta(t/2)^{-\alpha}(c_1(L + p\kappa) + 2B^2)/\lambda \\
\leq & c't^{-\alpha}\{\|\hat{\boldsymbol{\Delta}}_0\|_2^2 + c''b_p\eta B^2 \log(1.25/\Delta)/(\lambda\varepsilon^2) + \eta(L + p\kappa + 2B^2)/\lambda\}.
\end{aligned}
$$

$\square$

*Proof of Theorem 4.5.* Recall that $\hat{\boldsymbol{\theta}}_t = \hat{\boldsymbol{\theta}}_{t-1} - \eta_t(g_{t-1}(\hat{\boldsymbol{\theta}}_{t-1}) + \omega_t)$. By Assumption 4.3, we have

$$f(\hat{\boldsymbol{\theta}}_t) \leq f(\hat{\boldsymbol{\theta}}_{t-1}) + \langle \nabla f(\hat{\boldsymbol{\theta}}_{t-1}), \hat{\boldsymbol{\theta}}_t - \hat{\boldsymbol{\theta}}_{t-1} \rangle + \frac{\zeta}{2}\|\hat{\boldsymbol{\theta}}_t - \hat{\boldsymbol{\theta}}_{t-1}\|^2.$$

Thus, substituting the step sizes, we obtain

$$
\begin{aligned}
f(\hat{\boldsymbol{\theta}}_t) &\leq f(\hat{\boldsymbol{\theta}}_{t-1}) - \eta_t\langle \nabla f(\hat{\boldsymbol{\theta}}_{t-1}), g_{t-1}(\hat{\boldsymbol{\theta}}_{t-1}) + \omega_t \rangle + \frac{\zeta\eta_t^2}{2}\|g_{t-1}(\hat{\boldsymbol{\theta}}_{t-1}) + \omega_t\|^2 \\
&= f(\hat{\boldsymbol{\theta}}_{t-1}) - \eta_t\langle \nabla f(\hat{\boldsymbol{\theta}}_{t-1}), g_{t-1}(\hat{\boldsymbol{\theta}}_{t-1})\rangle - \eta_t\langle \nabla f(\hat{\boldsymbol{\theta}}_{t-1}), \omega_t\rangle \\
&\quad + \frac{\zeta\eta_t^2}{2}\left(\|g_{t-1}(\hat{\boldsymbol{\theta}}_{t-1})\|^2 + \|\omega_t\|^2 + 2\langle g_{t-1}(\hat{\boldsymbol{\theta}}_{t-1}), \omega_t\rangle\right) \\
&\leq f(\hat{\boldsymbol{\theta}}_{t-1}) - \eta_t\langle \nabla f(\hat{\boldsymbol{\theta}}_{t-1}), g_{t-1}(\hat{\boldsymbol{\theta}}_{t-1}) - \nabla f(\hat{\boldsymbol{\theta}}_{t-1}) + \nabla f(\hat{\boldsymbol{\theta}}_{t-1})\rangle - \eta_t\langle \nabla f(\hat{\boldsymbol{\theta}}_{t-1}), \omega_t\rangle \\
&\quad + \frac{\zeta\eta_t^2}{2}\left(\|g_{t-1}(\hat{\boldsymbol{\theta}}_{t-1})\|^2 + 8pB^2/\varepsilon^2\log(1.25/\delta) + 2\langle g_{t-1}(\hat{\boldsymbol{\theta}}_{t-1}), \omega_t\rangle)\right) \\
&\leq f(\hat{\boldsymbol{\theta}}_{t-1}) - \eta_t\langle \nabla f(\hat{\boldsymbol{\theta}}_{t-1}), g_{t-1}(\hat{\boldsymbol{\theta}}_{t-1}) - \nabla f(\hat{\boldsymbol{\theta}}_{t-1})\rangle - \eta_t\|\nabla f(\hat{\boldsymbol{\theta}}_{t-1})\|^2 - \eta_t\langle \nabla f(\hat{\boldsymbol{\theta}}_{t-1}), \omega_t\rangle \\
&\quad + \frac{\zeta\eta_t^2}{2}\left(\|\nabla f(\hat{\boldsymbol{\theta}}_{t-1})\|^2 + \|g_{t-1}(\hat{\boldsymbol{\theta}}_{t-1}) - \nabla f(\hat{\boldsymbol{\theta}}_{t-1})\|^2 + 2\langle \nabla f(\hat{\boldsymbol{\theta}}_{t-1}), g_{t-1}(\hat{\boldsymbol{\theta}}_{t-1}) - \nabla f(\hat{\boldsymbol{\theta}}_{t-1})\rangle\right) \\
&\quad + \frac{\zeta\eta_t^2}{2}\left(8pB^2/\varepsilon^2\log(1.25/\delta) + 2\langle g_{t-1}(\hat{\boldsymbol{\theta}}_{t-1}), \omega_t\rangle)\right) \\
&\leq f(\hat{\boldsymbol{\theta}}_{t-1}) - \frac{\eta_t}{2}\|\nabla f(\hat{\boldsymbol{\theta}}_{t-1})\|^2 + \eta_t\langle \nabla g_{t-1}(\hat{\boldsymbol{\theta}}_{t-1}) - f(\hat{\boldsymbol{\theta}}_{t-1}), \omega_t\rangle \\
&\quad + \frac{\zeta\eta_t^2}{2}\left(\|g_{t-1}(\hat{\boldsymbol{\theta}}_{t-1}) - \nabla f(\hat{\boldsymbol{\theta}}_{t-1})\|^2 + 8pB^2/\varepsilon^2\log(1.25/\delta)\right),
\end{aligned}
$$

where the first inequality follows from $\zeta$-smoothness and the last inequality holds due to $\eta_t \leq \frac{1}{\zeta}$. The result is obtained by rearranging terms.

$$
\begin{aligned}
\frac{\eta_t}{2}\|\nabla f(\hat{\boldsymbol{\theta}}_{t-1})\|^2 &\leq f(\hat{\boldsymbol{\theta}}_{t-1}) - f(\hat{\boldsymbol{\theta}}_t) + \eta_t\langle g_{t-1}(\hat{\boldsymbol{\theta}}_{t-1}) - \nabla f(\hat{\boldsymbol{\theta}}_{t-1}), \omega_t\rangle \\
&\quad + \frac{\zeta\eta_t^2}{2}\left(\|g_{t-1}(\hat{\boldsymbol{\theta}}_{t-1}) - \nabla f(\hat{\boldsymbol{\theta}}_{t-1})\|^2 + 8pB^2/\varepsilon^2\log(1.25/\delta)\right).
\end{aligned}
$$

Summing the inequalities over $t = 1, \ldots, T$, we have

$$
\begin{aligned}
\sum_{t=1}^{T} \eta_t\|\nabla f(\hat{\boldsymbol{\theta}}_{t-1})\|^2 &\leq 2(f(\hat{\boldsymbol{\theta}}_0) - f(\hat{\boldsymbol{\theta}}_{T-1})) + \sum_{t=1}^{T} 2\eta_t\langle g_{t-1}(\hat{\boldsymbol{\theta}}_{t-1}) - \nabla f(\hat{\boldsymbol{\theta}}_{t-1}), \omega_t\rangle \\
&\quad + \sum_{t=1}^{T} 2\zeta\eta_t^2\left(\|g_{t-1}(\hat{\boldsymbol{\theta}}_{t-1}) - \nabla f(\hat{\boldsymbol{\theta}}_{t-1})\|^2 + 16pB^2/\varepsilon^2\log(1.25/\delta)\right) \\
&\leq 2(f(\hat{\boldsymbol{\theta}}_0) - f(\boldsymbol{\theta}^{\star})) + \sum_{t=1}^{T} 2\eta_t\langle g_{t-1}(\hat{\boldsymbol{\theta}}_{t-1}) - \nabla f(\hat{\boldsymbol{\theta}}_{t-1}), \omega_t\rangle \\
&\quad + \sum_{t=1}^{T} 2\zeta\eta_t^2\left(\|g_{t-1}(\hat{\boldsymbol{\theta}}_{t-1}) - \nabla f(\hat{\boldsymbol{\theta}}_{t-1})\|^2 + 16pB^2/\varepsilon^2\log(1.25/\delta)\right).
\end{aligned}
$$

$$(11)$$

Dividing both sides by $\sum_{t=1}^{T} \eta_t$ yields

$$
\begin{aligned}
\frac{\sum_{t=1}^{T} \eta_t\|\nabla f(\hat{\boldsymbol{\theta}}_{t-1})\|^2}{\sum_{t=1}^{T} \eta_t} &\leq \frac{2(f(\hat{\boldsymbol{\theta}}_0) - f(\boldsymbol{\theta}^{\star}))}{\sum_{t=1}^{T} \eta_t} + \frac{\sum_{t=1}^{T} 2\eta_t\langle g_{t-1}(\hat{\boldsymbol{\theta}}_{t-1}) - \nabla f(\hat{\boldsymbol{\theta}}_{t-1}), \omega_t\rangle}{\sum_{t=1}^{T} \eta_t} \\
&\quad + \frac{\sum_{t=1}^{T} 2\zeta\eta_t^2\left(\|g_{t-1}(\hat{\boldsymbol{\theta}}_{t-1}) - \nabla f(\hat{\boldsymbol{\theta}}_{t-1})\|^2 + 16pB^2/\varepsilon^2\log(1.25/\delta)\right)}{\sum_{t=1}^{T} \eta_t}.
\end{aligned}
$$

Note that $E(\omega_t) = 0$, the expectation of gradient estimate $\nabla f(\hat{\boldsymbol{\theta}}_{t-1})$ is $g(\hat{\boldsymbol{\theta}}_{t-1})$, and $g_{t-1}(\hat{\boldsymbol{\theta}}_{t-1}) - g(\hat{\boldsymbol{\theta}}_{t-1})$ is a transformation of the martingale difference sequence $\nabla \mathcal{L}(\hat{\boldsymbol{\theta}}_{t-1}, \boldsymbol{z}_t) - \nabla f(\hat{\boldsymbol{\theta}}_{t-1})$, im-

plying

$$E(\langle g_{t-1}(\hat{\boldsymbol{\theta}}_{t-1}) - \nabla f(\hat{\boldsymbol{\theta}}_{t-1}), \omega_t \rangle) = 0.$$

Furthermore,

$$\|g_{t-1}(\hat{\boldsymbol{\theta}}_{t-1}) - \nabla f(\hat{\boldsymbol{\theta}}_{t-1})\|^2 \le \|g_{t-1}(\hat{\boldsymbol{\theta}}_{t-1}) - g(\hat{\boldsymbol{\theta}}_{t-1})\|^2 + \|g(\hat{\boldsymbol{\theta}}_{t-1}) - \nabla f(\hat{\boldsymbol{\theta}}_{t-1})\|^2$$
$$\le 2B^2 + c_1(L + p\kappa).$$

Taking the expectation with respect to these terms and substituting into (11), we obtain

$$\frac{\sum_{t=1}^T \eta_t E\|\nabla f(\hat{\boldsymbol{\theta}}_{t-1})\|^2}{\sum_{t=1}^T \eta_t} \le \frac{2(f(\hat{\boldsymbol{\theta}}_0) - f(\boldsymbol{\theta}^\star))}{\sum_{t=1}^T \eta_t}$$
$$+ \frac{\sum_{t=1}^T 2\zeta \eta_t^2 \left((c_1(L+p\kappa) + 2B^2) + 16pB^2/\varepsilon^2 \log(1.25/\delta)\right)}{\sum_{t=1}^T \eta_t}.$$

We then obtain

$$\min_{1 \le t \le T} E\|\nabla f(\hat{\boldsymbol{\theta}}_{t-1})\|^2 \le \frac{2(f(\hat{\boldsymbol{\theta}}_0) - f(\boldsymbol{\theta}^\star))}{\sum_{t=1}^T \eta_t}$$
$$+ \frac{\sum_{t=1}^T 2\zeta \eta_t^2 \left((c_1(L+p\kappa) + 2B^2) + 16pB^2/\varepsilon^2 \log(1.25/\delta)\right)}{\sum_{t=1}^T \eta_t}.$$

Recall that $\eta_t = \eta_0 t^{-\alpha}$. Following the integral bounding technique in Garrigos & Gower (2023), there exist constants $c_2$ and $c_3$ such that $\sum_{t=1}^T \eta_t = \eta_0 \sum_{t=1}^T t^{-\alpha} \le c_2 T^{1-\alpha}$ and $\sum_{t=1}^T \eta_t^2 = \eta_0 \sum_{t=1}^T t^{-2\alpha} \le c_3$. Therefore, the inequality simplifies to

$$\min_{1 \le t \le T} E\|\nabla f(\hat{\boldsymbol{\theta}}_{t-1})\|^2 \le c' \frac{(f(\hat{\boldsymbol{\theta}}_0) - f(\boldsymbol{\theta}^\star)) + \zeta((L+p\kappa) + B^2) + pB^2/\varepsilon^2 \log(1.25/\delta)}{T^{1-\alpha}}.$$

$\square$

*Proof of Theorem 4.5.* For simplicity, denote event $\{\lim_{k\to\infty} \boldsymbol{\theta}_k = \boldsymbol{\theta}^{opt}\}$ by $S_{opt}$ and $\Delta_t \triangleq \boldsymbol{\theta}_t - \boldsymbol{\theta}^{opt}$. We have the following decomposition,

$$E\left(\|\Delta_T\|^2 \mathbf{1}_{S_{opt}}\right) = E\left(\|\Delta_T\|^2 \mathbf{1}_{S_{opt}} \mathbf{1}\left\{\exists \frac{T}{4} \le t \le \frac{T}{2}, \boldsymbol{\theta}_T \in R_{good}\right\}\right)$$
$$+ E\left(\|\Delta_T\|^2 \mathbf{1}_{S_{opt}} \mathbf{1}\left\{\forall \frac{T}{4} \le t \le \frac{T}{2}, \boldsymbol{\theta}_T \notin R_{good}\right\}\right)$$
$$\triangleq A + B.$$

$A$ can be further decomposed as follows,

$$A \le E\left(\|\Delta_T\|^2 \mathbf{1}_{S_{opt}} \mathbf{1}\left\{\exists \frac{T}{4} \le t \le \frac{T}{2}, \boldsymbol{\theta}_T \in R_{good}\left(\boldsymbol{\theta}^{opt}\right)\right\}\right)$$
$$+ E\left(\|\Delta_T\|^2 \mathbf{1}_{S_{opt}} \mathbf{1}\left\{\exists \frac{T}{4} \le t \le \frac{T}{2}, \boldsymbol{\theta}_T \in R_{good} \backslash R_{good}\left(\boldsymbol{\theta}^{opt}\right)\right\}\right)$$
$$\triangleq A_1 + A_2.$$

Next, we have

$$A_1 \le E\left(\|\Delta_T\|^2 \mathbf{1}\left\{\exists \frac{T}{4} \le t \le \frac{T}{2}, \boldsymbol{\theta}_n \in R_{good}^L\left(\boldsymbol{\theta}^{opt}\right) \text{ for all } n \ge t\right\}\right)$$
$$+ E\left(\|\Delta_T\|^2 \mathbf{1}\left\{\exists \frac{T}{4} \le t \le \frac{T}{2}, \boldsymbol{\theta}_t \in R_{good}\left(\boldsymbol{\theta}^{opt}\right) \text{ but } \boldsymbol{\theta}_n \notin R_{good}^L\left(\boldsymbol{\theta}^{opt}\right) \text{ for some } n \ge t\right\}\right)$$
$$\triangleq A_{11} + A_{12}.$$

Now, we are to show that $A_{11} = O(T^{-\alpha})$. For any $t \in \mathbb{Z}_+$, we have

$$\hat{\boldsymbol{\Delta}}_t = \hat{\boldsymbol{\Delta}}_{t-1} - \eta_t g_{t-1}(\hat{\boldsymbol{\theta}}_{t-1}) + \eta_t \omega_t$$
$$= \hat{\boldsymbol{\Delta}}_{t-1} - \eta_t \nabla \tilde{f}(\hat{\boldsymbol{\Delta}}_{t-1}) + \eta_t \xi_{1t} + \eta_t \xi_{2t} + \eta_t \omega_t,$$

where $\xi_{1t} = \nabla \tilde{f}(\hat{\boldsymbol{\Delta}}_{t-1}) - \tilde{g}(\hat{\boldsymbol{\Delta}}_{t-1}), \quad \xi_{2t} = \tilde{g}(\hat{\boldsymbol{\Delta}}_{t-1}) - \tilde{g}_{t-1}(\hat{\boldsymbol{\Delta}}_{t-1}).$

Recall proof of Theorem 4.4 and based on condition **??**, we know that on $\{\theta_{t-1} \in R_{good}^L(\theta^{opt})\}$,

$$\langle \Delta_{t-1}, \nabla f(\theta_{t-1}) \rangle \geq \frac{1}{2} \tilde{\lambda}_{min} \|\Delta_{t-1}\|^2.$$

Therefore, on $\{\theta_{t-1} \in R_{good}^L(\theta^{opt})\}$, we have

$$E(\|\hat{\boldsymbol{\Delta}}_t\|_2^2) \leq (1 - \tilde{\lambda}_{min}\eta_t + a_p^2\eta_t^2)\|\hat{\boldsymbol{\Delta}}_{t-1}\|_2^2 + a_p\eta_t^2 B^2/\varepsilon^2 \log(1.25/\delta) + 4\eta_t^2(c_1(L + p\kappa) + 2B^2),$$

where $a_p$ is some positive constant depending on the dimension $p$.

Define $t_0 = \min\{t : \tilde{\lambda}_{min} \geq 2a_p^2\eta_t, \tilde{\lambda}_{min}\eta_t t \geq 8\alpha \log t\}$. Then, for any $t \geq t_0$ and some constant $b_p = O(a_p)$, the equation simplifies to

$$E(\|\hat{\boldsymbol{\Delta}}_t\|_2^2) \leq (1 - \tilde{\lambda}_{min}\eta_t/2)\|\hat{\boldsymbol{\Delta}}_{t-1}\|_2^2 + b_p\eta_t^2 B^2/\varepsilon^2 \log(1.25/\delta) + 4\eta_t^2(c_1(L + p\kappa) + 2B^2).$$

For the sake of simplicity, we let $C_0 = b_p^2 B^2/\varepsilon^2 \log(1.25/\delta) + 4^2(c_1(L + p\kappa) + 2B^2)$. As a result, we have

$$E\left(\left\|\hat{\boldsymbol{\Delta}}_T\right\|^2 \mathbf{1}\left\{\theta_t \in R_{good}^L(\theta^{\mathrm{opt}}), \frac{T}{2} \leq t \leq T-1\right\}\right)$$

$$= E\left(\left(E\left\|\hat{\boldsymbol{\Delta}}_T\right\|^2\right) \mathbf{1}\left\{\theta_t \in R_{good}^L(\theta^{\mathrm{opt}}), \frac{T}{2} \leq t \leq T-1\right\}\right)$$

$$\leq \left(1 - \frac{1}{2}\tilde{\lambda}_{\min}\gamma_T\right) E\left(\left\|\hat{\boldsymbol{\Delta}}_{T-1}\right\|^2 \mathbf{1}\left\{\theta_t \in R_{good}^L(\theta^{\mathrm{opt}}), \frac{T}{2} \leq t \leq T-1\right\}\right) + C_0\gamma_T^2$$

$$\cdots$$

$$\leq \left(\prod_{t=\frac{T}{2}+1}^{T}\left(1 - \frac{1}{2}\tilde{\lambda}_{\min}\gamma_t\right)\right) E\left\|\hat{\boldsymbol{\Delta}}_{\frac{T}{2}}\right\|^2 + C_0 \sum_{t=\frac{T}{2}+1}^{T}\left(\gamma_t^2 \prod_{j=t+1}^{T}\left(1 - \frac{1}{2}\tilde{\lambda}_{\min}\gamma_j\right)\right)$$

$$\leq \exp\left(-\frac{1}{2}C\tilde{\lambda}_{\min} \sum_{t=\frac{T}{2}+1}^{T} t^{-\alpha}\right) E\left\|\hat{\boldsymbol{\Delta}}_{\frac{T}{2}}\right\|^2 + C_0 \sum_{t=\frac{T}{2}+1}^{T}\left(\gamma_t^2 \left(1 - \frac{1}{2}\tilde{\lambda}_{\min}\gamma_T\right)^{T-t}\right)$$

$$\leq \exp\left(-\frac{C\tilde{\lambda}_{\min}}{4}T^{1-\alpha}\right) E\left\|\hat{\boldsymbol{\Delta}}_{\frac{T}{2}}\right\|^2 + C_0\left(\frac{T}{2}\right)^{-2\alpha} \sum_{t=\frac{T}{2}+1}^{T}\left(1 - \frac{1}{2}\tilde{\lambda}_{\min}\gamma_T\right)^{T-t}$$

$$\leq \exp\left(-\frac{C\tilde{\lambda}_{\min}}{4}T^{1-\alpha}\right) E\left\|\hat{\boldsymbol{\Delta}}_{\frac{T}{2}}\right\|^2 + C_0\left(\frac{T}{2}\right)^{-2\alpha}\left(\frac{1}{2}\tilde{\lambda}_{\min}\gamma_T\right)^{-1}$$

$$= O(T^{-\alpha}).$$

where the last step is similar with proof of Theorem 4.4. Then, we can see that

$$A_{11} \leq E\left(\|\hat{\boldsymbol{\Delta}}_T\|^2 \mathbf{1}\{\boldsymbol{\theta}_t \in R_{good}^L(\boldsymbol{\theta}^{opt}), T/2 \leq t \leq T-1\}\right) = O(T^{-\alpha}). \tag{12}$$

Using the same arguments as in Zhong et al. (2023), we have

$$A_{12} \leq \left(E\|\hat{\boldsymbol{\Delta}}_T\|^3\right)^{\frac{2}{3}} P^{\frac{1}{3}}\left(\exists \frac{T}{4} \leq t \leq \frac{T}{2}, \boldsymbol{\theta}_t \in R_{good}(\boldsymbol{\theta}^{opt})\right.$$

$$\left. \text{but } \boldsymbol{\theta}_s \notin R_{good}^L(\boldsymbol{\theta}^{opt}) \text{ for some } s \geq t\right) \tag{13}$$

$$\leq \left(E\|\hat{\boldsymbol{\Delta}}_T\|^3\right)^{\frac{2}{3}} T^{-2\alpha}$$

$$= O(T^{-\alpha}).$$

Based on (12) and (13), we have

$$A_1 = O(T^{-\alpha}).$$

To show $A_2 = O(T^{-\alpha})$, we have

$$A_2 \le \left(E\|\hat{\boldsymbol{\Delta}}_T\|^3\right)^{\frac{2}{3}} P^{\frac{1}{3}} \left(S_{opt} \cap \left\{\exists\, T/4 \le t \le T/2,\, \boldsymbol{\theta}_t \in R_{good} \backslash R_{good}(\boldsymbol{\theta}^{opt})\right\}\right)$$

$$\le \left(E\|\hat{\boldsymbol{\Delta}}_T\|^3\right)^{\frac{2}{3}} P^{\frac{1}{3}} \left(\exists\, T/4 \le t \le T/2,\, \boldsymbol{\theta}' \in \Theta^{opt},\, \boldsymbol{\theta}_t \in R_{good}(\boldsymbol{\theta}') \text{ but } \boldsymbol{\theta}_s \notin R_{good}(\boldsymbol{\theta}') \text{ for some } s \ge t\right)$$

$$= O(T^{-\alpha}),$$

where the last step is similar to the 2nd step of (13). Therefore, we have

$$A = A_1 + A_2 = O(T^{-\alpha}).$$

$\square$

## G   THE USE OF LARGE LANGUAGE MODELS

In the preparation of this manuscript, we employed a large language model (LLM) to assist in the polishing and refinement of the writing. The model was used exclusively for improving linguistic expression, enhancing clarity, and ensuring consistency of terminology—tasks that contribute to the overall readability and academic tone of the document. All technical content, mathematical reasoning, and scientific conclusions remain entirely formulated by the authors. The use of LLM-assisted editing did not alter the theoretical contributions or empirical results presented in this work.

