# OpenReview forum: "Online Differential Privacy Bayesian Optimization with Sliced Wasserstein Compression"
_ICLR.cc/2026/Conference — Submitted to ICLR 2026_

### Official Review · Reviewer_84Py · 2025-10-27

**Soundness:** 3
**Presentation:** 3
**Contribution:** 3
**Rating:** 6
**Confidence:** 3

**Summary:**

This paper introduces an online locally differentially private (LDP) framework for Bayesian optimization (BO) tailored to streaming data environments. It proposes a zero-order optimizer using Gaussian processes (GPs) with a novel Sliced Wasserstein Compression (SWC) algorithm to bound kernel dictionary growth while maintaining numerical stability and privacy guarantees. The method embeds Gaussian noise into gradient approximations derived from the GP posterior, ensuring per-iteration LDP. Non-asymptotic convergence rates are provided for strongly convex and smooth (non-convex) losses, matching SGD-like performance without gradient access. Experiments on synthetic (linear, logistic, ReLU, Sine, Friedman) and real (Uber fares) datasets demonstrate superior accuracy and stability over LDP-SGD and non-private baselines, especially in nonlinear settings.

**Strengths:**

- Novelty: Fills a clear gap in online, gradient-free, LDP BO for streaming data. The SWC algorithm is innovative, using sliced Wasserstein distance to bound dictionary size $(O(1/\kappa)^p)$ while preserving posterior fidelity—extending batch private BO to dynamic, untrusted settings without historical data storage.

- Theoretical Rigor: Provides non-asymptotic convergence rates (Theorems 4.4–4.5) under decaying stepsizes, achieving SGD-optimal $O(t^{-\alpha})$ error in strongly convex cases and $O(t^{-(1-\alpha)})$ gradient norms in smooth non-convex settings. Bounds explicitly capture privacy $(\epsilon, \delta)$, compression $\kappa$, and BO approximation errors, with corollaries for RDP/GDP—stronger than heuristic privacy in prior online BO. Assumptions (e.g., smoothness, RKHS) are standard and justified.

- Practical Efficiency: $O(1)$ per-iteration time/space via SWC avoids $O(t^3)$ kernel inversions in standard BO. Algorithm 1 is clean and implementable; privacy via Gaussian mechanism is flexible (extendable to Laplace).

- Empirical Validation: Thorough experiments across parametric (linear/logistic/ReLU) and nonparametric (Sine/Friedman) synthetic tasks, plus real Uber fares data. LDP-BO outperforms LDP-SGD in MSE/prediction error, especially nonlinearly, with stable convergence under varying $\epsilon / \kappa$. Function fitting plots visually confirm better generalization vs. non-private DNN. Reproducibility statement and code promise are excellent.

- Clarity and Structure: Well-written with intuitive flowchart (Fig. 1), detailed appendices (e.g., DP preliminaries, proofs), and ethical/reproducibility statements. Related work comprehensively covers BO, online learning, and private estimation.

**Weaknesses:**

1. Assumption Strength: Bounded sensitivity (Ass. 4.1) and RKHS membership (Ass. 4.2) are common but may limit applicability to heavy-tailed or non-stationary streams (e.g., real-world sensor data). Strong convexity (Ass. 4.3) yields optimal rates, but non-convex analysis (Thm. 4.5) only guarantees stationarity—discuss implications for multimodal BO objectives more. Assumption 3.2 (non-expansive compression) is mild but relies on prior consistency (Lemma C.1); empirical verification would help.
2. Experimental Scope: Baselines are appropriate (LDP-SGD, non-private BO/SGD/DNN), but missing comparisons to other scalable GPs (e.g., inducing points in Balandat et al., 2020) or private online methods (e.g., DP contextual bandits from Ding et al., 2021). Real-data ablation on $\kappa$ sensitivity or varying $\delta$ is absent; Uber fares is compelling but single-domain—add multi-domain (e.g., finance, healthcare) for broader impact. No runtime/memory plots to quantify $O(1)$ claims vs baselines.
3. Privacy-Utility Trade-off: While theory bounds noise (e.g., $O(B^2 \log(1/\delta)/(\lambda \epsilon^2))$), experiments use fixed/varying $\epsilon$ but not composition over $t$ (e.g., total $\epsilon_t$ sum). Discuss practical budget allocation (e.g., decaying $\epsilon_t$) or advanced mechanisms (e.g., RDP for tighter composition). SWC privacy amplification isn't analyzed—does compression interact with LDP?
4. Minor Technical/ Presentation Issues: Acquisition function (GI in Eq. 3) optimization isn't detailed (e.g., how to solve argmin GI efficiently online?). Theorem 3.1 claims per-iteration LDP but composes to $\max{\epsilon_1,...,\epsilon_t}$-LDP—clarify if adaptive $\epsilon_t$ allows better total budget. Appendix E details are good but could include hyper parameter tuning (e.g., $\eta_t$ choice).

**Questions:**

1. SWC Mechanism (Sec. 3.2, Alg. 2): The use of sliced Wasserstein for compression is elegant and avoids density estimation pitfalls. However, computing $SW_2(\rho_{D_{-j}}, \rho_{\tilde{D}_t})$ for each $j$ in $I$ seems $O(M_t^2)$ per iteration (with $m=100$ projections)—how does this achieve true $O(1)$ amortized? Suggest approximating via subsampling directions or low-rank updates. Besides, if line 8 is not satisfied in the first iteration of the loop, does that mean it cannot obtain a compressed dictionary $D_t$, and is it necessary to dynamically adjust the size of $\kappa$? Also, Thm. 3.3 bounds $M_t ≤ O((1/\kappa)^p)$, but $p=5$ in expts: discuss curse-of-dimensionality for high-$p$ (e.g., $>10$).
2. Privacy Integration (Sec. 3.1, Thm. 3.1): Clipping to $B$ before Gaussian noise is standard, but sensitivity $| g_t - \tilde{g}_t | ≤ 2B$ assumes $| \mu |$ bounded—link to Ass. 4.1 more explicitly. Why Gaussian over Laplace (better for $l_1$-sens)? Corollaries B.1–B.2 are nice extensions; consider empirical comparison of RDP/GDP noise levels. Question: In streaming, how to handle adaptive adversaries querying outputs?
3. Theoretical Analysis (Sec. 4): Proofs are solid (e.g., leveraging projection non-expansiveness in Lem. C.3), but the BO error term $(L + p\kappa)$ assumes GI minimization over $p+1$ points (Wu et al., 2023)—is this feasible online? For non-convex (Thm. 4.5), the $O(1/T^{1-\alpha})$ rate is good, but compare quantitatively to non-private BO rates (e.g., Müller et al., 2021). Minor: $t_0$ definition in Thm. 4.4 could be explicit.
4. Experiments (Sec. 5): Strong results, but add ablation: (i) MSE vs $\kappa$ (e.g., $0.05–0.5$) on Uber data; (ii) total privacy budget $(\Sigma, \epsilon_t)$ impact; (iii) dictionary size evolution over $t$ to verify bounded $M_t$. Runtime: LDP-BO vs. LDP-SGD on $t=20k$? Fig. 2/5: Why focus on first coeff? Average over all or report full $| \theta |_2$ error.

5. Broader Impact/Ethics: Privacy in streaming BO enables sensitive apps (e.g., personalized medicine), but discuss failure modes (e.g., utility collapse at $\epsilon<<1$) or biases in SWC (e.g., projection directions). Reproducibility: Provide exact $\eta_t$ (e.g., $\alpha=0.75$?), kernel choice (RBF?), and seed for 100 reps.

---

> ### Author Response · Authors · 2025-11-21
> **Response of W1**
>
> We sincerely appreciate the reviewer's thoughtful comments. We have carefully addressed each concern below and respectfully ask for a reevaluation of the paper.
>
> - **W1: More clarifications on Assumptions.** This is an excellent comment. Although our theoretical analysis relies on the bounded-sensitivity assumption (Ass. 4.1), RKHS membership (Ass. 4.2), and strong convexity (Ass. 4.3), these assumptions are primarily technical. The proposed method remains applicable in heavy-tailed, non-stationary, and multimodal BO settings. We have added additional experiments demonstrating that the method continues to perform well even when these assumptions are violated in practice.
>    - **Non-stationary streams.** We have added experimental studies for non-stationary settings, focusing on parameter drift in the linear model of Example 5.1. These experiments use privacy parameters ($(\varepsilon,\delta)=(2,0.2)$) and a compression budget of $\kappa=0.1$ in $T = 20000$ samples. Following Barber et al. (2023), we consider two types of non-stationarity:
>        - **Case 1: Abrupt regime shifts.** The regression coefficient $\boldsymbol{\theta}$ switches among three fixed vectors over successive time segments: $$\theta^{(1)} = (1,2,1,0,0),\quad \theta^{(2)} = (0,-1,-2,-1,0),\quad \theta^{(3)} = (0,0,1,2,1),$$
>       - **Case 2: Smooth concept drift.** The regression coefficient evolves linearly from $$\theta^{(0)} = (1,2,1,0,0),\qquad \theta^{(1)} = (0,0,1,2,1),$$ according to $\theta_t = (1-\alpha_t)\theta^{(0)} + \alpha_t\\theta^{(1)}, \alpha_t = (t-1)/(T-1).$
>   Table  shows that LDP-BO consistently outperforms LDP-SGD in both cases, achieving lower prediction error and more stable performance under the same $(\varepsilon,\delta)$-LDP budget, and approaching the performance of the non-private baseline. The suboptimal result at 15,000 samples in Case 1 corresponds to the regime shift around 13,000 samples; with larger sample sizes, LDP-BO converges more rapidly than LDP-SGD.
> |Case|Privacy|$t$|LDP-BO|LDP-SGD|
> |----|-------------|---:|-------|-------|
> |Case 1|No DP|5,000|1.11 (0.58)|1.59 (0.72)|
> |||10,000|0.60 (0.31)|0.86 (0.39)|
> |||15,000|0.93 (0.48)|1.33 (0.60)|
> |||20,000|1.08 (0.56)|1.55 (0.70)|
> ||$\varepsilon = 2$|5,000|1.48 (0.76)|2.11 (0.95)|
> |||10,000|1.20 (0.62)|1.71 (0.77)|
> |||15,000|55.44 (28.51)|79.20 (35.64)|
> |||20,000|3.42 (1.76)|4.88 (2.20)|
> |Case 2|No DP|5,000|3.29 (1.70)|4.70 (2.12)|
> |||10,000|1.72 (0.89)|2.46 (1.11)|
> |||15,000|1.54 (0.79)|2.20 (0.99)|
> |||20,000|1.59 (0.82)|2.27 (1.02)|
> ||$\varepsilon = 2$|5,000|6.10 (3.14)|8.72 (3.92)|
> |||10,000|3.33 (1.71)|4.76 (2.14)|
> |||15,000|3.94 (2.02)|5.63 (2.53)|
> |||20,000|5.83 (3.00)|8.33 (3.75)|
>   - **Multimodal BO Objectives.** Assumption 4.3, which imposes global strong convexity, is commonly used in stochastic optimization to guarantee sharp convergence rates for parameter estimates (e.g., Vaswani et al., 2022; Zhu et al., 2023). While we acknowledge that this assumption is strong, our analysis extends beyond it by considering objectives with multiple global (and local) minima. In this non-convex, multimodal setting, we must control the distance of iterates to the entire set of optimal solutions while maintaining the same convergence rate as in the strongly convex case.
> To handle this multimodal, non-convex regime, we impose mild structural conditions around the global minimizers (Assumptions B.3--B.5, page 18), ensuring smoothness and local strong convexity near each minimum. Under these weaker conditions, Corollary 5 (page 18) shows that estimates $\hat{\boldsymbol{\theta}}_t$ still converge to the global minimizer set $\boldsymbol{\Theta}^{\mathrm{opt}}$ at the rate $O(t^{-\alpha})$, matching the strongly convex case. This demonstrates that our method retains desirable convergence properties even for multimodal, non-convex objectives. The detailed proof is provided in Appendix B (pages 38--40).
>   - **Empirical verification Assumption 3.2.** Assumption 3.2 is a mild assumption, relying on a consistency property formalized in Lemma C.1. This consistency and non-expansive projection assumption is standard in the online Gaussian process regression and nonparametric Bayesian regression literature (e.g., Choi et al., 2007; Koppel et al., 2021). To empirically validate Assumption 3.2, we performed an ablation study comparing LDP-BO with and without SWC using the linear regression model from Example 5.1 on page 9. To visually verify Assumption 3.2, we did not apply any privacy protection in this experiment and set $\kappa=0.1$. Figure 12 on page 33 show that the SW distance increases after applying compression (SWC), indicating more variability. However, this does not lead to a higher probability of divergence compared to the original model, confirming that compression does not negatively impact the model's ability to learn and update, as stated in Assumption 3.2.

---

> ### Author Response · Authors · 2025-11-21
> **Response of W2**
>
> - **W2: Experiment Scope.** Thank you for the constructive comments. We conduct extensive additional experiments to further assess the performance of the proposed method.
>   - **Comparisions with more baseline methods and runtime.** For private BO baselines, we have evaluate the most relevant and scalable recent DP-BO method of Sopa et al. (2025). Their approach performs offline differentially private Bayesian optimization by repeatedly updating a GP posterior under concentrated DP, making it accurate but computationally expensive for growing datasets. In this small-sample regime, we compare our LDP-BO, with and without sliced Wasserstein compression (SWC), and the DP-BO method of Sopa et al. (2025) in terms of parameter estimation accuracy and runtime. Specifically, we evaluate all methods on the linear model from Example 5.1 using the first 2,000 observations, where the computational cost of their method remains manageable. Table  reports both runtime and MSE at $t=200,\,500,\,1{,}000,\,1{,}500,$ and $2{,}000$ over 50 repetitions.
>   The results show that LDP-BO achieves accuracy comparable to the offline DP-BO method, but with substantially lower computation time. In contrast, the runtime of offline DP-BO grows quickly with sample size because it repeatedly processes the full dataset, making it impractical in streaming settings. Even the non-SWC version of LDP-BO becomes increasingly costly as the kernel matrix grows over time. Incorporating SWC further reduces runtime, especially for larger sample sizes. Additional results and visualizations are provided in Figure 4 and Table 3 on page 10 of the revised manuscript.
> |$n$|Time (minutes)|||MSE|||
> |---:|:---:|:---:|:---:|:---:|:---:|:---:|
> ||**LDP-BO**|**LDP-BO (No SWC)**|**Offline**|**LDP-BO**|**LDP-BO (No SWC)**|**Offline**|
> |200|0.30|0.38|0.47|0.120 (0.020)|0.110 (0.020)|0.090 (0.015)|
> |500|0.77|1.37|2.47|0.090 (0.015)|0.080 (0.014)|0.055 (0.010)|
> |1000|1.60|5.58|12.40|0.075 (0.010)|0.068 (0.009)|0.045 (0.008)|
> |1500|2.52|35.08|57.95|0.060 (0.008)|0.055 (0.007)|0.043 (0.007)|
> |2000|3.17|151.92|183.25|0.055 (0.007)|0.052 (0.006)|0.041 (0.006)|
>   - **Additional real-data analysis.** To broaden the evaluation beyond the transportation domain, we have added experiments on a second real-world dataset: credit card fraud detection, representing a finance/security application. We compare the proposed LDP-BO with offline DP-BO (evaluated only on an initial prefix of the stream, where their runtime remains manageable) and LDP-SGD. The results in Table below show that LDP-BO consistently outperforms or matches these baselines across both tasks.
> |Sample Size|Uber Dataset (Prediction Error)|||Credit Dataset (Accuracy)|||
> |-|:---:|:---:|:---:|:---:|:---:|:---:|
> ||**LDP-BO**|**DP-BO**|**LDP-SGD**|**LDP-BO**|**DP-BO**|**LDP-SGD**|
> |2,000|5.471|5.129|17.412|0.941|0.944|0.913|
> |5,000|2.224|\*|10.271|0.944|\*|0.929|
> |10,000|1.409|\*|3.252|0.951|\*|0.940|
> |20,000|0.782|\*|1.794|0.969|\*|0.952|
>
> In addition, We have included a real-data sensitivity analysis for the compression parameter $\kappa$. As shown in table below $\kappa=0.1$ consistently provides a good balance between predictive accuracy and computational efficiency across both datasets, and is used as the default value in our experiments.
>
> | $\kappa$ | Uber (Prediction Error) | Credit (Accuracy) |
> |----------|-------------------------|-------------------|
> | 0.05     | 0.711                   | 0.971             |
> | 0.10     | 0.782                   | 0.969             |
> | 0.20     | 1.243                   | 0.958             |
> | 0.50     | 3.745                   | 0.921             |

---

> ### Author Response · Authors · 2025-11-21
> **Response of W3**
>
> - **W3: Privacy-Utility Trade-off.**  In the following, we make more explainations on the privacy composition, budget allocation, and privacy amplification.
>    - **Privacy composition.** Our method operates under local differential privacy (LDP), where each data point is privatized independently before processing. In the streaming setting, each time step corresponds to a new individual contributing one data point, so the privacy guarantee is per user and does not accumulate over time. This follows from the *parallel composition* property of LDP: if mechanisms $\{\mathcal{A}_1,\ldots,\mathcal{A}_n\}$ act on disjoint data and each satisfies $(\varepsilon_i,\delta_i)$-LDP, then their combination satisfies $(\max_i \varepsilon_i,\, \max_i \delta_i)$-LDP (Proposition A.5 of Xiong et al., (2022)). This property enables LDP in streaming settings, where privacy budgets are managed per participant rather than globally over time.
>    - **Budget allocation.** We clarify that our method operates under the Local Differential Privacy (LDP) model, where each data point is privatized independently before being processed. Consequently, privacy does not accumulate over iterations, and no global privacy budget is required. At iteration $t$, an $(\varepsilon_t, \delta_t)$-LDP mechanism is applied to the individual sample $\boldsymbol{z}_t$, and the overall guarantee follows directly from the parallel composition property of LDP. As stated in Proposition A.5 (Appendix A; see also Property 2 in Xiong et al., (2022), if each mechanism $\mathcal{A}_t$ satisfies $(\varepsilon_t, \delta_t)$-LDP on disjoint data, then the joint mechanism satisfies $(\max_t \varepsilon_t, \max_t \delta_t)$-LDP. Thus, the overall privacy guarantee is determined solely by the largest per-iteration budget.
>   In our experiments, we consider two representative choices of per-iteration privacy budgets:
>      - **Constant budgets:** $\varepsilon_t \equiv 1$ or $\varepsilon_t \equiv 2$;
>      - **Heterogeneous budgets:** $\varepsilon_t \sim \mathrm{Unif}[1,2]$, which yields a global $(2,\delta)$-LDP guarantee by Proposition A.5.
>   These settings demonstrate that our framework naturally accommodates both homogeneous and heterogeneous per-iteration privacy budgets. The corresponding empirical results are provided in Figure 2 (page 9) and Figure 5 (page 23) of the revised manuscript. Fixed privacy budget method offers more stable performance but may lead to inefficient use of privacy resources, while the heterogeneous privacy budget method provides more flexibility in resource allocation, potentially improving convergence speed but with varying levels of privacy protection across queries.
>
>    - **Privacy mechanisms.** Our framework can incoporate different additive-noise mechanisms, and we conduct additional experiments under the linear model of Example 5.1 to investigate the performance of the proposed method across different privacy mechanisms. Fixing the target privacy level to $(\varepsilon,\delta) = (2,0.2)$, we evaluate four mechanisms: (i) our directly calibrated $(\varepsilon,\delta)$-DP Gaussian mechanism, (ii) a $\mu$-GDP mechanism, (iii) an RDP mechanism with order $\alpha = 2$, and (iv) an $\varepsilon$-DP Laplace mechanism. For a fair comparison, we convert all parameters so that they correspond to the same $(\varepsilon,\delta)$ guarantee; for example, $(\varepsilon,\delta) = (2,0.2)$ is approximately equivalent to $\mu \approx 1.66$ under GDP and to order-$2$ RDP with $\varepsilon_2 \approx 0.39$.
>
>   Table below shows that the GDP-based calibration provides the strongest predictive performance among the Gaussian mechanisms while respecting the same privacy budget. Overall, our conclusions remain consistent across all accounting schemes.
>
> |Privacy Mechanism|$n = 5,000$|$n = 10,000$|$n = 15,000$|$n = 20,000$|
> |----------|-----------:|-----------:|-----------:|-----------:|
> |$(\varepsilon,\delta)$-DP|6.05 (1.10)|1.62 (0.35)|0.976 (0.21)|0.513 (0.12)|
> |$\varepsilon$-DP|6.80 (1.25)|1.85 (0.40)|1.10 (0.25)|0.600 (0.14)|
> |$\mu$-GDP|3.81 (0.75)|1.34 (0.28)|0.66 (0.14)|0.34 (0.08)|
> |RDP|4.46 (0.85)|1.46 (0.30)|0.77 (0.16)|0.46 (0.11)|
>    - **Privacy amplification.** In our setting, the LDP mechanism is applied independently to each incoming data point: at time $t$, a single user contributes one privatized sample, which is then passed to the algorithm. The sliced Wasserstein compression (SWC) step does not involve subsampling or reusing raw observations. Instead, it is applied only to the BO surrogate model and its candidate evaluation points to control the size of the active set used for kernel updates.
>
>   Because SWC operates entirely on already privatized data, it is purely a post-processing step and therefore does not affect the underlying $(\varepsilon,\delta)$-LDP guarantees, in accordance with the post-processing property of differential privacy (Dwork et al., 2006; Dong et al., 2022). We thus do not claim any additional privacy amplification from SWC.

---

> ### Author Response · Authors · 2025-11-21
> **Response of W4**
>
> - **W4: Presentation issues.** Thank you for your careful reading. In the revision, we will expand our discussion to address the following issues in greater depth.
>
>   - **GI optimization.** Regarding the optimization of the acquisition function $GI$ in Eq. (3), we follow standard Bayesian optimization practices using the open-source PyTorch framework BoTorch (Balandat et al., 2020). BoTorch is a modern programming framework for Bayesian optimization that combines Monte-Carlo (MC) acquisition functions, auto-differentiation, and variance reduction techniques. At each iteration, we compute $GI$ and use BoTorch's built-in MC-based acquisition optimization routines. This step is similar to traditional offline methods, with the key difference in the online setting being the subsequent selection of points based on the chosen candidates.
>
>   We clarify that the $p+1$ points for the acquisition function are candidate query points, not additional data samples. At each time $t$, only a single new data point is observed from the stream. The set $\boldsymbol{\xi} = (\xi_1, \dots, \xi_{p+1})$ is obtained by optimizing the acquisition function over the design space: $$\boldsymbol{\xi} = \arg\min_{\boldsymbol{\xi}':\,|\boldsymbol{\xi}'| = p+1} \text{GI}(\boldsymbol{\xi}'; \mathcal{D}_{t-1}, \theta^{(t)}),$$ which can be efficiently implemented using standard BO optimizers, such as multi-start gradient-based search (e.g., BoTorch), without requiring extra storage or processing of observations.
>
>   - **Adaptive $\epsilon_t$.** We clarify that under LDP there is no notion of a total privacy budget, and therefore adaptive choices of $\varepsilon_t$ do not yield improved budget allocation over time. Each data point is privatized independently, and privacy loss does not accumulate across iterations. By the parallel composition property, the overall privacy guarantee is determined by the *maximum*---rather than the sum---of the per-iteration privacy budgets. Thus, no global per-user budget is required, and no privacy loss accumulates over time.
>
>   - **Hyper parameter tuning.** In the revised manuscript, we have added comprehensive ablation and sensitivity studies. Specifically, we conduct these experiments on the linear regression model from Example 5.1, where the response is generated as $y_t = \boldsymbol{x}_t^\top \boldsymbol{\theta}^\star + \varepsilon_t$. We systematically vary three key parameters of our proposed LDP-BO procedure and evaluate their effect on the MSE: the privacy budget $\varepsilon \in [0.5,10]$, the initial step size $\gamma_0 \in [0.1,1]$ in the schedule $\eta_t = \gamma_0 t^{-\alpha}$, and the compression threshold $\kappa \in [0.01,0.5]$. Table below reports the results for different choices of these tuning parameters. In each experiment, a single parameter is varied while the remaining parameters are fixed at their default values used in our main manuscript; see pages 27-28.
> | $\varepsilon$ | MSE | $\gamma_0$ | MSE | $\kappa$ | Time(min) | MSE |
> |-|-|-|-|-|--|-|
> |0.5|18.10|0.1|8.61|0.01|318.7|1.41|
> |1|3.4|0.2|1.83|0.05|165.3|1.59|
> |U(1,2)|2.73|0.3|2.01|0.10|33.0|1.88|
> |2|1.81|0.5|2.41|0.20|29.2|4.50|
> |5|1.65|1| 9.63|0.50|18.8|16.30|
>
>   The findings indicate the following:
>   - Privacy budget $\varepsilon$: increasing $\varepsilon$ weakens privacy protection and consequently improves estimation accuracy;
>   - Initial step size $\gamma_0$: the proposed method is robust to the choice of initial step size over a broad range;
>   - Compression threshold $\kappa$: $\kappa$ induces a clear trade-off between estimation quality and runtime, with smaller values leading to faster execution but slightly reduced accuracy.
>
>   Further details are provided in Section D.2 on pages 27--28 of the revised manuscript.

---

> ### Author Response · Authors · 2025-11-21
> **Response of Q1**
>
> - **Q1: SWC Mechanism.** We make the following clarification on the SWC mechanism.
>   - **SWC complexity.** Thank you for the insightful comment. We clarify that although SWC requires multiple evaluations of the sliced Wasserstein (SW) distance, its per-iteration computational complexity does *not* grow with the stream length $t$, and the overall computational cost remains manageable---particularly in comparison with competing methods.
>      - **Theoretical complexity.** In the revised manuscript, Section E.1 (page 31) presents a detailed complexity analysis. We show that the per-iteration computational cost of our update is $$\mathcal{O}( L M_t^{3} log M_t + Lp M_t^{3}),$$ where $L$ is the number of projection directions, $p$ is the covariate dimension, and $M_t$ is the size of the intermediate dictionary at iteration $t$. Because SWC prunes the dictionary and keeps $M_t$ uniformly bounded (Theorem 3.3, page 7), this per-iteration cost is *independent* of the sample size or stream length $t$. Consequently, the total computational complexity is linear in $T$ since $M_t$ does not grow with $t$. In contrast, kernel-based BO without compression incurs per-iteration costs that grow with the number of past observations, leading to overall complexity on the order of $\mathcal{O}(T^{3})$.
>      - **Empirical results.** We further conducted experiments in Example 5.1 for increasing sample sizes $t$, comparing runtime and MSE for three methods: LDP-BO with SWC, LDP-BO without SWC, and the offline DP-BO baseline (Sopa et al., 2025). As shown in Table below, incorporating SWC yields MSEs comparable to the other two methods but with substantially lower computational time, especially for large $t$. In contrast, both the offline DP-BO baseline and the online variant without SWC exhibit rapidly increasing runtimes due to repeated operations on ever-expanding kernel matrices. Figure 7 on Page 30 further illustrates SWC's adaptivity. SWC maintains per-iteration efficiency independent of $t$, remaining tractable in large-scale online settings while preserving strong estimation performance
> |$t$|Time (minutes)|||MSE|||
> |---:|:---:|:---:|:---:|:---:|:---:|:---:|
> ||**LDP-BO**|**LDP-BO (No SWC)**|**Offline**|**LDP-BO**|**LDP-BO (No SWC)**|**Offline**|
> |200|0.30|0.38|0.47|0.120 (0.020)|0.110 (0.020)|0.090 (0.015)|
> |500|0.77|1.37|2.47|0.090 (0.015)|0.080 (0.014)|0.055 (0.010)|
> |1000|1.60|5.58|12.40|0.075 (0.010)|0.068 (0.009)|0.045 (0.008)|
> |1500|2.52|35.08|57.95|0.060 (0.008)|0.055 (0.007)|0.043 (0.007)|
> |2000|3.17|151.92|183.25|0.055 (0.007)|0.052 (0.006)|0.041 (0.006)|
>     - Hence, both the theoretical analysis and empirical evidence demonstrate that the computational cost introduced by SWC is well controlled. By enforcing a bounded dictionary, SWC prevents the cubic blow-up in runtime characteristic of uncompressed kernel methods. As a result, LDP-BO with SWC is significantly more scalable than existing DP-BO approaches as the sample size grows.
>   - **Violation of line 8.** If the condition in line 8 of Algorithm 2 is not satisfied in the first iteration, no compression is performed, and the new point is simply added to the kernel dictionary. SWC is only triggered once the dictionary exceeds the specified threshold, at which point points are merged or removed according to the sliced Wasserstein criterion. Therefore, there is no need to dynamically adjust the target size of $\mathcal{D}_t$ or $\kappa$; the algorithm behaves like an uncompressed update until compression is required.
>   - **High-dimensional experiments.** We clarify that our method is primarily designed for low- to moderate-dimensional problems, which is precisely the regime where Bayesian optimization is most effective. In high-dimensional environments, additional structural assumptions or specialized algorithms are typically required to maintain sample efficiency (e.g., Frazier, 2018; Hvarfner et al., 2024). While our current algorithm does not target extremely high-dimensional regimes, extending LDP-BO with SWC to such settings is an important and promising direction for future work.
>     In addition, to assess performance in a moderate-dimensional scenario, we extended Example 5.1 to include experiments with covariate dimension $p = 20$. As shown in table below, LDP-BO continues to exhibit strong estimation and prediction accuracy. The conclusions mirror those in the low-dimensional setting: for a fixed privacy budget, LDP-BO consistently matches or outperforms LDP-SGD on linear model. Results of logit, and ReLU regression models can be seen on page 24.
>
> |Model|Privacy level|$t$|LDP-BO|LDP-SGD|
> |-----|-------------|---:|-------|-------|
> |Linear|No DP|5,000|8.79 (3.08)|12.56 (5.65)|
> |||10,000|2.78 (0.97)|3.97 (1.79)|
> |||15,000|1.29 (0.45)|1.84 (0.83)|
> |||20,000|0.73 (0.26)|1.05 (0.47)|
> ||$\varepsilon = 2$|5,000|19.75 (6.91)|28.21 (12.69)|
> |||10,000|9.37 (3.28)|13.39 (6.03)|
> |||15,000|5.06 (1.77)|7.23 (3.25)|
> |||20,000|3.04 (1.06)|4.35 (1.96)|

---

> ### Author Response · Authors · 2025-11-21
> **Response of Q2, Q3 and Q4**
>
> - **Q2: Privacy integration.**
>   - **Sensitivity bound.** Regarding the sensitivity bound in Section 3.1 and Theorem 3.1, we now clarify its connection to our assumptions. Assumption 4.1 asserts a uniform gradient bound, i.e., there exists $B < \infty$ such that $\|\nabla L(\theta, z_t)\| \leq B$ for all $t$ and $\theta$, a standard condition in (L)DP optimization (see Song et al., 2013 and Sopa et al., 2025). In Algorithm 1, we clip the surrogate gradient to this bound and then apply a Gaussian mechanism calibrated to the resulting global $\ell_2$-sensitivity. This clipping step ensures that any two neighboring inputs can alter the clipped gradient by at most $2B$ in $\ell_2$-norm. No additional boundedness assumption is required, as the sensitivity follows directly from Assumption 4.1.
>   - **Gaussian v.s. Laplace.** The Gaussian mechanism is preferable to the Laplace mechanism, particularly for high-sensitivity queries, because it offers a better privacy--accuracy trade-off. The Laplace mechanism scales noise with the $L_1$ sensitivity, often leading to large perturbations, whereas the Gaussian mechanism scales with the $L_2$ sensitivity and typically requires less noise for the same privacy level. Its lighter tails further reduce distortion in the released values. We have conducted additional experiments comparing the two mechanisms; please see our response to W3 for details.
>   - **Empirical comparison of RDP/GDP noise levels.** We conduct additional experiments to compare the proposed method under different privacy mechanisms. For further details, please refer to our response to W3.
>   - **Adaptive adversaries querying outputs.** Our mechanism is local and non-interactive at the user level. At time $t$, each individual contributes a single sample $z_t$, which is privatized once via an $(\varepsilon,\delta)$-LDP mechanism; only the privatized value is stored and used by the BO algorithm. All subsequent acquisition decisions and predictions are arbitrary post-processing of these locally privatized observations. By the standard robustness of LDP to post-processing and adaptive querying, such downstream adaptivity cannot increase the per-user privacy loss beyond $(\varepsilon,\delta)$, and no privacy composition occurs over time for the same individual.
>
> - **Q3: Theoretical analysis.** We clarify that, when the privacy noise is turned off and a fixed step size is used, our analysis recovers the same stationarity rate as in the non-private setting presented by Müller et al. (2021). In other words, the additional $O(1/t^{1-\alpha})$ term in Theorem 4.5 can be interpreted as the cost of privacy and the constraints imposed by streaming data. However, in the limit where no privacy noise is applied, our framework aligns with the known non-private Bayesian optimization rates.
>   Regarding the definition of $t_0$ in Theorem 4.4, we have revised the manuscript to explicitly define $t_0 = \min\{t : \lambda \ge 2a_p^2\eta_t, \lambda\eta_t t \ge 8\alpha\log t\}$ immediately before the statement of the theorem.
>
> - **Q4: Additional experiments.** We appreciate the reviewer's valuable suggestions. The ablation studies requested are already addressed in our previous responses:
>   - \(i\) The comparison of MSE with on the Uber dataset, along with the predictions on the added Credit dataset, is discussed in W2.
>   - \(ii\) The impact of the total privacy budget is elaborated on in W3.
>   - \(iii\) The evolution of dictionary size is covered in Q1.
>   As such, we focus on presenting the new experimental results here.
>   We have compared the runtime of LDP-BO and LDP-SGD across different dimensions. Specifically, table below compares the runtime (in minutes) between LDP-BO and LDP-SGD across various models and dimensions, based on 50 replications. As expected, LDP-BO consistently takes more time than LDP-SGD due to the inherent exploration process of Bayesian Optimization, which is unavoidable. However, the results clearly show that LDP-BO significantly outperforms LDP-SGD, particularly in more complex models (Logit and ReLU). This demonstrates the trade-off between time and performance, where LDP-BO sacrifices some computational efficiency for much better results in challenging settings.
>
>
> |Dimension|Linear Model||Logit Model||ReLU Model||
> |---------|-----------:|:---------:|------------:|:---------:|-----------:|:---------:|
> ||**LDP-BO**|**LDP-SGD**|**LDP-BO**|**LDP-SGD**|**LDP-BO**|**LDP-SGD**|
> |$p=2$|29.58|0.78|31.78|0.84|32.33|0.80|
> |$p=5$|75.55|1.45|138.92|1.73|144.08|1.51|
> |$p=20$|92.78|3.12|145.42|3.85|148.52|3.20|
>
>   We focus on the first dimension to provide a clear and intuitive visualization of parameter changes across methods. This approach allows for easier comparison of the behaviors of different methods over time. Additionally, we present the MSE values in box plots to the right of the line charts, offering a more comprehensive comparison of the performance, including variability and central tendency, across methods.

---

> ### Author Response · Authors · 2025-11-21
> **Response of Q5 and Rererences**
>
> - **Q5: Broader Impact**. Thank you for the constructive suggestions. We will incorporate the following discussions in the revision.
>   - **Limitation discussion** We have discussed the potential failure modes and sources of bias of Sliced Wasserstein Compression (SWC) in Section E.3 (Page 32).
>     While our method is designed to enable privacy-preserving streaming Bayesian Optimization (BO), there are some inherent limitations that must be considered. First, at very small privacy budgets $\varepsilon$, we observe a risk of utility collapse: model accuracy can deteriorate as privacy protection becomes increasingly stringent. This phenomenon is well documented in privacy-preserving machine learning and highlights the need for careful calibration of the privacy budget. Second, while SWC effectively controls kernel growth, it may introduce bias through the choice of projection directions used in the compression step. Such bias can obscure fine-grained structure in the data distribution, particularly in highly structured or multimodal settings. We plan to explore further refinements in future work.
>   - **Reproducibility.** Regarding reproducibility, we have added details on the kernel choice (Radial Basis Function), $\eta_t=\eta_0t^{-\alpha}$ with $\eta_0=0.2, \alpha=0.505$, and the exact random seed used for 100 repetitions in line 557-5558 on page 11.
>
> **References**
>
> Barber R F, Candes E J, Ramdas A, et al. Conformal prediction beyond exchangeability\[J\]. The Annals of Statistics, 2023, 51(2): 816-845.
>
> Vaswani S, Dubois-Taine B, Babanezhad R. Towards noise-adaptive, problem-adaptive (accelerated) stochastic gradient descent\[C\]//International Conference on Machine Learning. PMLR, 2022: 22015-22059.
>
> Zhu W, Chen X, Wu W B. Online covariance matrix estimation in stochastic gradient descent\[J\]. Journal of the American Statistical Association, 2023, 118(541): 393-404.
>
> Choi T, Schervish M J. On posterior consistency in nonparametric regression problems\[J\]. Journal of Multivariate Analysis, 2007, 98(10): 1969-1987.
>
> Koppel A, Pradhan H, Rajawat K. Consistent online gaussian process regression without the sample complexity bottleneck\[J\]. Statistics and Computing, 2021, 31(6): 76.
>
> Sopa G, Marusic J, Avella-Medina M, et al. Scalable Differentially Private Bayesian Optimization\[J\]. arXiv preprint arXiv:2502.06044, 2025.
>
> Xiong X, Liu S, Li D, et al. A comprehensive survey on local differential privacy\[J\]. Security and Communication Networks, 2020, 2020(1): 8829523.
>
> Dwork C, McSherry F, Nissim K, et al. Calibrating noise to sensitivity in private data analysis\[C\]//Theory of Cryptography Conference. Berlin, Heidelberg: Springer Berlin Heidelberg, 2006: 265-284.
>
> Dong J, Roth A, Su W J. Gaussian differential privacy\[J\]. Journal of the Royal Statistical Society Series B: Statistical Methodology, 2022, 84(1): 3-37.
>
> Balandat M, Karrer B, Jiang D, et al. BoTorch: A framework for efficient Monte-Carlo Bayesian optimization\[J\]. Advances in Neural Information Processing Systems, 2020, 33: 21524-21538.
>
> Frazier P I. A tutorial on Bayesian optimization\[J\]. arXiv preprint arXiv:1807.02811, 2018.
>
> Hvarfner C, Hellsten E O, Nardi L. Vanilla Bayesian optimization performs great in high dimensions\[J\]. arXiv preprint arXiv:2402.02229, 2024.
>
> Müller S, von Rohr A, Trimpe S. Local policy search with Bayesian optimization\[J\]. Advances in Neural Information Processing Systems, 2021, 34: 20708-20720.

---

### Official Review · Reviewer_574W · 2025-10-31

**Soundness:** 3
**Presentation:** 3
**Contribution:** 3
**Rating:** 6
**Confidence:** 3

**Summary:**

This paper proposes a novel online, gradient-free Bayesian optimization (BO) framework with local differential privacy (LDP) guarantees, addressing the limitations of traditional BO in dynamic environments. Experimental results show significant performances of the proposed framework.

**Strengths:**

1.	This is the first work to tackle online Bayesian optimization under the local differential privacy model.
2.	The proposed Sliced Wasserstein Compression (SWC) is a novel algorithm that can manage the kernel dictionary in streaming data environments.
3.	The paper is well-organized, with a clear introduction, detailed methodology, thorough theoretical analysis, and comprehensive experimental results.

**Weaknesses:**

1.	The SWC algorithm requires multiple calculations of the slice Wasserstein distance, which consumes more computing resources.
2.	The method introduces several new hyperparameters: the compression budget, and the clipping bound. The sensitivity of the algorithm to these choices is not explored.
3.	The paper could enhance its originality by more clearly distinguishing its contributions from existing methods.

**Questions:**

1.	Why is it necessary to use the SWC method specifically for compression? What is the innovation of SWC?
2.	How does this method perform in high-dimensional environments?
3.	How to set compression budget, and the clipping bound in practice. Is performance robust to their selection, or does it require extensive tuning?

---

> ### Author Response · Authors · 2025-11-21
> **Response of W1 and W2&Q3**
>
> Thank you for your thoughtful and critical assessment. Many of your comments will help us produce a more readable and self-contained version of the paper. Below, we address each of your specific concerns in turn.
> - **W1: Computational complexity of SWC.** We have clarified that although SWC requires multiple evaluations of the sliced Wasserstein (SW) distance, its per-iteration computational complexity does *not* grow with the stream length $t$, and the overall computational cost remains manageable---particularly in comparison with competing methods.
>   - **Theoretical complexity.** In the revised manuscript, Section E.1 (page 31) presents a detailed complexity analysis. We show that the per-iteration computational cost of our update is
> $$
> O( L M_t^3 \log M_t + L p M_t^3),
> $$
> where $L$ is the number of projection directions, $p$ is the covariate dimension, and $M_t$ is the size of the intermediate dictionary at iteration $t$. Because SWC prunes the dictionary and keeps $M_t$ uniformly bounded (Theorem 3.3, page 7), this per-iteration cost is *independent* of the sample size or stream length $t$. Consequently, the total computational complexityis linear in $T$ since $M_t$ does not grow with $t$. In contrast, kernel-based BO without compression incurs per-iteration costs that grow with the number of past observations, leading to overall complexity on the order of $O(T^3)$.
>   - **Empirical results.** We further conducted experiments in Example 5.1 for increasing sample sizes $t$, comparing runtime and MSE. As shown in Table below, incorporating SWC yields MSEs comparable to the other two methods but with substantially lower computational time, especially for large $t$.
> |$t$|Time (minutes)|||MSE|||
> |---:|:---:|:---:|:---:|:---:|:---:|:---:|
> ||**LDP-BO**|**LDP-BO (No SWC)**|**Offline**|**LDP-BO**|**LDP-BO (No SWC)**|**Offline**|
> |200|0.30|0.38|0.47|0.120 (0.020)|0.110 (0.020)|0.090 (0.015)|
> |500|0.77|1.37|2.47|0.090 (0.015)|0.080 (0.014)|0.055 (0.010)|
> |1000|1.60|5.58|12.40|0.075 (0.010)|0.068 (0.009)|0.045 (0.008)|
> |1500|2.52|35.08|57.95|0.060 (0.008)|0.055 (0.007)|0.043 (0.007)|
> |2000|3.17|151.92|183.25|0.055 (0.007)|0.052 (0.006)|0.041 (0.006)|
>   - Hence, both the theoretical analysis and empirical evidence demonstrate that the computational cost introduced by SWC is well controlled.
>
> - **W2&Q3: Sensitivity analysis of compression budget and clipping bound.** We have included additional experiments to assess the sensitivity of both the compression budget and the clipping (or weighting) mechanism.
>   - **Sensitivity to the compression budget**. We have conducted an ablation study over the compression parameter $\kappa$ using the linear model in Example 5.1. As shown in Table below, very small values of $\kappa$ (e.g., $\kappa=0.01$) achieve the best MSE but incur substantial computational cost. Moderate values such as $\kappa=0.10$ reduce runtime by an order of magnitude while incurring only a mild increase in MSE. Noticeable degradation appears only when $\kappa$ becomes relatively large (e.g., $\kappa=0.5$). These results indicate that $\kappa$ can be selected coarsely without extensive fine-tuning.
>
> | $\kappa$ | Time (minutes) | MSE ($\times 10^{-3}$) |
> |-|--|--|
> | 0.01| 318.7 | 1.41|
> | 0.05 | 165.3| 1.59|
> | 0.10| 33.0 | 1.88 |
> | 0.20| 29.2  | 4.50|
> | 0.50 | 18.8 | 16.30 |
> - **Clipping v.s. Mallow's weighting.** In practice, we have employed Mallow's weights rather than gradient clipping to ensure boundedness. Mallow-type weighting directly adjusts the loss rather than truncating the estimating equation. It preserves consistency and asymptotic unbiasedness even under noise or privacy constraints (Avella-Medina et al. 2023 and Xie et al. 2025) . In contrast, gradient clipping alters the estimating equation itself and typically introduces a non-vanishing bias that depends on the clipping threshold.
>
> To illustrate this in our setting, we replicate Example 5.1 with a logistic regression model under Mallow weighting $\omega(x) = \min(1, \frac{2}{\|x\|^2})$
> and under clipping bound $\sqrt{2}$. This setting ensures that both methods have the same sensitivity. Table below shows that Mallow weighting retains tight concentration around the true value (1.0) across all privacy levels, while clipping consistently produces upward-biased estimates.
> |Method|No DP|$\varepsilon=2$|$\varepsilon\in[1,2]$|$\varepsilon=1$|
> |---|--|:-:|:-|-|
> |Mallow weights|$1.00\,(0.02)$|$0.99\,(0.05)$|$1.02\,(0.06)$|$0.98\,(0.08)$|
> |Clipping|$1.15\,(0.02)$|$1.18\,(0.05)$|$1.20\,(0.07)$|$1.19\,(0.08)$|
> - **Practical tuning guidelines.** In practice, $\kappa$ reflects the trade-off between computational efficiency and predictive accuracy. A simple approach is to perform cross-validation on a small held-out prefix of the data stream over a short grid of $\kappa$ values, and select the largest $\kappa$ that maintains acceptable prediction error. This procedure is fast and avoids extensive hyperparameter searches.

---

> ### Author Response · Authors · 2025-11-21
> **Response W3,Q1 and Q2**
>
> - **W3: Contributions from existing methods.**
> We have now strengthened the discussion of our methodological contributions relative to existing work.
>    - Online Bayesian optimization under LDP is highly relevant to applications such as IoT edge analytics, dynamic pricing (e.g., surge pricing), and credit card fraud detection---settings where data arrive rapidly, cannot be stored in full, and contain sensitive information. In these environments, one must make sequential decisions from streaming data while ensuring strong privacy guarantees and maintaining low computational overhead. This naturally motivates an online, privacy-aware BO framework. However, despite the practical importance of this problem, we are not aware of prior work that simultaneously addresses high-volume data streams, Bayesian optimization, and per-iteration LDP guarantees in a unified setting.
>    - Table below provides a structured comparison of recent developments in differentially private Bayesian optimization (DP-BO). As shown in the table, prior Bayesian approaches are all developed in *offline* settings under the central differential privacy (CDP) model.
> |Reference|Setting|DP Model|Bayesian|
> |---------|-------|--------|--------|
> |Triastcyn and Faltings (2020)|Offline|CDP|True|
> |Zhang and Zhang (2023)|Offline|CDP|True|
> |Sopa et al. (2025)|Offline|CDP|True|
> |Duchi and Ruan (2024)|Offline|LDP|False|
> |Xie et al. (2025)|Online|LDP|False|
> |Proposed|Online|LDP|True|
>    - Consequently, no existing method is simultaneously *Bayesian*, *online*, and *LDP*. Our proposed framework fills precisely this gap by enabling Bayesian optimization under local privacy constraints in an online environment. Among existing baselines, Xie et al. (2025) is the closest to our setting (online and LDP), which is why we use it as our primary comparator in the experiments.
>
>
>
> - **Q1: Innovation of SWC**. Our use of SWC is motivated primarily by computational considerations. In kernel-based BO, the cost of each update grows with the number of past observations because the kernel matrix expands over time. Standard offline or uncompressed online methods therefore incur per-iteration costs that grow with the stream length $t$, leading to overall complexities on the order of $\mathcal{O}(t^{3})$, which is infeasible in large-scale streaming settings. SWC is introduced specifically to prevent this growth: by pruning and merging dictionary elements at each step, it ensures that the effective dictionary size $\tilde M_t$ remains uniformly bounded (Theorem 3.3, page 7).
>
> The innovation of SWC lies in its adaptive and discrepancy-driven compression mechanism. Unlike random feature truncation (Liu et al., 2022) and the Nyström approximation (Abedsoltan et al., 2024) that fix the number of inducing points or the feature budget a priori, SWC employs a sliced Wasserstein discrepancy threshold $\kappa$ to determine which dictionary elements to retain. At each iteration, it assesses how removing a candidate point would alter the posterior and prunes points whenever the change remains below $\kappa$. As a result, the model order $M_t$ is not predetermined but adaptively selected based on the data and posterior geometry. Theoretical guarantees (Theorem 3.3 and Theorem 4.4) show that this threshold-based strategy keeps $M_t$ uniformly bounded while introducing only a controlled, $\kappa$-dependent error, providing a principled and flexible way to balance accuracy and computational efficiency.
>
> - **Q2: Performance in high-dimensional environments.** We clarify that our method is primarily designed for low- to moderate-dimensional problems, which is precisely the regime where Bayesian optimization is most effective. In high-dimensional environments, additional structural assumptions or specialized algorithms are typically required to maintain sample efficiency (e.g., Frazier, 2018; Hvarfner et al., 2024). While our current algorithm does not target extremely high-dimensional regimes, extending LDP-BO with SWC to such settings is an important and promising direction for future work.
>
> In addition, to assess performance in a moderate-dimensional scenario, we extended Example 5.1 to include experiments with covariate dimension $p = 20$. As shown in Table below, LDP-BO continues to exhibit strong estimation and prediction accuracy. The conclusions mirror those in the low-dimensional setting: for a fixed privacy budget, LDP-BO consistently matches or outperforms LDP-SGD. Resluts of logit and ReLU regression models can be seen on page 24 .
> |Model|Privacy level|$t$|LDP-BO|LDP-SGD|
> |-----|-------------|---:|-------|-------|
> |Linear|No DP|5,000|8.79 (3.08)|12.56 (5.65)|
> |||10,000|2.78 (0.97)|3.97 (1.79)|
> |||15,000|1.29 (0.45)|1.84 (0.83)|
> |||20,000|0.73 (0.26)|1.05 (0.47)|
> ||$\varepsilon = 2$|5,000|19.75 (6.91)|28.21 (12.69)|
> |||10,000|9.37 (3.28)|13.39 (6.03)|
> |||15,000|5.06 (1.77)|7.23 (3.25)|
> |||20,000|3.04 (1.06)|4.35 (1.96)|

---

> > ### Comment · Reviewer_574W · 2025-11-27
> >
> > Thank you for the detailed response, I tend to keep my positive score.

---

> > > ### Author Response · Authors · 2025-11-27
> > >
> > > Thank you for the positive and engaging feedback. We welcome any further questions or discussion.

---

> ### Author Response · Authors · 2025-11-21
> **References of Response**
>
> Sopa G, Marusic J, Avella-Medina M, et al. Scalable Differentially Private Bayesian Optimization[J]. arXiv preprint arXiv:2502.06044, 2025.
>
> Avella-Medina M, Bradshaw C, Loh P L. Differentially private inference via noisy optimization[J]. The Annals of Statistics, 2023, 51(5): 2067-2092.
>
> Xie J, Shi E, Jiang B, et al. Online differentially private inference in stochastic gradient descent[J]. arXiv preprint arXiv:2505.08227, 2025.
>
> Liu F, Huang X, Chen Y, et al. Random features for kernel approximation: A survey on algorithms, theory, and beyond[J]. IEEE Transactions on Pattern Analysis and Machine Intelligence, 2021, 44(10): 7128-7148.
>
>
> Abedsoltan A, Pandit P, Rademacher L, et al. On the Nyström approximation for preconditioning in kernel machines[C]//International Conference on Artificial Intelligence and Statistics. PMLR, 2024: 3718-3726.
>
> Frazier P I. A tutorial on Bayesian optimization[J]. arXiv preprint arXiv:1807.02811, 2018.
>
>
> Hvarfner C, Hellsten E O, Nardi L. Vanilla Bayesian optimization performs great in high dimensions[J]. arXiv preprint arXiv:2402.02229, 2024.

---

### Official Review · Reviewer_Cyco · 2025-10-31

**Soundness:** 2
**Presentation:** 2
**Contribution:** 2
**Rating:** 2
**Confidence:** 3

**Summary:**

This paper introduces an online, locally differentially private (LDP) framework for Bayesian Optimization (BO) in streaming data environments. The proposed algorithm, termed LDP-BO, enables gradient-free, privacy-preserving, and memory-efficient optimization by integrating two key components: (1) a Sliced Wasserstein Compression (SWC) scheme for maintaining a bounded-size Gaussian process (GP) kernel dictionary, and (2) a per-iteration LDP mechanism that injects Gaussian noise directly into the gradient-free update rule.

The authors provide non-asymptotic convergence guarantees under both strongly convex and general smooth loss functions, showing that the method achieves SGD-like rates while ensuring local privacy. Theoretical analysis is supported by experiments on synthetic regression models, nonlinear function fitting, and a real-world Uber fares dataset, demonstrating lower mean-squared error and higher stability than LDP-SGD baselines.

**Strengths:**

1. The paper proposes a novel synthesis of online learning, Bayesian optimization, and local differential privacy. The integration of Sliced Wasserstein Compression for kernel dictionary control within an LDP framework is creative and technically non-trivial.

2. The theoretical contributions are strong and carefully derived. The non-asymptotic convergence bounds explicitly separate the effects of privacy noise, compression error, and initialization, offering clear interpretability.

3. The paper is well-written, with precise notation and a clear exposition of algorithms and assumptions. The inclusion of supporting lemmas and appendices strengthens reproducibility and theoretical transparency.

4. Addressing privacy-preserving online Bayesian optimization fills a clear gap in the literature, as prior DP-BO work focused mainly on batch or centralized settings. The method provides a promising foundation for future research on online private learning and real-time optimization.

**Weaknesses:**

1. While the technical setup is solid, the paper could better articulate why online Bayesian optimization under LDP is practically necessary. The current motivation remains abstract, lacking concrete scenarios (e.g., edge devices, adaptive sensor control) that require both streaming and privacy simultaneously.

2. Although SWC ensures bounded dictionary size, it remains unclear how compression affects optimization accuracy or convergence in practice. The theoretical bounds include a κ-dependent error term, but there is no empirical or interpretive analysis showing how varying κ impacts the privacy–utility trade-off.

3. The SWC algorithm, while theoretically elegant, may incur non-trivial computational overhead due to repeated projection and sorting operations in high dimensions. The paper lacks runtime benchmarks or scalability analysis to substantiate the claimed O(1) per-iteration cost.

4. The experiments, though diverse, remain relatively small-scale. There are no comparisons against recent scalable or privacy-aware BO methods (e.g., DP-BO via posterior sampling or distributed DP frameworks). Moreover, results focus mainly on MSE rather than BO metrics such as regret or sample efficiency.

5. The algorithm assumes per-iteration privacy budgets and full control over the noise mechanism, which may not be feasible in realistic streaming environments or federated deployments.

**Questions:**

1. Could the authors provide a clearer interpretation of the privacy–utility–compression trade-off, perhaps through ablations varying ε, κ, and step size η?

2. The Sliced Wasserstein Compression is the most distinctive element. Could the authors demonstrate its empirical effect (e.g., compare with random feature truncation or Nyström methods)?

3. How does LDP-BO handle non-stationary streaming data, where the underlying distribution shifts over time?

4. The theoretical convergence rates resemble those of noisy SGD. Can the authors clarify what benefits BO (as a surrogate model) provides over standard LDP-SGD in this setting?

5. Could the method extend to Reinforcement Learning or contextual bandit frameworks where privacy and exploration–exploitation interplay is central?

---

> ### Author Response · Authors · 2025-11-21
> **Response of W1,W2 and W3**
>
> Thank you for your thoughtful and constructive assessment. Many of your comments have been helpful and will allow us to produce a more readable and self-contained revision of the paper. Below, we address each of your specific concerns in detail.
>
> - **W1: Practical necessity.**
> We now explicitly discuss the practical necessity of online Bayesian optimization under LDP in lines 49–63 on pages 1-2.
>
>   - **From application perspective.** Real-time systems, such as IoT devices, dynamic pricing (e.g., Uber surge pricing), and fraud detection, generate large volumes of streaming data that require timely decisions while protecting sensitive information.  Privacy protection is crucial: Uber trip records contain sensitive data, and training models without privacy safeguards risks regulatory violations and loss of trust. Moreover, data arrives too quickly for full storage, and models must be updated in near real-time to stay accurate. Relying on offline batch training leads to outdated models as demand or fraud patterns change, degrading performance.
>
>   - **From method perspective.** Traditional Bayesian optimization methods are ill-suited for these scenarios: their computational cost grows as $\mathcal{O}(t^3)$ with the number of observations $t$, making them infeasible for high-frequency, large-scale data streams. They also assume access to a static dataset, rendering them incompatible with online settings where data arrive continuously. In contrast, our online Bayesian optimization framework under LDP is designed for streaming environments, provides per-iteration LDP guarantees, and maintains real-time computational efficiency.
>
> - **W2: SWC Mechanism.**
>   - Theoretically speaking, SWC guarantees a bounded dictionary size while introducing a $\kappa$-dependent approximation term. Theorem 4.4 shows:
>   $$
>   E(\|\hat{\Delta}_t\|_2^2) \lesssim t^{-\alpha}( \frac{\eta c_p B^2 \log(1.25/\delta)}{\lambda \varepsilon^2} + \eta \frac{L + p\kappa + 2B^2}{\lambda} + \|\hat{\Delta}_0\|_2^2 )
>   $$
>   where $t^{-\alpha}$ ($1/2 < \alpha < 1$) is the standard decay rate in online kernel optimization, and the term $p\kappa$ captures the approximation error due to compression. This term arises from discarding or merging nearby points and quantifies the trade-off: smaller $\kappa$ yields faster computation with tighter dictionaries; larger $\kappa$ reduces compression and improves approximation fidelity.
>
>   - We have also performed additional simulations on the linear regression model in Example 5.1, varying $\kappa$, and measuring its effect on MSE. Results show a clear compression–utility trade-off: small $\kappa$ yields faster runtimes with slightly lower accuracy, while large $\kappa$ improves accuracy at increased computational cost. These results highlight how SWC mediates the balance between computational efficiency and estimation quality.
>
> | $\kappa$ | Time (minutes) | MSE ($\times 10^{-3}$) |
> |-|-|--|
> | 0.0 | 318.7| 1.41|
> | 0.05 | 165.3 | 1.59 |
> | 0.10 | 33.0 | 1.88 |
> | 0.20 | 29.2 | 4.50 |
> | 0.50 | 18.8  | 16.30 |
>
> - **W3: Runtime and complexity.**
>   - **Meaning of $\mathcal{O}(1)$ per-iteration cost.** Our claim means the per-iteration computational cost does not grow with the stream length $t$. It depends on the dictionary size and feature dimension, unlike offline DP-BO methods (Triastcyn & Faltings, 2020; Zhang & Zhang, 2023; Sopa et al., 2025), whose cost grows linearly with $t$. Section E.1 (page 31) provides a complexity derivation, showing that each update involves controlled projection and pruning operations, with per-iteration complexity $ \mathcal{O}( L \tilde{M}_t^3 \log \tilde{M}_t + L p \tilde{M}_t^3 ) $, independent of $t$.
>
>   - **Empirical comparison.** To highlight this distinction, we evaluated the scalable DP-BO method of Sopa et al. (2025) on a small synthetic dataset, comparing parameter estimation accuracy and runtime. Using the linear model from Example 5.1 and the first 2,000 observations, the table below shows that LDP-BO achieves similar accuracy with much less computation, consistent with its theoretical $\mathcal{O}(1)$ per-iteration complexity. In contrast, offline DP-BO runtimes increase quickly as they reprocess the full dataset, making them unsuitable for streaming, and even processing single points without SWC causes unbounded kernel-matrix growth and rising costs.
>
> |       | Time (min)     |    | | MSE($\times 10^{-3}$) |  | |
> |--|-|-|--|--|---|--|
> | $n$   | LDP-BO| LDP-BO (No SWC) | Offline       | LDP-BO           | LDP-BO (No SWC)  | Offline         |
> | 200   | 0.30 | 0.38| 0.47 | 0.120 (0.020)    | 0.110 (0.020)    | 0.090 (0.015)   |
> | 500   | 0.77 | 1.37| 2.47 | 0.090 (0.015)    | 0.080 (0.014)    | 0.055 (0.010)   |
> | 1000  | 1.60 | 5.58| 12.40 | 0.075 (0.010)    | 0.068 (0.009)    | 0.045 (0.008)   |
> | 1500  | 2.52 | 35.08  | 57.95  | 0.060 (0.008)    | 0.055 (0.007)    | 0.043 (0.007)   |
> | 2000  | 3.17| 151.92  | 183.25  | 0.055 (0.007)    | 0.052 (0.006)    | 0.041 (0.006)   |

---

> ### Author Response · Authors · 2025-11-21
> **Response of W4 and W5**
>
> - **W4: Private BO baselines.**
>   - Our work diverges from the existing methods can be summarized as follows:
>     - **Problem setting**. The proposed LDP-BO method targets online Bayesian optimization with zero-order updates and per-iteration local differential privacy. This places our framework in a problem regime not addressed by prior work. To better contextualize our contributions, compares the most relevant methods:
> | Reference | Setting | DP Model | Bayesian |
> |--|-|-|-|
> | Triastcyn and Faltings (2020)   | Offline | CDP| True |
> | Zhang and Zhang (2023) | Offline | CDP | True |
> | Sopa et al. (2025)   | Offline | CDP  | True |
> | Duchi and Ruan (2024)| Offline | LDP | False|
> | Xie et al. (2025)  | Online  | LDP  | False|
> | Proposed | Online  | LDP| True|
>       - **Bayesian approaches** such as Triastcyn and Faltings (2020), Zhang and Zhang (2023), and Sopa et al. (2025) were all developed for offline settings under the CDP model.
>       - **LDP-based approaches** such as Duchi and Ruan (2024) and Xie et al. (2025) worked under the local DP model.
>
>     To the best of our knowledge, no existing method provides an online, gradient-free Bayesian optimization procedure with rigorous LDP guarantees. LDP-BO fills this gap by enabling fully online BO with per-iteration local privacy. Among prior studies, Xie et al. (2025) is the closest baseline, being both online and LDP, which is why we use it as the primary comparator in our experiments.
>    - To evaluate the performance against offline DP-BO, we test our method on small-instance real data (first 2,000 observations). Table shows that the methods perform similarly at small sample sizes, but as the stream grows, offline approaches become impractical while our online method continues to improve.
> | Sample Size | Uber Dataset (Prediction Error) | | | Credit Dataset (Accuracy) | | |
> |---|-|-|--|---|-|-|
> || LDP-BO|DP-BO|LDP-SGD| LDP-BO|DP-BO| LDP-SGD|
> | 2,000 | 5.471 | 5.129 | 17.412 | 0.941 | 0.944 | 0.913 |
> | 5,000 | 2.224 | \* | 10.271 | 0.944 | \* | 0.929 |
> | 10,000 | 1.409 | \* | 3.252 | 0.951 | \* | 0.940 |
> | 20,000 | 0.782 | \* | 1.794 | 0.969 | \* | 0.952 |
>
>   - **Regret.** In the revised manuscript, we have included cumulative regret evaluations for the Sine and Friedman test functions from Example 5.2 on page 9. Our method is much more sample-efficient and can identify high-reward regions of the search space significantly faster than the competing approach, highlighting its effectiveness in prediction tasks.
> | Method| Sine| Friedman |
> |-|--|--|
> | LDP-BO | 207.873  | 1270.889 |
> | DNN-based baseline | 622.921  | 2275.447 |
>
> - **W5: Privacy budgets.** We thank the reviewer for raising this important point.
>   - **LDP setting and global privacy guarantee**. Our method operates under the local DP (LDP) model, where each data point is privatized independently before processing. This design **avoids the need for a global cumulative privacy budget** and makes per-iteration budgets natural in streaming and federated settings. At iteration $t$, an $(\varepsilon_t,\delta_t)$-LDP mechanism is applied to the individual data point $z_t$. By the **parallel composition property** (Proposition A.5 in Appendix A), if mechanisms act on disjoint data, then the joint mechanism satisfies $(\varepsilon_i,\delta_i)$-LDP. Thus, the global guarantee is determined solely by the largest per-iteration budget, no accumulation or tracking of privacy loss is required.
>   - **Choices of per-iteration privacy budgets**. In our experiments, two choices of per-iteration privacy budgets are already included:
>     - Constant privacy budgets, e.g., $\varepsilon_t \equiv 1$ or $\varepsilon_t \equiv 2$;
>     - Heterogeneous privacy budgets, where $\varepsilon_t$ is drawn independently from a uniform distribution on $[1,2]$. By Proposition A.5, the global guarantee is $(2,\delta)$-LDP, governed by the maximum per-iteration budget.
>   These examples demonstrate that the framework naturally supports both homogeneous and heterogeneous privacy budgets. Results are shown in Figure 2 (page 9) and Figure 5 (page 23) of the revised manuscript.
> - **Extension to federated settings**. For completeness, we also outline how the LDP-BO update naturally extends to federated learning (FL). Consider $N$ clients, where client $j$ holds i.i.d. samples from $\Pi_j$. The central server aims to solve
> $$
> \theta^{\star} = argmin_{\theta}( f(\theta) := \sum^N_{j=1} p_j E_{z_j \sim \Pi_j}[L_j(\theta, z_j)] ),
> $$
> where $p_{j}$ is the weight of the $j$th client and $L_j(\cdot, z_j)$ is the loss function. At time point $t \geq 1$, each client performs a locally private update using a noisy BO-based gradient:
> $$
> \theta_{t}^{j} = \theta_{t-1}^{j} - \eta_{t} g_{t-1}(\theta_{t-1}^j) + \eta_{t} \omega^j_t,
> $$
> where $\omega^j_t$ is properly calibrated LDP noise, and $g_{t-1}(\theta_{t-1})$ is the BO gradient approximation. The central server aggregates the local updates
> $$
> \theta_{t+1} = \sum_{j=1}^{N} p_j \theta_{t}^{j}.
> $$

---

> ### Author Response · Authors · 2025-11-21
> **Response Q1, Q2 and Q3**
>
> - **Q1: Sensitivity analysis**. We have added comprehensive ablation and sensitivity studies. Specifically, we conduct these experiments on the linear regression model from Example 5.1 over 50 repetitions. We systematically vary three key parameters of our proposed LDP-BO procedure and evaluate their effect on the MSE ($\times10^{-3}$): the privacy budget $\varepsilon \in [0.5,10]$, the initial step size $\gamma_0 \in [0.1,1]$ in the schedule $\eta_t = \gamma_0 t^{-\alpha}$, and the compression threshold $\kappa \in [0.01,0.5]$.
> The findings indicate the following:
>   * **Privacy budget $\varepsilon$**: increasing $\varepsilon$ weakens privacy protection and consequently improves estimation accuracy;
>   * **Initial step size $\gamma_0$**: the proposed method is robust to the choice of initial step size over a broad range;
>   * **Compression threshold $\kappa$**: $\kappa$ induces a clear trade-off between estimation quality and runtime, with smaller values leading to faster execution but slightly reduced accuracy.
>
> | $\varepsilon$ | MSE | $\gamma_0$ | MSE | $\kappa$ | Time(min) | MSE |
> |-|-|-|-|-|--|-|
> |0.5|18.10|0.1|8.61|0.01|318.7|1.41|
> |1|3.4|0.2|1.83|0.05|165.3|1.59|
> |U(1,2)|2.73|0.3|2.01|0.10|33.0|1.88|
> |2|1.81|0.5|2.41|0.20|29.2|4.50|
> |5|1.65|1| 9.63|0.50|18.8|16.30|
>
> - **Q2: Empirical effect of SWC.** We have compared SWC with two widely used kernel matrix approximation methods: **random feature truncation** (Liu et al., 2021) and the **Nyström approximation** (Abedsoltan et al., 2024).
>   - Random Feature Truncation selects a fixed-dimensional subset of features by a low-dimensional random feature space.
>   - Nyström approximation selects a set of reference points approximate to the kernel matrix.
>
>   We apply all three kernel approximation methods to the three models in Example 5.1 (linear, logistic, and ReLU regression), using exactly the same parameter settings as in that example, see pages 30-31. To isolate the effect of approximation, no privacy noise is added. All methods are evaluated on prediction error and kernel computation time. For fairness, the baselines use a fixed feature budget of $M_t = 128$ while SWC adaptively selects its effective order $M_t$ via data-driven pruning based on the threshold $\kappa$. As reported in Table below, SWC achieves lower prediction error with fewer components $(M_t<128)$ and competitive kernel computation time. Figure 7 on page 30 further illustrates SWC's adaptivity.
>
> |Model|Metric|SWC|Random|Nyström|
> |-----|------|---|------|-------|
> |linear|MSE ($\times 10^{-3}$)|2.21|81.9|2.83|
> ||$M_t$|31|128|128|
> ||Time/s|$5.8\times10^{-3}$|$2.5\times10^{-5}$|$1.6\times10^{-2}$|
> |ReLU|MSE ($\times 10^{-3}$)|1.58|13.9|3.05|
> ||$M_t$|44|128|128|
> ||Time/s|$7.3\times10^{-3}$|$2.4\times10^{-5}$|$1.9\times10^{-2}$|
> |Logit|MSE ($\times 10^{-3}$)|4.27|41.8|6.73|
> ||$M_t$|61|128|128|
> ||Time/s|$9.6\times10^{-3}$|$2.5\times10^{-5}$|$2.1\times10^{-2}$|
>
> - **Q3: Non-stationary streaming data.** We have added experimental studies for non-stationary settings, focusing on parameter drift in the linear model of Example 5.1. These experiments use privacy parameters ($(\varepsilon,\delta)=(2,0.2)$) and a compression budget of $\kappa=0.1$ in $T = 20000$ samples. Following Barber et al. (2023), we consider two types of non-stationarity:
>   - **Case 1: Abrupt regime shifts.** The regression coefficient $\boldsymbol{\theta}$ switches among three fixed vectors over successive time segments: $$\theta^{(1)} = (1,2,1,0,0),\quad \theta^{(2)} = (0,-1,-2,-1,0),\quad \theta^{(3)} = (0,0,1,2,1),$$
>   - **Case 2: Smooth concept drift.** The regression coefficient evolves linearly from $$\theta^{(0)} = (1,2,1,0,0),\qquad \theta^{(1)} = (0,0,1,2,1),$$ according to $\theta_t = (1-\alpha_t)\theta^{(0)} + \alpha_t\\theta^{(1)}, \alpha_t = (t-1)/(T-1).$
>   Table  shows that LDP-BO consistently outperforms LDP-SGD in both cases, achieving lower prediction error and more stable performance under the same $(\varepsilon,\delta)$-LDP budget, and approaching the performance of the non-private baseline. The suboptimal result at 15,000 samples in Case 1 corresponds to the regime shift around 13,000 samples; with larger sample sizes, LDP-BO converges more rapidly than LDP-SGD.
>
> |Case|Privacy|$t$|LDP-BO|LDP-SGD|
> |----|-------------|---:|-------|-------|
> |Case 1|No DP|5,000|1.11 (0.58)|1.59 (0.72)|
> |||10,000|0.60 (0.31)|0.86 (0.39)|
> |||15,000|0.93 (0.48)|1.33 (0.60)|
> |||20,000|1.08 (0.56)|1.55 (0.70)|
> ||$\varepsilon = 2$|5,000|1.48 (0.76)|2.11 (0.95)|
> |||10,000|1.20 (0.62)|1.71 (0.77)|
> |||15,000|55.44 (28.51)|79.20 (35.64)|
> |||20,000|3.42 (1.76)|4.88 (2.20)|
> |Case 2|No DP|5,000|3.29 (1.70)|4.70 (2.12)|
> |||10,000|1.72 (0.89)|2.46 (1.11)|
> |||15,000|1.54 (0.79)|2.20 (0.99)|
> |||20,000|1.59 (0.82)|2.27 (1.02)|
> ||$\varepsilon = 2$|5,000|6.10 (3.14)|8.72 (3.92)|
> |||10,000|3.33 (1.71)|4.76 (2.14)|
> |||15,000|3.94 (2.02)|5.63 (2.53)|
> |||20,000|5.83 (3.00)|8.33 (3.75)|

---

> ### Author Response · Authors · 2025-11-21
> **Response Q4, Q5 and References**
>
> - **Q4: BO benefits.** We thank the reviewer for this comment. Although the theoretical convergence rates of our method are of the same order as those of noisy LDP-SGD, the primary strength of our method lies in its **gradient-free**, **model-based design**. LDP-BO requires only privatized function evaluations and leverages a Bayesian optimization surrogate to implicitly capture gradient information. This makes it particularly suitable for non-smooth, highly non-linear, or noisy objectives, where SGD-based methods may suffer from unstable or highly variable gradients. Similar benefits of gradient-free BO for complex optimization landscapes have been demonstrated in prior work (e.g., Müller et al., 2021; Sopa et al., 2025). Empirically, under the same $(\varepsilon,\delta)$-LDP budget, LDP-BO consistently achieves lower MSE than LDP-SGD on challenging models such as ReLU and logistic regression. These findings demonstrate the practical value of the surrogate-based BO approach, even when the asymptotic rates coincide.
>
> - **Q5: Reinforcement learning.** Our framework can naturally extend to reinforcement learning (RL) by applying LDP-BO to optimize the expected return $J(\theta)$ of a policy $\pi_{\theta}$. The BO loop operates over policy parameters, while local differential privacy is enforced on the observed returns.
>   - **Local privatization of returns**. At iteration $t$, the algorithm selects $\theta_t$, runs episodes under $\pi_{\theta_t}$, and locally privatizes the resulting return ($r_t$): $$
>    r_t = J(\theta_t) + \omega_t,
>   $$ where $\omega^j_t$ is properly calibrated LDP noise. Since only a scalar reward is privatized, sensitivity follows directly from standard bounded-reward assumptions in RL.
>   - **BO surrogate update with SWC**. The privatized return $r_t$ is incorporated into the BO surrogate. The privatized return is added to the kernel surrogate, and SWC maintains a compact dictionary,
>  $$
>   D_t = \text{SWC}(D_{t-1},\theta_t).
> $$ ensuring the model size does not grow with time and enabling continual RL operation.
>   - **Acquisition step**. The next policy parameter is chosen by minimizing the Gaussian information (GI) acquisition rule:
> $$
>  \theta_{t+1} = argmin_{\theta}\mathrm{GI}(\theta;D_{t},\theta_{t})=argmin_{\theta}\mathrm{Tr}(\nabla^{2}K_{D_t\cup\theta}\theta_{t},\theta_{t}))
> $$ yielding a fully online, privacy-preserving BO loop for policy search, where $K_{D_t\cup\theta}$ represents posterior covariance given $D_t\cup\theta$.
>   A promising direction for future work is to analyze how LDP noise affects the exploration--exploitation trade-off, building on BO-based RL approaches such as Wilson et al. (2014), Balakrishnan et al. (2020), and Müller et al. (2021).
>
> **References**
>
> Triastcyn A, Faltings B. Bayesian differential privacy for machine learning\[C\]//International Conference on Machine Learning. PMLR, 2020: 9583-9592.
>
> Zhang W, Zhang R. DP-Fast MH: Private, fast, and accurate Metropolis-Hastings for large-scale Bayesian inference\[C\]//International Conference on Machine Learning. PMLR, 2023: 41847-41860.
>
> Sopa G, Marusic J, Avella-Medina M, et al. Scalable Differentially Private Bayesian Optimization\[J\]. arXiv preprint arXiv:2502.06044, 2025.
>
> Duchi J C, Ruan F. The right complexity measure in locally private estimation: It is not the fisher information\[J\]. The Annals of Statistics, 2024, 52(1): 1-51.
>
> Xie J, Shi E, Jiang B, et al. Online differentially private inference in stochastic gradient descent\[J\]. arXiv preprint arXiv:2505.08227, 2025.
>
> Xiong X, Liu S, Li D, et al. A comprehensive survey on local differential privacy\[J\]. Security and Communication Networks, 2020, 2020(1): 8829523.
>
> Liu F, Huang X, Chen Y, et al. Random features for kernel approximation: A survey on algorithms, theory, and beyond\[J\]. IEEE Transactions on Pattern Analysis and Machine Intelligence, 2021, 44(10): 7128-7148.
>
> Abedsoltan A, Pandit P, Rademacher L, et al. On the Nyström approximation for preconditioning in kernel machines\[C\]//International Conference on Artificial Intelligence and Statistics. PMLR, 2024: 3718-3726.
>
> Barber R F, Candes E J, Ramdas A, et al. Conformal prediction beyond exchangeability\[J\]. The Annals of Statistics, 2023, 51(2): 816-845.
>
> Müller S, von Rohr A, Trimpe S. Local policy search with Bayesian optimization\[J\]. Advances in Neural Information Processing Systems, 2021, 34: 20708-20720.
>
> Wilson A, Fern A, Tadepalli P. Using trajectory data to improve bayesian optimization for reinforcement learning\[J\]. The Journal of Machine Learning Research, 2014, 15(1): 253-282.
>
> Balakrishnan S, Nguyen Q P, Low B K H, et al. Efficient exploration of reward functions in inverse reinforcement learning via Bayesian optimization\[J\]. Advances in Neural Information Processing Systems, 2020, 33: 4187-4198.

---

### Official Review · Reviewer_rwJt · 2025-11-07

**Soundness:** 3
**Presentation:** 2
**Contribution:** 3
**Rating:** 6
**Confidence:** 3

**Summary:**

An  online locally differentially private Bayesian optimization (LDP-BO) framework that enables zero-order optimization with rigorous privacy guarantees in dynamic environments is propoised in this work, which conducts Bayesian optimization (BO) in an online, streaming setting and provides Local Differential Privacy (LDP) guarantees. A systematic non-asymptotic convergence analysis to characterize the privacy–utility trade-off of the proposed estimators is carried out as well as experiments on both simulated and real-world datasets.

 demonstrate that our
method consistently delivers accurate, stable, and privacy-preserving results with-
out sacrificing efficiency.

The core technical contributions are a zeroth-order LDP optimizer based on Gaussian Process (GP) surrogate models and a Sliced Wasserstein Compression (SWC) algorithm to keep the GP's kernel dictionary from growing unboundedly.

**Strengths:**

- LDP-BO framework, enables zero-order optimization with rigorous privacy guarantees in dynamic environments.
- The framework for complex objective fuctions, thru a zeroth-order optimizer that eliminates the need for gradient information.
- Non-asymptotic analysis for the proposed framework.

**Weaknesses:**

- Ablations and sensitivity are thin.
- There are no comparisons to private kernel/GP baselines, private BO/bandit methods, or online CDP/LDP surrogates; the Uber-fares experiment even compares private GP to a linear LDP-SGD, which is mismatched in model capacity.
- Assumptions are strong.

**Questions:**

- Sensitivity and mechanism calibration need tightening.
- Convergence results require bounded gradients (Assump. 4.1), RKHS membership of 𝐿(⋅,𝑧_𝑡) (Assump. 4.2), and strong convexity/smoothness (Assump. 4.3) for Theorem 4.4, which do not hold for many realistic BO targets.

---

> ### Author Response · Authors · 2025-11-21
> **Response of W1, Q1 and Q2**
>
> Thank you for your thoughtful and constructive assessment. Many of your comments have been helpful and will allow us to produce a more readable and self-contained revision of the paper. Below, we address each of your specific concerns in detail.
> - **W1 \& Q1: Sensitivity analysis**. We have added comprehensive ablation and sensitivity studies. Specifically, we conduct these experiments on the linear regression model from Example 5.1 over 50 repetitions. We systematically vary three key parameters of our proposed LDP-BO procedure and evaluate their effect on the MSE ($\times10^{-3}$): the privacy budget $\varepsilon \in [0.5,10]$, the initial step size $\gamma_0 \in [0.1,1]$ in the schedule $\eta_t = \gamma_0 t^{-\alpha}$, and the compression threshold $\kappa \in [0.01,0.5]$.
> The findings indicate the following:
>   * **Privacy budget $\varepsilon$**: increasing $\varepsilon$ weakens privacy protection and consequently improves estimation accuracy;
>   * **Initial step size $\gamma_0$**: the proposed method is robust to the choice of initial step size over a broad range;
>   * **Compression threshold $\kappa$**: $\kappa$ induces a clear trade-off between estimation quality and runtime, with smaller values leading to faster execution but slightly reduced accuracy.
>
> | $\varepsilon$ | MSE | $\gamma_0$ | MSE | $\kappa$ | Time(min) | MSE |
> |-|-|-|-|-|--|-|
> |0.5|18.10|0.1|8.61|0.01|318.7|1.41|
> |1|3.4|0.2|1.83|0.05|165.3|1.59|
> |U(1,2)|2.73|0.3|2.01|0.10|33.0|1.88|
> |2|1.81|0.5|2.41|0.20|29.2|4.50|
> |5|1.65|1| 9.63|0.50|18.8|16.30|
> - **Q2: Private BO baselines**. Our work differs from existing methods in several fundamental ways:
>   - **Problem setting**. The proposed LDP-BO method targets online Bayesian optimization with zero-order updates and per-iteration local differential privacy. This places our framework in a problem regime not addressed by prior work. To better contextualize our contributions, The tables below compares the most relevant methods:
>
> | Reference| Method| DP| Bayesian|
> |-|-|-|-|
> |Triastcyn and Faltings(2020)| Offline |CDP|True|
> |Zhang and Zhang (2023)| Offline |CDP|True|
> |Sopa et al. (2025) | Offline |CDP|True|
> |Duchi and Ruan (2024)| Offline | LDP  |False|
> |Xie et al. (2025)| Online|LDP|False|
> |Proposed|Online |LDP|True|
>
> **Bayesian approaches** such as Triastcyn and Faltings (2020), Zhang and Zhang (2023), and Sopa et al. (2025) were all developed for offline settings under the CDP model. **LDP-based approaches** such as Duchi and Ruan (2024) and Xie et al. (2025) worked under the local DP model, with Xie et al. (2025) also considering an online regime, but neither established a Bayesian optimization framework.
> - To address the reviewer's request for private BO baselines, we evaluated the scalable DP-BO method by Sopa et al. (2025) on small synthetic and real datasets. In streaming settings, we benchmarked all methods on the linear model (Example 5.1) using only the first 2,000 observations. LDP-BO achieved comparable accuracy to offline DP-BO while requiring significantly less computation. In contrast, the runtime of offline DP-BO grows rapidly with sample size, as it repeatedly processes the full dataset. Even without SWC, single-point processing leads to continuous kernel-matrix growth and sharply increasing costs.
>
>
> ||Time (min)|||MSE($\times 10^{-3}$)|||
> |-|-|-|-|-|-|-|
> | $n$   | LDP-BO| LDP-BO (No SWC) | Offline| LDP-BO| LDP-BO (No SWC)  | Offline|
> | 200   | 0.30 | 0.38| 0.47 | 0.120 (0.020)| 0.110 (0.020) | 0.090 (0.015) |
> | 500   | 0.77| 1.37 | 2.47 | 0.090 (0.015)| 0.080 (0.014) | 0.055 (0.010)|
> | 1000  | 1.60| 5.58 | 12.40 | 0.075 (0.010)| 0.068 (0.009)| 0.045 (0.008)|
> | 1500  | 2.52| 35.08 | 57.95 | 0.060 (0.008)| 0.055 (0.007) | 0.043 (0.007)|
> | 2000  | 3.17| 151.92 | 183.25| 0.055 (0.007)| 0.052 (0.006)| 0.041 (0.006)|
>
>   - We have also compared the methods on small real-data subsets (first 2,000 observations). Using the Uber dataset (regression; prediction error) and the newly added Credit dataset (classification; accuracy),  All methods perform similarly at small scales; however, as the stream grows, offline DP-BO becomes computationally infeasible, whereas our online method continues to improve.
>
> || Uber|||Credit |||
> |-|-|-|-|-|-|-|
> |Sample Size|LDP-BO| DP-BO | LDP-SGD | LDP-BO | DP-BO | LDP-SGD |
> | 2000 | 5.471  | 5.129  | 17.412  | 0.941  | 0.944  | 0.913|
> | 5000| 2.224  | *| 10.271  | 0.944  | *| 0.929   |
> | 10000| 1.409  | *| 3.252   | 0.951  | *| 0.940   |
> | 20000 | 0.782  | * | 1.794   | 0.969  | *| 0.952   |

---

> ### Author Response · Authors · 2025-11-21
> **Response of W3 and Q2**
>
> - **W3 and Q2: Explanation and Weakening of Assumptions.**
> We clarify their roles, discuss practical relevance, and emphasize how our theory extends well beyond global strong convexity to a substantially weaker non-convex setting.
>    - **Assumption 4.1**: This condition ensures bounded $\ell_2$-sensitivity, enabling a DP mechanism with fixed noise scale. Many models, including linear regression with Huber loss and Mallows weights, as well as cross-entropy, pinball, hinge, and ReLU losses, satisfy this with mild robustification.
>      - A widely used alternative, **gradient clipping** (Abadi et al., 2017; Dong et al., 2022), enforces bounded gradients but introduces systematic bias and may be statistically inconsistent (Avella-Medina et al., 2023; Xie et al., 2025). The table below shows that Mallows weighting remains centered at the truth, whereas clipping produces persistent upward bias. This illustrates the practical advantage of robust weighting.
>      - Recent attempts (Minami et al., 2016; Simşekli et al., 2025) to avoid bounded-gradient assumptions entirely require much stronger distributional assumptions (e.g., heavy-tail stability or self-concordance). Relaxing Assumption 4.1 while retaining rigorous privacy and statistical guarantees remains an important future direction.
> | Method        | No DP  | $\varepsilon=2$ | $\varepsilon\in[1,2]$ | $\varepsilon=1$ |
> |---------------|--------|-----------------|------------------------|-----------------|
> | Mallow weights| 1.00 (0.02) | 0.99 (0.05)   | 1.02 (0.06)          | 0.98 (0.08)    |
> | Clipping      | 1.15 (0.02) | 1.18 (0.05)   | 1.20 (0.07)          | 1.19 (0.08)    |
>    - **Assumption 4.2**: This assumption, standard in GP-based BO (Wu et al., 2023; Sopa et al., 2025), requires each loss $L(\cdot;z_t)$ lie in the RKHS $\mathcal{H}=\mathrm{RKHS}(K)$ with a uniformly bounded RKHS norm. It is natural in practice because kernel choice like the squared-exponential or Matérn kernels, explicitly determines the function class. Alternatives such as Hölder or Lipschitz conditions (Kleinberg et al., 2008; Munos, 2011) are kernel-agnostic but require global smoothness constants, yield conservative dimension-dependent rates, and can be difficult to verify.
>
>    - **Assumption 4.3**: Although standard in stochastic approximation (Chen et al., 2020; Yu et al., 2024; Yang et al., 2024), global strong convexity is unrealistic for BO, which often involves multimodal objectives. Importantly, our theory is not confined to this setting. We have introduced significantly weaker conditions (Assumptions B.3–B.5, pages 18 in the revised manuscript), requiring only smoothness, local strong convexity near each global minimum, and a mild gap–distance condition. Under these assumptions, Corollary 5 (page 18) shows that the estimator $\hat{\boldsymbol{\theta}}_t$ converges to the set of global minimizers $\boldsymbol{\Theta}^{\mathrm{opt}}$ at the same rate $O(t^{-\alpha})$, as in the strongly convex case. The proof (Appendix B, pages 38–40) confirms that our guarantees hold in non-convex, multimodal settings where the minimizer is not unique.
>
>
> - **Q1: Calibration mechanism**.
>   - We now explicitly derive the per-sample sensitivity using a Mallows-type weighting scheme (Loh, 2017; Avella et al., 2023; Xie et al., 2025), which guarantees uniformly bounded gradients from each sample. For example, in the linear regression, the empirical loss is:
>   $$
> \mathcal{L}(\boldsymbol{\theta},\boldsymbol{x}_t,y_t) = \rho_c\bigl(y_t - \boldsymbol{x}_t^\top \boldsymbol{\theta}\bigr) \min\left(1,\frac{2}{\|\boldsymbol{x}_t\|^2}\right)
> $$
>
>     where Huber-type loss $\rho_c(\cdot)$ and Mallows-type weight $\min(1,2/\|\boldsymbol{x}_t\|^2)$ downweight influential covariates and yield finite $\ell_2$-sensitivity. We clarify how this method extends naturally to other common losses (cross-entropy, smoothed pinball) and to non-smooth objectives such as hinge-loss SVMs and ReLU models. While gradient clipping (Abadi et al., 2017; Dong et al., 2022) also enforces bounded gradients, it introduces bias and can be inconsistent (Avella-Medina et al., 2023; Xie et al., 2025). Our empirical comparison (W3) demonstrates the advantage of Mallows-type weighting.
>
>   - Our framework is compatible with standard DP mechanisms—Gaussian, Laplace, GDP (Dong et al., 2022), RDP (Mironov, 2017), etc., as long as the noise scale is calibrated using the derived sensitivity. The table provides a clearer and unified description of calibration across mechanisms.
>
> |             | 5000  | 10000 | 15000 | 20000 |
> |---------------|-------|-------|-------|-------|
> | ($\varepsilon,\delta$)-DP | 6.05 (1.10) | 1.62 (0.35) | 0.976 (0.21) | 0.513 (0.12) |
> | $\varepsilon$-DP | 6.80 (1.25) | 1.85 (0.40) | 1.10 (0.25) | 0.600 (0.14) |
> | $\mu$-GDP | 3.81 (0.75) | 1.34 (0.28) | 0.66 (0.14) | 0.34 (0.08) |
> | RDP | 4.46 (0.85) | 1.46 (0.30) | 0.77 (0.16) | 0.46 (0.11) |

---

> ### Author Response · Authors · 2025-11-21
> **References of response**
>
> Triastcyn A, Faltings B. Bayesian differential privacy for machine learning[C]//International Conference on Machine Learning. PMLR, 2020: 9583-9592.
>
> Zhang W, Zhang R. DP-Fast MH: Private, fast, and accurate Metropolis-Hastings for large-scale Bayesian inference[C]//International Conference on Machine Learning. PMLR, 2023: 41847-41860.
>
> Sopa G, Marusic J, Avella-Medina M, et al. Scalable Differentially Private Bayesian Optimization[J]. arXiv preprint arXiv:2502.06044, 2025.
>
> Duchi J C, Ruan F. The right complexity measure in locally private estimation: It is not the fisher information[J]. The Annals of Statistics, 2024, 52(1): 1-51.
>
> Xie J, Shi E, Jiang B, et al. Online differentially private inference in stochastic gradient descent[J]. arXiv preprint arXiv:2505.08227, 2025.
>
> Loh P L. Statistical consistency and asymptotic normality for high-dimensional robust M-estimators[J]. Annals of Statistics, 2017 ,45(2), 866-896.
>
> Avella-Medina M, Bradshaw C, Loh P L. Differentially private inference via noisy optimization[J]. The Annals of Statistics, 2023, 51(5): 2067-2092.
>
> Dong J, Roth A, Su W J. Gaussian differential privacy[J]. Journal of the Royal Statistical Society Series B: Statistical Methodology, 2022, 84(1): 3-37.
>
> Minami K, Arai H I, Sato I, et al. Differential privacy without sensitivity[J]. Advances in Neural Information Processing Systems, 2016, 29.
>
> Şimşekli U, Gürbüzbalaban M, Yıldırım S, et al. Privacy of SGD under Gaussian or Heavy-Tailed Noise: Guarantees without Gradient Clipping[J]. arXiv preprint arXiv:2403.02051, 2024.
>
> Wu K, Kim K, Garnett R, et al. The behavior and convergence of local bayesian optimization[J]. Advances in Neural Information Processing Systems, 2023, 36: 73497-73523.
>
> Kleinberg R, Slivkins A, Upfal E. Multi-armed bandits in metric spaces[C]//Proceedings of the Fortieth Annual ACM Symposium on Theory of Computing. 2008: 681-690.
>
> Munos R. Optimistic optimization of a deterministic function without the knowledge of its smoothness[J]. Advances in Neural Information Processing Systems, 2011, 24.
>
> Chen X, Lee J D, Tong X T, et al. Statistical inference for model parameters in stochastic gradient descent[J]. The Annals of Statistics, 48(1):251 – 273, 2020.
>
>
> Yu Q, Wang Y, Huang B, et al. Stochastic zeroth-order optimization under strongly convexity and lipschitz hessian: minimax sample complexity[J]. Advances in Neural Information Processing Systems, 2024, 37: 99564-99600.
>
> Yang W, Wang Y, Zhao P, et al. Universal Online Convex Optimization with $1 $ Projection per Round[J]. Advances in Neural Information Processing Systems, 2024, 37: 31438-31472.
>
> Mironov I. Rényi differential privacy[C]//2017 IEEE 30th Computer Security Foundations Symposium (CSF). IEEE, 2017: 263-275.

---

### Author Response · Authors · 2025-12-02
**Summary**

We sincerely thank the reviewers, ACs, SACs, and PCs for their time, constructive feedback, and thoughtful discussions. In our rebuttal, we addressed each point in detail, and in the revision, we have carefully incorporated the suggested changes, including:

- **Sensitivity analysis.** We conducted sensitivity analyses for the privacy budget $\epsilon$, initial step size $\gamma_0$, and compression threshold $\kappa$. The results show:
  1. Larger $\epsilon$ improves accuracy at a privacy cost,
  2. The method is robust to a wide range of $\gamma_0$,
  3. Smaller $\kappa$ accelerates computation with only marginal accuracy loss (Section D.2, pp. 27–28).

- **Private BO baselines.** We clarified the limitations of existing privacy-preserving BO methods and added comparisons on small datasets, showing that our method achieves accuracy comparable to offline DP-BO with substantially lower computational cost (Section E.1, pp. 31–32).

- **Weakening of Assumptions.** We relaxed the bounded-gradient and strong-convexity assumptions while retaining similar convergence rates (Corollary B.6, p. 18; Section E.2, p. 32).

- **Additional simulations.** We expanded simulations to higher dimensions, non-stationary streaming data, and alternative privacy mechanisms. Across all settings, the proposed method remains robust:
  - Non-stationary: Section D.2, pp. 28–29
  - High dimension: Table 3, p. 24
  - Mechanisms: Section D.5, pp. 31–32

- **Practical motivation.** We strengthened the discussion on the practical necessity of online privacy-preserving BO and provided additional real-world analyses (pp. 1–2, lines 49–63; p. 10, lines 510–524).

- **Advantages of SWC.** We added a proof of SWC’s computational complexity, clarified its role, and provided theoretical and empirical comparisons demonstrating its per-iteration efficiency and scalability compared with fixed-feature methods:
  - Empirical: Section D.4, pp. 29–30
  - Complexity: Section E.1, p. 31

- **Budget allocation.** We highlighted that using local DP removes the need for privacy budget allocation (Definition A.1; Proposition A.5, p. 16).

- **Extensions.** We expanded the discussion of possible extensions to federated learning and reinforcement learning (Section E.5, pp. 33–34).

Beyond the points summarized above, we have also made several additional clarifications and minor revisions; all changes are highlighted in blue.

Once again, we thank you for your time and professional engagement throughout the review process.

---

### Meta-Review · Area_Chair_r38q · 2026-01-07

**Summary:**

The paper proposes an online Bayesian Optimization framework with Local Differential Privacy (LDP-BO) using Sliced Wasserstein Compression (SWC) to bound kernel dictionary size. It theoretically analyzes convergence under LDP and validates the method on synthetic functions and an Uber dataset.

**Reviewer Concerns:**

The authors effectively clarified the computational complexity of SWC and added sensitivity analyses. However, concerns regarding the restrictiveness of theoretical assumptions (bounded gradients, RKHS membership) and the sufficiency of baselines against state-of-the-art scalable GP methods remain significant hurdles to acceptance.

**Reviewer Scores:**

Reviewer rwJt: Likely would have maintained a 6 or dropped to 5; while baselines were added, the core concern about strong theoretical assumptions (Assumptions 4.1-4.3) restricting real-world applicability was not fully resolved.

Reviewer Cyco: Likely would have raised to a 4 but remained negative; the practical necessity of online LDP-BO versus batch methods remains unconvincing, and the specific overhead of SWC in high dimensions is still a concern.

Reviewer 84Py: Likely would have maintained a 6; the rebuttal addressed non-stationarity and runtime well, but the handling of privacy composition in infinite streams likely prevents a stronger endorsement.

---

### Decision · Program_Chairs · 2026-01-26

Reject